



# Synergistic radar and radiometer retrievals of ice hydrometeors

Simon Pfreundschuh[1], Patrick Eriksson[1], Stefan A. Buehler[2], Manfred Brath[2], David Duncan[4], Richard Larsson[3], and Robin Ekelund[1]

[1]Department of Space, Earth and Environment, Chalmers University of Technology, 41296 Gothenburg, Sweden
[2]Meteorologisches Institut, Fachbereich Geowissenschaften, Centrum für Erdsystem und Nachhaltigkeitsforschung (CEN), Universität Hamburg, Bundesstraße 55, 20146 Hamburg, Germany
[3]Max Planck Institute for Solar System Research, Justus-von-Liebig-Weg 3, 37077 Göttingen, Germany
[4]European Centre for Medium-Range Weather Forecasts, Shinfield Park, Reading RG2 9AX, United Kingdom

**Correspondence:** Simon Pfreundschuh (simon.pfreundschuh@chalmers.se)

**Abstract.** The upcoming Ice Cloud Imager (ICI) radiometer, to be launched on board the second generation of European operational meteorological satellites (Metop-SG), will be the first microwave imager to provide sub-millimeter observations of the atmosphere. The Microwave Imager (MWI) radiometer will be flown on the same satellites and complement the ICI sensor with observations at traditional millimeter wavelengths. The addition of these two new passive microwave sensors to the global system of earth observation satellites opens up opportunities for synergistic satellite missions aiming to maximize the scientific return of the Metop-SG program. This study analyzes the potential benefits of combining observations of the MWI and ICI radiometers with a 94-GHz cloud radar for the retrieval of frozen hydrometeors. Starting from a simplified numerical experiment, it is shown that the complementary information content in the radar and radiometer observations can help to better constrain the particle size distribution of ice particles in the atmosphere. The feasibility of the combined retrieval is demonstrated by applying a one-dimensional, variational cloud-retrieval algorithm to simulated observations from a high-resolution atmospheric model. Comparison of the results with passive- and radar-only versions of the retrieval algorithm confirms that synergies between the active and passive observations allow an improved retrieval of microphysical properties of frozen hydrometeors. The effect of the assumed ice particle shape on the results is analyzed and found to be critical for obtaining good retrieval performance. In addition to this, the synergistic retrieval shows improved sensitivity to liquid water in both warm and supercooled clouds. The results of this study clearly demonstrate the potential of the combined observations to constrain the microphysical properties of ice hydrometeors, which can help to reduce errors in retrieved profiles of mass- and number densities.

## 1 Introduction

Ice clouds play an important role in many weather- and climate-related processes in the atmosphere. They interact with incoming and outgoing radiation and thus influence the Earth's energy budget. Moreover, as part of the global hydrological cycle and due to their relation to the dynamics of the atmosphere (Bony et al., 2015), observations of ice clouds provide important information to constrain the state of the atmosphere in numerical weather prediction (NWP) models (Geer et al., 2017) as well as to validate predictions from climate models (Waliser et al., 2009).


Despite the importance of observations of ice clouds for climate and weather prediction, today's global observing system cannot provide accurate information on the global distribution of ice in the atmosphere (Eliasson et al., 2011; Duncan and Eriksson, 2018). The main difficulty in sensing atmospheric ice from space is the large variability of sizes and concentrations in which ice particles occur in the atmosphere. The wide spectrum of ice crystal sizes, which ranges from micro- to millimeter scales, can only be partially resolved by currently available space-borne sensors.

The sensitivity of a remote sensing system to ice particles of a given size is determined mainly by its observing frequencies. The scattering of radiation by ice particles is strongest for sizes roughly equal to the wavelength, $\lambda$, of the radiation. For particles with sizes much smaller than $\lambda$, the sensitivity decreases rapidly, making them practically invisible to the sensor. Although the strength of the interaction between particles and radiation decreases as the wavelength becomes much larger than the particle size, it remains strong enough for the cloud signal to saturate in the presence of thicker clouds, leading to loss of sensitivity further down the line of sight.

The observing frequencies that are currently available for measuring ice from space are limited to the microwave, infrared and optical domain. Infrared and optical sensors provide sensitivity to small ice particles but cannot sense significant parts of the ice mass of thicker clouds due to saturation of the signal. Microwave observations, in contrast, provide sensitivity throughout the whole atmospheric column but are insensitive to small ice particles. Although radars and lidars generally provide greater sensitivity than their passive counterparts, they are ultimately limited by the same principles.

To narrow the size-sensitivity gap between the infrared and traditional microwave sensors, the upcoming Ice Cloud Imager (ICI) will extend the microwave frequencies available for studying clouds with channels at 243, 325, 448 and 664 GHz (Eriksson et al., 2019). This extension of the smallest currently available microwave wavelength from 1.6 mm at 183 GHz down to the sub-millimeter domain (0.45 mm at 664 GHz) will significantly improve the size-sensitivity of space-borne microwave observations of clouds.

Together with ICI, also the newly developed Microwave Imager (MWI) will be flown on the satellites of the Metop-SG program. MWI will complement ICI's observations with measurements at traditional millimeter wavelengths. The observations of MWI, which cover the frequency range from 19 GHz up to 183 GHz, will provide additional sensitivity to liquid and frozen precipitation as well as water vapor.

The advent of space-borne sub-millimeter radiometry of clouds brings with it great potential for the study ice in the atmosphere. The information content and retrieval performance from radiometer observations alone has been studied in detail for column-integrated ice mass (Jiménez et al., 2007; Wang et al., 2017; Brath et al., 2018) as well as for the vertical distribution of ice in the atmosphere (Birman et al., 2017; Grützun et al., 2018; Aires et al., 2019). Also the concept of combining millimeter and sub-millimeter radiometer observations with active observations from a cloud radar has been investigated (Evans et al., 2005; Jiang et al., 2019).

This work applies the concept of synergistic radar and sub-millimeter radiometer retrievals to the upcoming ICI and MWI sensors by combining them with a conceptual W-band cloud radar. It extends previous studies on this observational technique by providing an in-depth analysis of the fundamental synergies between the active and passive observations that help to improve the retrieval ice in the atmosphere. In particular, this study investigates to which extent the combined active and passive





observations can constrain the microphysics of ice particles in the atmosphere. Starting from a simplified numerical experiment,

the complementarity of the information content of the active and passive observations is demonstrated. In addition to this, simulated results from a synergistic, variational cloud-retrieval algorithm are presented. The algorithm is applied to synthetic observations of cloud scenes from a high-resolution atmospheric model and used to further explore the synergies between the active and passive observations.

     The presented research has been conducted as part of a larger study funded by the European Space Agency, which evaluated

the concept of a future radar mission to fly in constellation with ICI on board the satellites of the Metop-SG program. Inspired by the concept of the Global Precipitation Measurement (GPM, Hou et al. (2014)) mission, the approach of this tentative mission is to perform vertically-resolved, high-accuracy retrievals of hydrometeors from the co-located active and passive observations at the swath center of the passive imager. The results of combined retrieval could then be used to constrain passive-only profile retrievals with the aim of extending the profiling capabilities of the radar to the wide swath of the passive

imager.

     Following this introduction, Section 2 introduces the test data, sensor configuration and the developed retrieval algorithm on which the study is based. This is followed by the experimental results on the information content of the combined observations and the retrieval results of the joint retrieval on selected test scenes in Section 3. The article closes with a discussion of the results in Section 4 and conclusions in Section 5.

**2   Methods and data**

The synergistic retrieval is tested using simulated observations of cloud scenes from a high-resolution atmospheric circulation model. This section presents the selected reference cloud scenes, sensor configuration and basic modeling assumptions used in the radiative transfer simulations. In addition to this, the theoretical formulation of the combined cloud-retrieval algorithm is introduced.

**2.1   Reference cloud scenes**

The cloud scenes that will be used for the testing of the retrieval were produced by Environment and Climate Change Canada using a high-resolution NWP configuration of the Global Environmental Multiscale (GEM) Model (Côté et al. (1998)). For this study, we restrict ourselves to two designated, two-dimensional test scenes, which are displayed in Fig. 1. The test scenes have a horizontal resolution of 1 km and both extend over 800 km. The scenes were chosen with the aim of covering a large

range of cloud structures and compositions so as to ensure a realistic assessment of the retrieval. The first test scene, shown in panel (a), is located in the tropical Pacific and contains a convective storm system in the right half of the scene and its anvil that extends into the left half of the scene. The second scene, shown in panel (b), is located in the North Atlantic and contains an ice cloud in the first quarter and a low-level, mixed-phase cloud in the remainder of the scene.

     The GEM model uses six types of hydrometeors to represent clouds and precipitation (Milbrandt and Yau, 2005): Two

classes of liquid hydrometeors (rain and liquid cloud) and four of frozen hydrometeors (cloud ice, snow, hail and graupel).





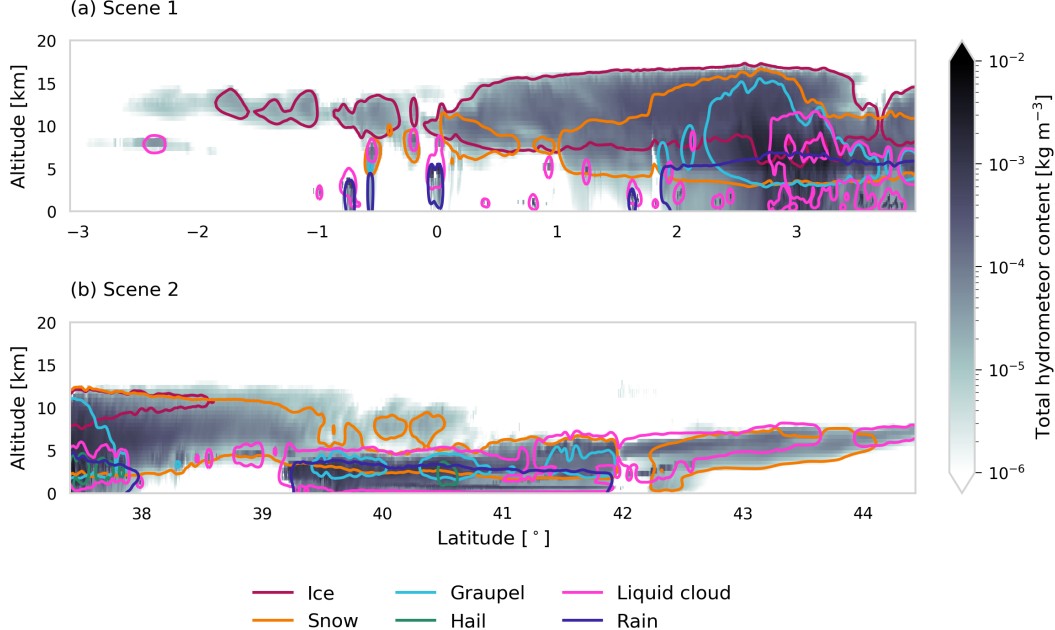

**Figure 1.** The distribution of total hydrometeor mass content in the two cloud scenes used to test the retrieval. Colored lines show the $m = 10^{-5}$ kg m$^{-3}$ contour for different hydrometeor species.

**Table 1.** Particle-model names, IDs and parameters $\alpha, \beta$ of the mass-size relationships $m = \alpha D_{\max}^{\beta}$, where $D_{\max}$ is the maximum diameter of the particle. The ID column contains the particle shape identifier of the particle model in the Eriksson et al. (2018) scattering database.

| Hydrometeor species | Particle shape | ID | $\alpha$ | $\beta$ |
|---|---|---|---|---|
| Cloud ice | GemCloudIce | 31 | 440 | 3 |
| Snow | GemSnow | 32 | 52.4 | 3 |
| Graupel | GemGraupel | 33 | 209.4 | 3 |
| Hail | GemHail | 34 | 471.2 | 3 |
| Rain | LiquidSphere | 25 | 523.6 | 3 |
| Liquid cloud | LiquidSphere | 25 | 523.6 | 3 |

The particle size distribution (PSD) of each hydrometeor type is parametrized by its particle number concentration and mass density. The full particle size distribution can be prognosed from the two moments using a species-dependent parametrization and mass-size relationship. The parameters of the mass-size relationship are given in Tab. 1. As shown in the table, the masses of all ice particles in the model are assumed to scale with a power of three, which leads to high densities for large particles.

Examples of particle size distributions of frozen hydrometeors are displayed in Fig. 2. The four panels display the prognosed particle size distributions for the four frozen hydrometeor types together with renderings of the particle shapes used in the





forward simulations. As these plots show, the assumed particle size distributions across different ice species vary mostly in their horizontal and vertical scaling, whereas the function shape shows less variability. Furthermore, an important characteristic of the model can be identified here, which will help to better understand the retrieval results presented later: Cloud ice in the model
is characterized by high particle number densities and small particle sizes, whereas snow exhibits lower number concentrations and larger particles.

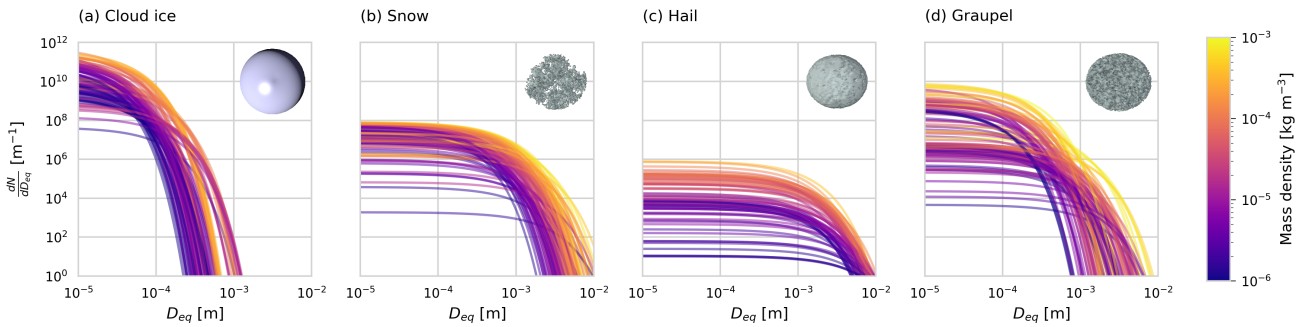

**Figure 2.** Realizations of particle size distributions from the cloud scenes used in this study. The number particle density is plotted with respect to the volume-equivalent diameter $D_{\mathrm{eq}}$. Shown are the PSDs corresponding to 100 randomly chosen grid points with a mass concentration higher than $10^{-6}$ kg m$^{-3}$. Line color encodes the corresponding mass density.

## 2.2   Simulated cloud observations

A simulated observation is generated for each vertical profile in the model test scenes. The simulations apply the same microphysics scheme as the model, which means that they use the same six hydrometeor classes and PSD parametrizations.

### 2.2.1   Sensor configuration

Simulations of observed passive brightness temperatures are performed for the 11 highest-frequency channels of the MWI radiometer and all channels of the ICI radiometer. The passive observations are combined with a W-band cloud radar similar to the CloudSat Cloud Profiling Radar (CPR) (Stephens et al., 2002; Tanelli et al., 2008).

A number of simplifications are applied for the generation of the synthetic cloud observations: Firstly, the beams of all three
sensors are modeled as perfectly coincident pencil beams. Secondly, a synthetic observation is generated for each vertical profile from the model scenes by simulating a one-dimensional, plane-parallel atmosphere, the properties of which are taken from the corresponding model profile. It follows from these modeling decisions that the atmosphere is assumed to be homogeneous across the beams of the active and passive sensors and that they all sense the same atmospheric volume. This is certainly not the case for space-borne observations and will incur a forward modeling error that is not accounted for in this study. Since the
focus of this study are the fundamental synergies between the active and passive observations, quantifying the impact of beam width and inhomogeneity is left for future investigation.



Observations from the ICI radiometer are simulated by performing a single, non-polarized radiative transfer simulation located at the centers of the pass bands of each channel and averaging the resulting brightness temperatures. For channels with multiple polarizations, only a single simulation is performed. To compensate for this, the noise of the corresponding channel is reduced by a factor of $\sqrt{2}$. The simulated ICI channels and assumed noise levels are presented in Tab. 2. The off-nadir viewing angle of ICI is assumed to be $48°$ at the sensor.

Observations from the MWI radiometer are simulated in a similar manner as those from ICI. However, from MWI only channels with frequencies larger than or equal to 89 GHz are used. The reason for this is that the footprints of the channels with frequencies lower than 89 GHz have full-width at half maximum of 50 km compared to only 10 km for the higher-frequency channels. Due to the very small overlap of the footprints of these low-frequency channels with that of the radar, it is assumed they would not be beneficial for a synergistic retrieval and are therefore disregarded here. The included MWI channels are listed in Tab. 2.

**Table 2.** Channels of the MWI and ICI radiometers used in the retrieval.

| | MWI | | | | ICI | | |
| --- | --- | --- | --- | --- | --- | --- | --- |
| Channel | Freq. [GHz] | Noise [K] | | Channel | Freq. [GHz] | Noise [K] |
| MWI-8 | 89 | 1.1 | | ICI-1 | $183.31 \pm 7.0$ | 0.8 |
| MWI-9 | $118.75 \pm 3.2$ | 1.3 | | ICI-2 | $\pm 3.4$ | 0.8 |
| MWI-10 | $\pm 2.1$ | 1.3 | | ICI-3 | $\pm 2.0$ | 0.8 |
| MWI-11 | $\pm 1.4$ | 1.3 | | ICI-4 | $243 \pm 2.5$ | $\frac{1}{\sqrt{2}} \cdot 0.7$ |
| MWI-12 | $\pm 1.2$ | 1.3 | | ICI-5 | $325.15 \pm 9.5$ | 1.2 |
| MWI-13 | $165.5 \pm 0.75$ | 1.3 | | ICI-6 | $\pm 3.5$ | 1.3 |
| MWI-14 | $183.31 \pm 7.0$ | 1.2 | | ICI-7 | $\pm 1.5$ | 1.5 |
| MWI-15 | $\pm 6.1$ | 1.2 | | ICI-8 | $448 \pm 7.2$ | 1.4 |
| MWI-16 | $\pm 4.9$ | 1.2 | | ICI-9 | $\pm 3.0$ | 1.6 |
| MWI-17 | $\pm 3.4$ | 1.2 | | ICI-10 | $\pm 1.4$ | 2.0 |
| MWI-18 | $\pm 2.0$ | 1.3 | | ICI-11 | $664 \pm 4.2$ | $\frac{1}{\sqrt{2}} \cdot 1.6$ |

The frequency of the the cloud radar is chosen to be 94 GHz similar to CloudSat CPR. The vertical resolution of the radar observations is assumed to be 500 m ranging from 0.5 to 20 km in altitude. The minimum sensitivity is set to be $-30$ dBZ and the noise at each range gate is modeled to be independent with standard deviation 0.5 dBZ. As mentioned above, the same incidence angle as for the passive radiometers is assumed also for the radar. In practice, this could be achieved by remapping the radar observations to the lines of sights of the passive beams.

### 2.2.2 Radiative transfer simulations

All simulations presented in this study were performed using Version 2.3.1245 of the Atmospheric Radiative Transfer Simulator (ARTS, Buehler et al. (2018)). Radar reflectivities are computed using ARTS' built-in single-scattering radar solver. For the





simulation of passive radiances, a hybrid solver is used which combines the DISORT (Stamnes et al., 2000) scattering solver with the ARTS standard scheme for pencil beam radiative transfer. The hybrid solver has been added to ARTS specifically for this study and provides approximate, analytical Jacobians, which are required for the variational retrievals of hydrometeors. All simulations are performed assuming an ocean surface with emissivities calculated using the Tool to Estimate Sea-Surface

Emissivity from Microwaves to sub-Millimeter waves (TESSEM, Prigent et al. (2017)). Polarization is neglected in all simulations performed in this study. Particle scattering data are taken from the ARTS scattering data base (hereafter ARTS SSDB, Eriksson et al. (2018)). Gaseous absorption is modeled using the absorption models from Rosenkranz (1993) for $N_2$, $O_2$ and from Rosenkranz (1998) for $H_2O$.

## 2.3 Retrieval algorithm

A one-dimensional, variational cloud retrieval algorithm is proposed to retrieve distributions of frozen hydrometeors from the combined active and passive observations. The algorithm uses the optimal estimation method (OEM) developed by Rodgers (2000). The retrieved state $\mathbf{x} \in \mathbb{R}^n$ is determined by fitting a forward model $\mathbf{F} : \mathbb{R}^n \to \mathbb{R}^m$ to a set of observations $\mathbf{y} \in \mathbb{R}^m$. The best fit is determined by minimizing a cost function of the form

$$\mathcal{L}(\mathbf{x},\mathbf{y}) \propto (\mathbf{F}(\mathbf{x}) - \mathbf{y})^T \mathbf{S}_e^{-1} (\mathbf{F}(\mathbf{x}) - \mathbf{y}) + (\mathbf{x} - \mathbf{x}_a)^T \mathbf{S}_a^{-1} (\mathbf{x} - \mathbf{x}_a). \tag{1}$$

The cost function $\mathcal{L}(\mathbf{x},\mathbf{y})$ corresponds to the negative log-likelihood of the a posteriori distribution of the state $\mathbf{x}$ under the assumptions of Gaussian a priori distribution with mean $\mathbf{x}_a$ and covariance matrix $\mathbf{S}_a$ as well as zero-mean Gaussian measurement error with covariance matrix $\mathbf{S}_e$.

To assess the quality of a retrieved state $\hat{\mathbf{x}}$ and corresponding simulated observation $\hat{\mathbf{y}} = \mathbf{F}(\hat{\mathbf{x}})$, we define the following diagnostic quantity

$$\chi_y^2 = \delta\mathbf{y}^T \mathbf{S}_e^{-1} \delta\mathbf{y}, \tag{2}$$

where $\delta\mathbf{y} = \mathbf{y} - \hat{\mathbf{y}}$. The quantity $\chi_y^2$ is here used to approximate a $\chi^2$-test for the misfit between the observations $\mathbf{y}$ and the retrieval fit $\hat{\mathbf{y}}$. Although a formally correct $\chi^2$-test for $\delta\mathbf{y}$ should apply a different covariance matrix (c.f. Chapter 12 in Rodgers (2000)), such tests were found to yield very high values that deviate strongly from the expected chi-square distribution. The $\chi_y^2$ value used here provides a less strict test in the sense that it will generally be smaller than if the formally correct covariance

matrix was used.

The amount of information contained in a retrieval can be quantified by computing the degrees of freedom for signal (DFS). Let $\mathbf{K} \in \mathbb{R}^{m \times n}$ be the Jacobian of the forward model $\mathbf{F}$. Then the DFS of the observations can be computed as the trace of the averaging kernel matrix

$$\mathbf{A} = (\mathbf{K}^T \mathbf{S}_e^{-1} \mathbf{K} + \mathbf{S}_a^{-1})^{-1} \mathbf{K}^T \mathbf{S}_e^{-1} \mathbf{K}. \tag{3}$$





### 2.3.1 Measurement space

The input for the retrieval algorithm is the combined observation vector **y** consisting of the concatenated single-instrument observations from the cloud radar and the two radiometers. Measurement errors are assumed to be independent and Gaussian distributed with standard deviations according to the noise characteristics given in Section 2.2.1.

### 2.3.2 State space

The proposed retrieval solves for distributions of one frozen and one liquid hydrometeor species in the atmospheric column together with profiles of atmospheric humidity and liquid-cloud mass density. The retrieval uses the same vertical grid as the model scenes, which have a vertical resolution of about $500\,\mathrm{m}$ throughout the troposphere. If not specified otherwise, retrieval quantities are retrieved at this resolution.

Distributions of hydrometeors in the atmospheric column are represented using the normalized particle size distribution formalism proposed by Delanoë et al. (2005). The PSD of a hydrometeor species at a given height level is represented by a vertical and a horizontal scaling parameter, the mass-weighted mean diameter $D_m$ and the normalized number density $N_0^*$. Alternative parametrizations using mass density and $D_m$ or the mass density and $N_0^*$ have been tested but no considerable effect on retrieval performance has been observed.

The retrieval computes vertical profiles of the two scaling parameters $D_m$ and $N_0^*$ for each of the two hydrometeor species. The remaining shape of each PSD is described by the shape parameters $\alpha$ and $\beta$, not to be confused with the parameters of the mass-size relationship shown in Tab. 1. The shape parameters are set to fixed, species-specific values. This principle is illustrated in Fig. 3. The plot displays the a-priori-assumed shapes of the particle size distribution of frozen and liquid hydrometeors. The retrieved horizontal and vertical scaling parameters, $D_m$ and $N_0^*$, are used as units for the axes of the plot so that the shape of the PSD becomes independent of the retrieved mass density and number concentration. For frozen hydrometeors, the values of the shape parameters $\alpha$ and $\beta$ are chosen identical to version 3 of the DARDAR-CLOUD product (Cazenave et al., 2018). For liquid hydrometeors, the shape parameters are chosen so that they are equivalent to the shape used by the GEM model for rain drops. All calculations involving particles size distributions use the volume-equivalent diameter $D_{\mathrm{eq}}$ as size variable.

The temperature-dependent a priori profile for $N_0^*$ of frozen hydrometeors is determined using the relation from Delanoë et al. (2014)

$$N_0^* = \exp\left(-0.076586 \cdot (T - 272.5) + 17.948\right), \tag{4}$$

where $T$ is in K. The a priori profile for $D_m$ for frozen hydrometeors is chosen so that the a priori mass density is equal to $10^{-6}\,\mathrm{kg\,m^{-3}}$. For liquid hydrometeors, a fixed value for $N_0^*$ of $10^6\,\mathrm{m^4}$ is assumed and the a priori profile for $D_m$ is determined similarly as for frozen hydrometeors. Values of the mass-weighted mean diameter $D_m$ for both hydrometeor species are retrieved in linear space, whereas the normalized number concentration parameter $N_0^*$ is retrieved in $\log_{10}$ space. As additional constraints, the retrieval of frozen hydrometeors is restricted to the region between the freezing layer and the tropopause, whereas the retrieval of liquid hydrometeors is restricted to below the freezing layer.





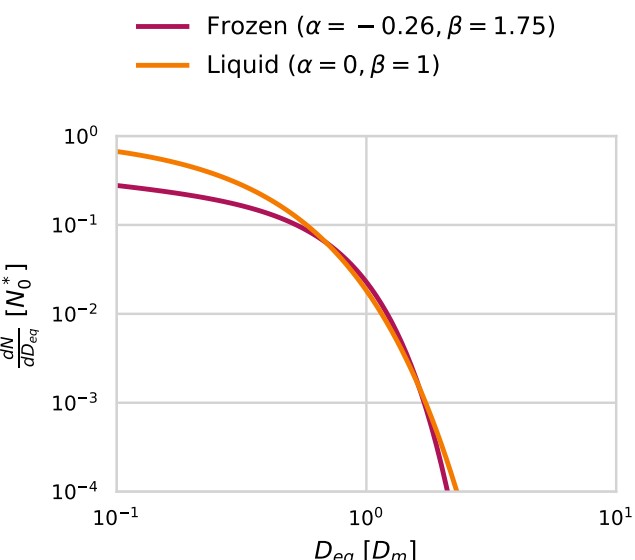

**Figure 3.** PSD parametrizations for frozen and liquid hydrometeors used in the cloud retrieval.

To further regularize the retrieval, $N_0^*$ for ice is retrieved at only 10 equally-spaced grid points between freezing layer and the tropopause. Similarly, $D_m$ and $N_0^*$ for rain are retrieved at 10 respectively 4 points between surface and freezing layer. This was necessary to avoid the retrieval from getting stuck in spurious local minima. An approach similar to this one is also taken in the GPM combined precipitation retrievals (Grecu et al., 2016).

Humidity in the atmospheric column is retrieved in units of relative humidity at a vertical resolution of $1\,\mathrm{km}$. However, instead of retrieving relative humidity directly, an inverse hyperbolic tangens transformation is applied to the relative humidity profile $\phi$:

$$x = \mathrm{arctanh}(\frac{2\phi}{1.1} - 1.0) \tag{5}$$

The transformation restricts the retrieved relative humidity values to the range of $[0.0, 1.1]$. The a priori profile for relative humidity is arbitrarily chosen as

$$\phi(t) = \begin{cases} 0.7 & , 270\,\mathrm{K} < t \\ 0.7 - 0.01 \cdot (t - 270) & , 220 < t \leq 270\,\mathrm{K} \\ 0.2 \cdot (t - 270) & , t < 220 \end{cases} \cdot \tag{6}$$

The retrieval of liquid cloud mass density, here referred to as liquid water content (LWC), is performed at seven equally spaced altitude levels between the surface and the 230 K isotherm. In contrast to frozen and liquid hydrometeors, cloud water is modeled in the retrieval forward model to be purely absorbing using the absorption model by Liebe et al. (1993) for suspended





liquid cloud droplets. Liquid cloud mass density is retrieved in $\log_{10}$-space and the a priori profile is set to a fixed value of $10^{-6} \text{ kg m}^{-3}$ in the permitted region of the atmosphere.

The a priori distributions of the 6 retrieval quantities ($N_0^*$ and $D_m$ for frozen and liquid hydrometeors, relative humidity
$\phi$, cloud water) are assumed to be independent so that the overall a priori covariance matrix $\mathbf{S}_a$ has block-diagonal structure. Within each block, vertical correlations between the values of a given retrieval quantity at different altitudes are assumed to be exponentially decaying. Hence, the correlation of the values of retrieval quantity $q$ at points $i$ and $j$ of the retrieval grid is computed as

$$(\mathbf{S}_{a,q})_{i,j} = \sigma_{q,i} \sigma_{q,j} \cdot \exp\left(-\frac{d(i,j)}{l_q}\right), \tag{7}$$

where $\sigma_{q,i}$ is the a priori uncertainty assumed for retrieval quantity $q$ at grid point $i$, $d(i,j)$ the distance between the grid points and $l_q$ the quantity-specific correlation length. The assumed a priori uncertainties and correlation lengths for the retrieval quantities are summarized in Tab. 3.

**Table 3.** A priori uncertainties and correlation lengths used in the retrieval.

| Quantity $q$ | $\sigma_q$ | $l_q$ [km] |
|---|---|---|
| $\log_{10}(N_{0,\text{frozen}}^*)$ | 2 | 5 |
| $D_{m,\text{ice}}$ | 300 µm | 5 |
| $\log_{10}(N_{0,\text{liquid}}^*)$ | 2 | 2 |
| $D_{m,\text{liquid}}$ | 500 µm | 2 |
| $\text{arctanh}(\frac{2\cdot\phi}{1.1} - 1.0)$ | 2 | 2 |
| $\log_{10}(m_{\text{liquid cloud}})$ | 1 | 2 |

As baselines for the assessment of the combined retrieval, also a radar-only and a passive only-retrieval are performed. The radar-only retrieval uses the same implementation as the combined retrieval, but only retrieves frozen and liquid hydrometeors.
For the radar-only retrieval, perfect knowledge of the atmospheric humidity profile is assumed but liquid cloud is ignored in the retrieval forward model.

The setup and retrieval quantities of the passive-only retrieval are similar to the combined retrieval, with the only difference being that frozen and liquid hydrometeors are retrieved at reduced resolution. For ice, $N_0^*$ is retrieved at three equally spaced grid points between freezing layer and troposphere, while $D_m$ is retrieved at five. For liquid hydrometeors, the retrieval grids
for $N_0^*$ and $D_m$ are reduced to two equally spaced points between surface and freezing layer. Relative humidity is retrieved at a vertical resolution of 2 km.





## 3 Results

The first part of this section presents results from a numerical experiment that investigates the complementary information content of the active and passive microwave observations. Results of the combined and the baseline retrievals applied to the

reference cloud scenes are presented in the remaining part of this section.

### 3.1 Complementary information content

A fundamental question regarding the benefit of combining two remote sensing observations in a retrieval is to what extent the observations contain non-redundant information. The degree of non-redundancy in the combined observations is what we refer to here as complementary information content. In order to explore this complementary information content in the

radar and radiometer observations, an idealized, homogeneous cloud layer with a thickness of $4$ km located at an altitude of $10$ km in a tropical atmosphere is considered. The cloud is assumed to consist of a single species of frozen hydrometeors represented using the PSD parametrization which is also used in the retrieval and described in Sec. 2.3.2. As particle model, the 8-ColumnAggregate (ID 8) from the ARTS SSDB is used.

      The question that is addressed here is whether the combination of active and passive observations is able to constrain both

the horizontal and the vertical scaling factors of the PSD of the ice particles in the cloud. To investigate this, the $N_0^*$ and $D_m$ parameters of the homogeneous cloud layer are varied and observations of the cloud layer are simulated. Figure 4 displays the simulated passive cloud signal, i.e. the brightness temperature difference between clear sky and cloudy sky simulation, as filled contours for a selection of channels of the MWI and ICI sensors. For given values of $N_0^*$ and $D_m$, the corresponding ice mass density is given by the relation

$$m = \frac{\pi\rho}{4^4} N_0^* D_m^4. \tag{8}$$

In the figure, the cloud signal is displayed in $D_m$-mass density space and thus shows how the measured passive cloud signal varies with the horizontal and vertical scaling parameters of the PSD. Overlaid onto the contours of the passive cloud signal are the isolines of the maximum radar reflectivity returned from the cloud.

      The contours of the measured active and passive cloud signals show the ambiguity of the single-instrument measurements

with respect to the parameters of the PSD: Along these contours the signal does not change, while the cloud composition does. A necessary condition for a combined cloud retrieval to be able to resolve this ambiguity is that the contours of the active and passive signals cross each other. The panels in Fig. 4 thus provide an indication to what extent the information in the radar measurement and the corresponding passive radiometer channel provide complementary information on the parameters of the PSD. Considering the panels corresponding to the MWI channels, the results show that the observations contain com-

plementary information only for very dense clouds consisting of very large particles. In contrast to that, the ICI observations exhibit crossing contours already at lower $m$ and $D_m$ values, indicating that the complementary information content in these observations is higher for less dense clouds consisting of smaller particles.





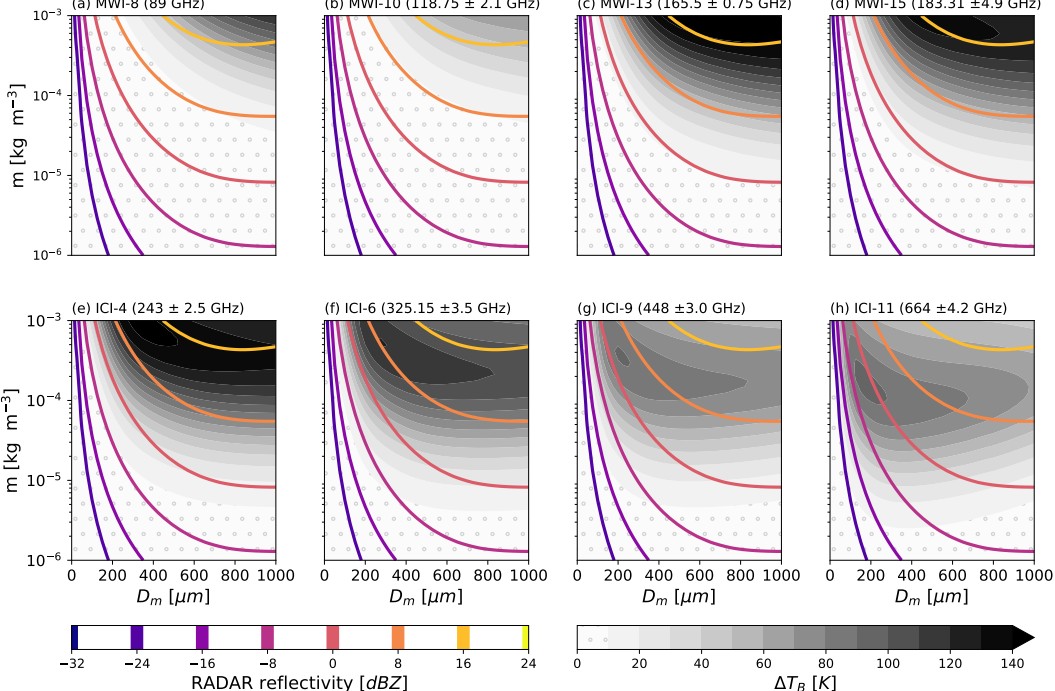

**Figure 4.** Simulated observations of a homogeneous cloud layer with varying mass density $m$ and mass-weighted mean diameter $D_m$. The panels display the maximum radar reflectivity in dBZ overlaid onto the cloud signal measured by selected radiometer channels of the MWI (first row) and ICI radiometers (second row).

## 3.2 Retrieval results

To assess the performance of the combined cloud retrieval, the developed algorithm has been applied to the two designated cloud scenes. The same retrievals have been performed with a radar-only and a passive-only version of the algorithm to serve as baselines for the combined retrieval. Each retrieval was performed multiple times using different ice particle models. The tested particle shapes are listed together with the corresponding mass size relations and ARTS SSDB identifier in Tab. 4. Since the results for both test scenes are qualitatively similar, not all analyses are shown for both scenes. Instead, these are provided as a digital supplement to this article.

The forward-simulated observations that were generated to test the retrievals are shown for the first test scene in Fig. 5. Independent Gaussian noise with standard deviations according to sensor specifications has been added to the simulated observations to account for sensor noise. It is important to note, that the simulated observations which are used to test the retrieval assume different microphysics than what is assumed in the retrieval: The synthetic observations are computed using the six hydrometeor classes from the GEM model, while the retrieval forward model assumes only two classes of hydrometeors.





**Table 4.** Particle model name, ARTS scattering database ID and parameters $\alpha, \beta$ of the mass-size relationships of the particle habits used in the retrieval.

| Name | ID | $\alpha$ | $\beta$ |
|---|---|---|---|
| GemCloudIce | 31 | 440 | 3 |
| GemSnow | 32 | 24.0072 | 2.8571 |
| GemGraupel | 33 | 172.7527 | 2.9646 |
| 8-ColumnAggregate | 8 | 65.4480 | 3 |
| PlateType1 | 9 | 2.4770 | 0.7570 |
| LargePlateAggregate | 20 | 2.2571 | 0.2085 |

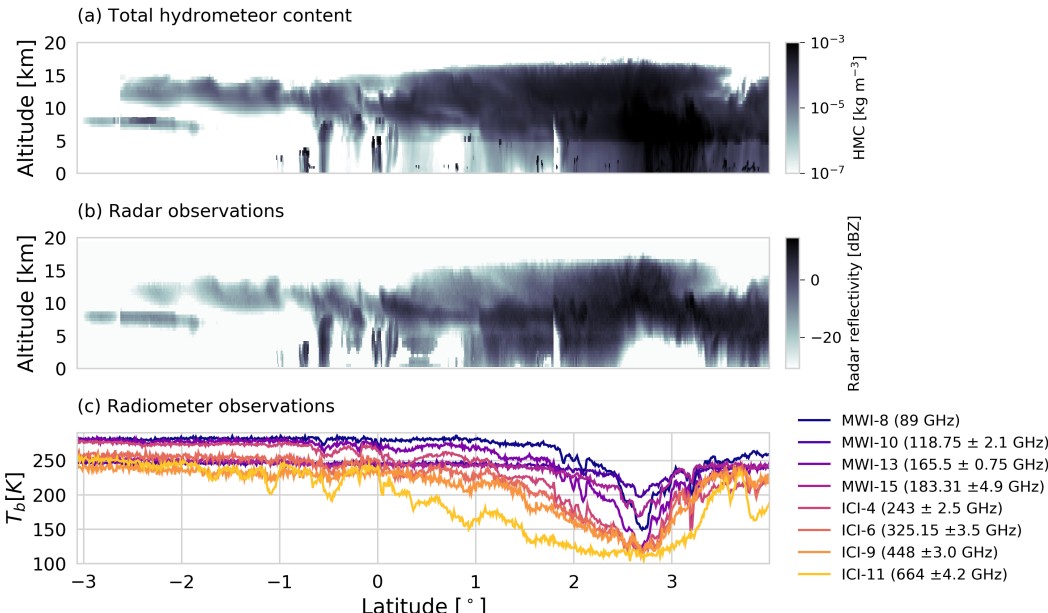

**Figure 5.** Total hydrometeor content (HMC) and simulated observations for the first test scene. Panel (a) displays the total hydrometeor content in the scene, i.e. the sum of the mass densities of all hydrometeor species of the GEM model. Panel (b) shows the simulated radar reflectivities. Panel (c) displays the simulated brightness temperatures for a selection of the channels of the MWI and ICI radiometers.





### 3.2.1 Mass concentrations

To provide an overview over how well the different retrieval methods are able to reproduce the cloud structures in the test scene, the retrieved ice water content (IWC) for the first test scene is shown in Figure 6. IWC is defined here as the sum of the mass densities of all frozen hydrometeor species. This means that the reference IWC is the sum of the four frozen hydrometeor species represented in the GEM model, whereas the retrieved IWC is simply the mass density of the single frozen hydrometeor species assumed in the retrieval. The results shown here were obtained using the LargePlateAggregate as particle model, which was found to be one of the best performing particle models.

Panel (a) of the figure displays the $\chi_y^2$ value (normalized by the dimension of the measurement space) for each profile in the scene. A high value of $\chi_y^2$ indicates that the retrieved state is not consistent with the input observations. The $\chi_y^2$ value for the radar-only retrieval is remarkably low throughout most of the scene. This may indicate that the retrieval is insufficiently regularized, allowing it to fit the noise in the observations. The passive-only and combined retrieval, on the contrary, have a normalized $\chi_y^2$ value around 1 over most of the scene. Since the presented values are normalized, the value 1 corresponds to the expected value of the approximated chi-square distribution of $\chi_y^2$. All three retrievals exhibit a region of elevated $\chi_y^2$ values near the core of the convective system. In particular the high values of the passive-only and combined retrievals indicate that the retrieval was not able to find a good fit to the observations here.

Panel (b) displays the retrieved column-integrated IWC, the ice water path (IWP). The IWP is given in $\mathrm{dB}$ relative to the reference IWP since, owing to the high dynamic range of the reference values, the curves could otherwise not be distinguished. Although all methods reproduce the reference IWP fairly well, the combined retrieval yields the best overall agreement with the reference values. Exceptions are the regions of high $\chi_y^2$ values where the retrieval failed to find a good fit to the observations.

Panel (c) shows the IWC field retrieved using the passive-only retrieval. Despite a certain resemblance in the overall structure between the retrieved and reference IWC field, the results do not reproduce the vertical structure of the cloud very well. It should be noted, however, that the displayed mass-density range extends below the sensitivity limit of the passive-only observations around $10^{-5}\ \mathrm{kg\ m^{-3}}$ (c.f. Fig. 4), which explains the smeared-out appearance of the results to some extent.

The radar-only results, shown in panel (d), reproduce the vertical structure of the cloud well. Nonetheless, when compared to the reference IWC field, certain discrepancies are visible: The radar-only retrieval tends to overestimate the mass density at the bottom of the cloud and underestimate the mass concentrations at the top of the cloud.

The results of the combined retrieval are displayed in panel (e). Although some artifacts are clearly visible in the retrieved IWC field, the retrieval reproduces the vertical structure well. In particular, the combined retrieval succeeds to correct some of the systematic deviations of the radar-only retrieval: The mass density at cloud base is reduced and increased at cloud top.

To make the assessment of the retrieval performance more quantitative, the reference mass concentrations are plotted against the retrieved values in Fig. 7 and 8. The plots show the results for all different retrieval configurations and tested particle models. Markers in the plots are color-coded according to the prevailing hydrometeor type (by mass density) in the reference scene in order to allow assessment of the retrieval performance for the different hydrometeor types of the GEM model.





**Figure 6.** Results of the ice hydrometeor retrieval for the first test scene. Panel (a) displays the value of the $\chi_y^2$ diagnostic normalized by the dimension of the measurement space of the corresponding retrieval. Panel (b) displays retrieved IWP in dB relative to the reference IWP. Panel (c) shows the reference IWC from the model scene. Panel (d), (e) and (f) display the retrieval results for the passive-only, radar-only and combined retrieval, respectively.



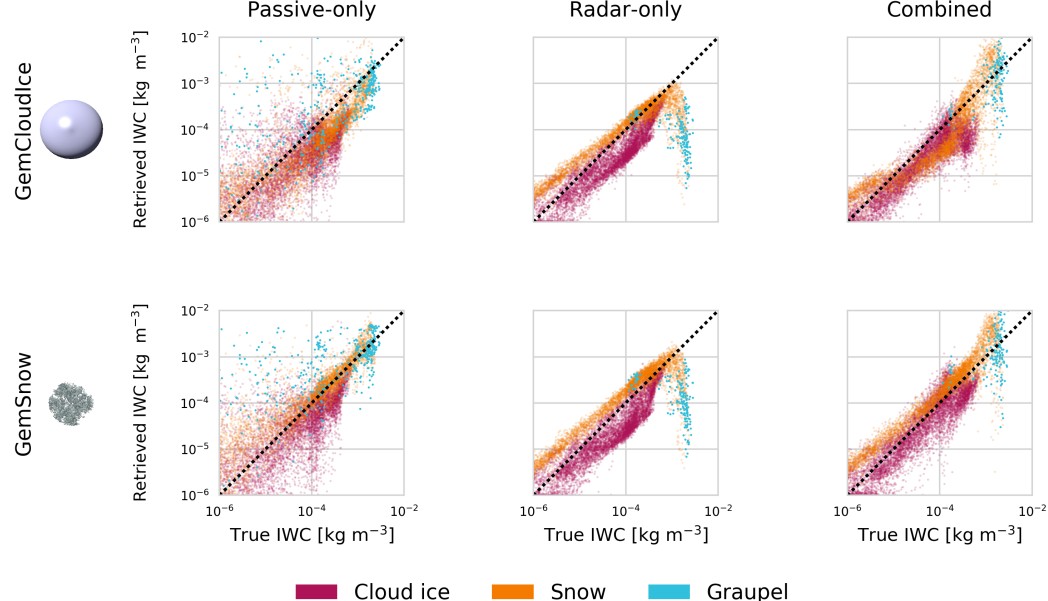

**Figure 7.** Reference IWC plotted against retrieved IWC for the tested retrieval configurations. Each row shows the retrieval results for the particle shape shown in the first panel. The following panels show the retrieval results for the passive only (first column), the radar only (second column) and the combined retrieval (third column). Markers are colored according to the prevailing hydrometeor type at the corresponding grid point in the test scene. Due to their sparsity, markers corresponding to graupel are drawn at twice the size of the other markers.

Not surprisingly, the results from the passive-only retrieval exhibit the strongest deviations from the diagonal. Since the passive channels alone contain only limited information on the vertical distribution of ice in the atmosphere, the retrieval cannot

be expected to yield accurate results at the resolution considered here. Although rather weak, a certain effect of the ice particle model on the retrieval results can be observed. In particular, the GemCloudIce model leads to a systematic underestimation of ice mass densities, which are less pronounced for the other particle models.

The results from the radar-only retrieval are more accurate than the passive-only retrieval, with almost all retrieval results located fairly close to the diagonal. The most distinct feature of the radar-only results, however, is the emergence of two

clusters that extend along the diagonal but are displaced above respectively below it. The color coding of the markers reveals that these clusters correspond to grid points dominated by ice for the cluster below the diagonal and snow for the cluster located above the diagonal. This indicates that the radar-only retrieval systematically underestimates mass densities for cloud ice but overestimates the mass density of snow. The effect is observed for all tested particle shapes and thus likely independent of it. In general, the radar-only results exhibit only very weak dependency on the particle model, making the results for different

particle shapes virtually indistinguishable.





Another feature that stands out in the radar-only results is that the retrieval does not work for graupel. This, however, can be understood by comparing the radar reflectivities shown in Fig. 5 with the cloud structure displayed in Fig. 1. It becomes apparent that graupel in this scene is located where the radar signal is fully attenuated. Since there is no signal to retrieve the mass density from, this explains the bad performance of the radar-only retrieval for these grid points.

Similar to the radar-only retrieval, the results of the combined retrieval are located close to the diagonal. But the clusters observed in the radar-only results are to large extent merged in the combined results. Moreover, except for the results obtained with the GemCloudIce particle shape, the two clusters move in closer towards the diagonal. The combined retrieval thus improves the IWC retrieval for the specific hydrometeor species in the scene.

Nonetheless, the results for the GemCloudIce particle stand out in the results. Even though the systematic deviations ob-
served in the radar-only retrieval are reduced for most particle shapes, for this specific shape they are instead increased. The retrieval error is particularly large for snow, which is strongly underestimated for reference mass concentrations around $10^{-4}$ kg m$^{-3}$.

The results for the second test scene obtained using the LargePlateAggregate particle model are shown in Fig. 9. As mentioned above, the results are qualitatively very similar to those of the first scene. Also here, the final OEM cost, shown in
Panel (a), displays a region of increased cost for the passive-only and combined retrievals. This is again a region of very dense cloud which consists of graupel and snow. Also similar to the first scene, the passive only retrieval does not reproduce the structure of the cloud well. Although the cloud top is placed at the right position, neither the vertical structure of the cloud nor its base are resolved. The radar-only retrieval resolves the vertical structure of the cloud well, but overestimates the ice mass density in the scene. The combined retrieval also resolves the vertical structure of the cloud well and corrects the overestimation
observed in the radar-only results to some extent.

Scatter plots for the retrieval results from the second scene are shown in Fig. 10. Except for the lack of cloud ice in the scene, the results are similar to what has been observed in the first scene: The radar-only retrieval overestimates the mass density of snow in the scene. This effect is corrected by the combined retrieval for most of the tested particle shapes. The exception is the GemCloudIce particle for which the retrieval of snow particle deteriorates quite drastically.

To summarize retrieval performance for all tested retrieval methods and particle shapes, the logarithmic error

$$E_{\log_{10}} = \log_{10}\left(\frac{x_{\text{retrieved}}}{x_{\text{reference}}}\right) \tag{9}$$

for the retrieved IWC and IWP are displayed in Fig. 11. The logarithmic error in the IWC retrieval has been computed only for grid points where either reference or retrieved IWC is larger than $10^{-6}$ kg m$^{-3}$. Considering first the results of the IWC retrieval, shown in Panel (a) and (b), the plots confirm the findings from the analysis above: The combined retrieval generally
yields the smallest retrieval errors. Although the spread of the retrieval errors of the radar-only retrieval is lower in the second scene, the combined retrieval yields smaller systematic errors.

Compared in terms of IWP, however, the results are different. Especially the passive-only retrieval yields much lower errors for the retrieved IWP, making the results comparable if not better than those of the other methods. For the radar-only and





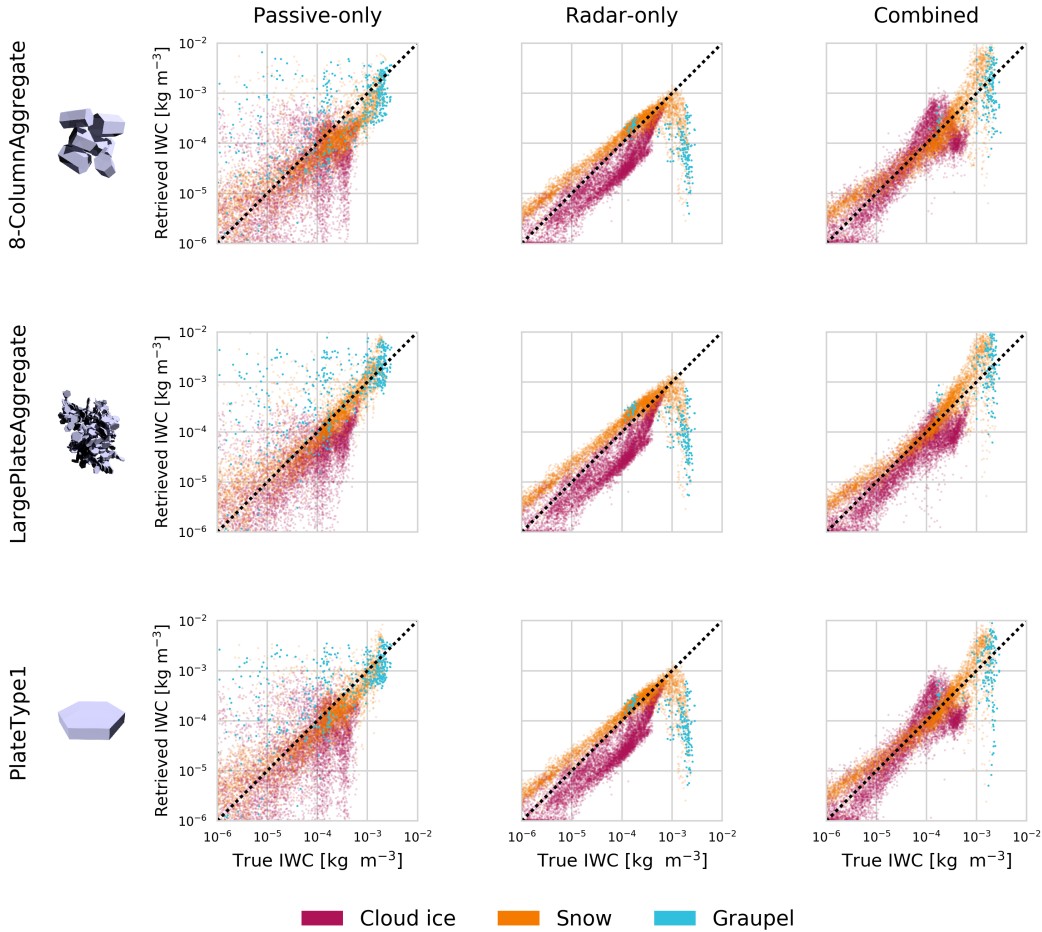

**Figure 8.** Same as Fig. 7 but for the remaining particle shapes.

combined retrievals the precision is generally increased but systematic deviations observed in the IWC persist. This leads,

particularly for the second test scene, to significant systematic errors in the radar-only-retrieved IWP.

In addition to this, the passive-only and the combined retrieval exhibit a strong dependence of the retrieval error on the applied particle model. Especially the GemCloudIce and GemSnow particle models yield large retrieval errors for IWC and IWP. The other three particle models, however, consistently yield smaller retrieval than the GemCloudIce and GemSnow models.

**3.2.2  Particle number densities**

Particle number densities of frozen hydrometeors have been derived from the retrieved $N_0^*$ and $D_m$ parameters by computing the zeroth moment of the corresponding PSD. The resulting particle number density fields are displayed together with the







**Figure 9.** Results of the ice hydrometeor retrieval for the second test scene. Panel (a) displays the value of the $\chi_y^2$ diagnostic normalized by the dimension of the measurement space of the corresponding retrieval. Panel (b) shows retrieved IWP in dB relative to the reference IWP. Panel (c) displays the reference mass concentrations from the model scene. Panel (d), (e) and (f) display the retrieval results for the passive-only, radar-only and combined retrieval, respectively.





**Figure 10.** Scatter plots of the reference and retrieved ice mass densities for the second test scene. The rows show the retrieval results for a given assumed ice particle model. The first column of each row displays a rendering of the particle model. The following rows display the results for the passive-only, the radar-only and the combined retrieval.

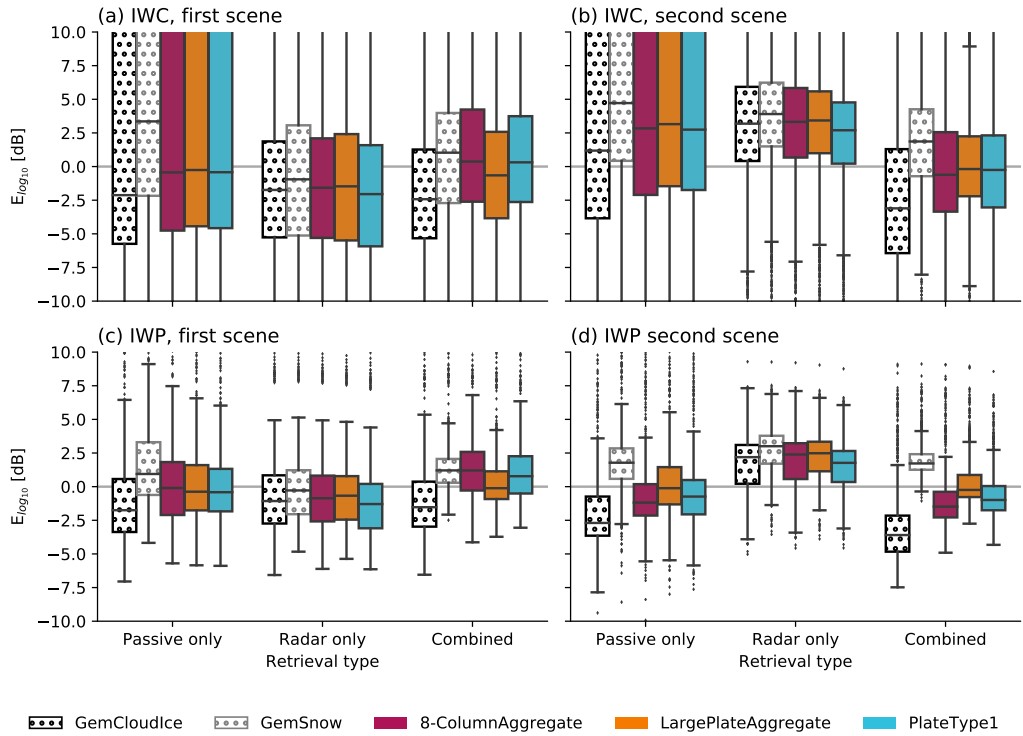

**Figure 11.** Distributions of the logarithmic retrieval error in IWC and IWP for all tested retrieval methods and particle shapes displayed as box plots. Colored boxes display the interquartile range (IQR) while whiskers show full range of all points not considered outliers. Points whose distance to the IQR is larger than 1.5 times the width of the IQR are considered outliers and drawn as markers.

reference field in Fig. 12. To simplify the comparison number densities are displayed only where the corresponding reference or retrieved IWC is larger than $10^{-6}$ kg m$^{-3}$.

Comparing the passive-only and the radar-only retrieval to the reference field shows that both methods have little to no skill in predicting number density concentrations. Although the passive-only retrieval partly captures the gradient between very high concentrations at the top of the cloud and the low concentrations at the bottom, it is not at all resolved in the radar-only retrieval. The combined retrieval, however, manages to reproduce this gradient in some parts of the scene. Although its exact structure is not fully reproduced, this clearly shows sensitivity of the retrievals to particle number concentrations.





The combined retrieval shows the strongest deviations from the reference field between 2 and 3° latitude. Here, the results strongly underestimate the true number concentrations. Comparison with the cloud composition displayed in Panel (a) of Fig. 1 shows that this region contains large amounts of both cloud ice and snow. Since the retrieval uses only a single hydrometeor species to represent ice in the atmosphere it is not able to represent such heterogeneous conditions. Since snow will have the stronger impact on the observations, the retrieval in these regions tends to predict snow rather than ice, which leads to the low

retrieved number densities.

To further investigate this, Fig. 13 displays scatter plots of the reference and retrieved number density concentrations for all three methods and two particle models from the first test scene. Markers in the plot are color coded according to their homogeneity in the reference scene, here defined as the ratio of the maximum mass density of any of the frozen hydrometeor species and total IWC.

These results confirm that the passive-only retrieval possesses certain sensitivity to the particle number density since the cluster at low reference number densities corresponding to snow is placed correctly on the diagonal. The radar-only retrieval does not exhibit any retrieval skill, hardly reproducing any of the variation of the references values. Contrary to this, the combined retrieval moves both clusters towards the diagonal, indicating that it is capable of distinguishing the microphysical properties of cloud ice and snow. Furthermore, the color coding shows that the strongest deviations between retrieved and reference number

densities occur for grid points where the cloud composition is heterogeneous. Even for the combined retrieval, however, the accuracy of a single retrieval value remains fairly low.

The effect of particle shape on the retrieval results is somewhat similar to what has been observed for IWC. For the passive-only and combined retrieval, the GemCloudIce model again yields the worst retrieval results, leading to a general underestimation of the true particle number density. For the radar-only retrieval no noticeable differences are observed between different

particle models. Only the results for the GemCloudIce and LargePlateAggregate particle models are shown here since the results for the other particles are mostly similar to those obtained with the LargePlateAggregate model.

### 3.2.3   Information content

The retrieval results presented above show that the combined observations allow a more accurate retrieval of both mass and particle number density. This confirms the experimental results from Sec. 3.1, that active and passive observations provide

complementary information on the microphysics of ice particles. The information content of the retrievals can be assessed more quantitatively using the averaging kernel matrix. The trace of the AVK, commonly referred to as the number of degrees of freedom for signal (DFS), quantifies the number of independent pieces of information contained in the observations.

The distributions of the degrees of freedom of each retrieved profile in the test scenes are displayed in Fig. 14. Notsurprisingly, the combined observations exhibit the highest information content. Nevertheless, comparison with the DFS values

of the active- and passive-only retrieval shows that the observations contain a certain degree of redundancy leading to a lower combined DFS value than the direct sum of the two.

The grouping into retrieval quantities furthermore reveals that the largest increase in the information content comes from water vapor, which is not retrieved in the radar-only retrieval. Although small, a significant increase in information content



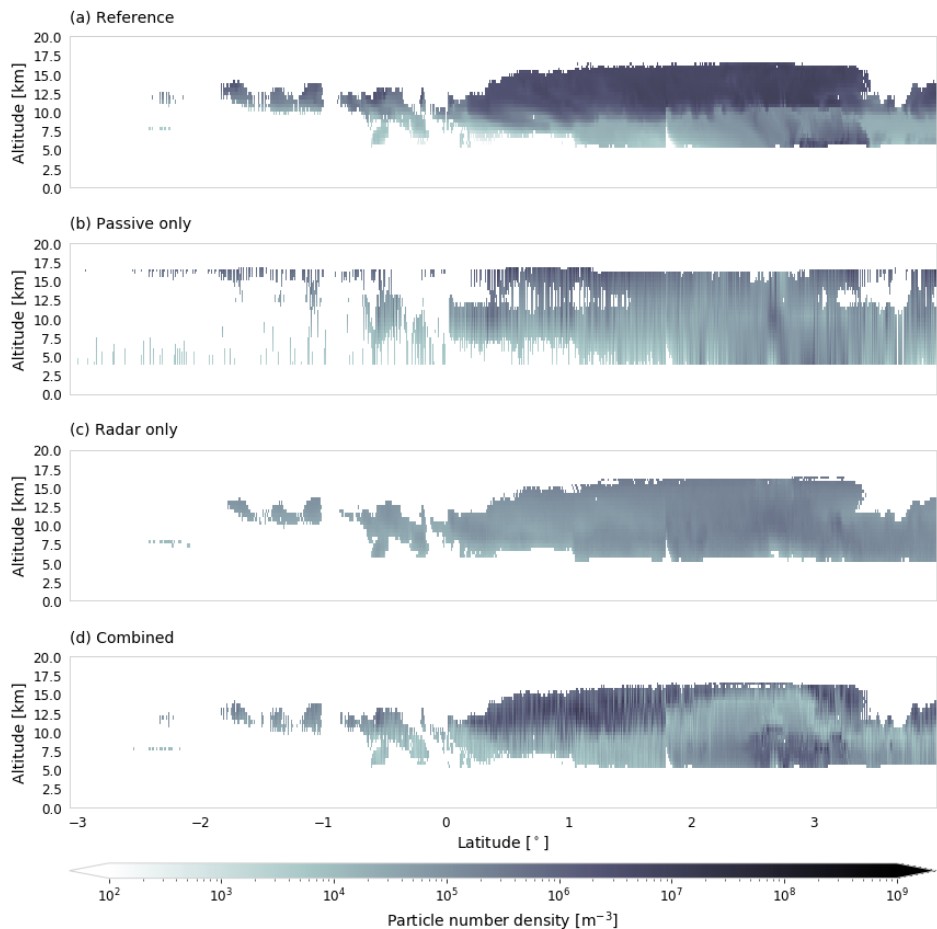

**Figure 12.** Reference and retrieved particle number concentrations of frozen hydrometeors for the first test scene obtained with the Large-PlateAggregate particle model. Panel (a) displays the reference mass concentrations from the model scene. Panel (b), (c) and (d) display the retrieval results for the passive-only, radar-only and combined retrieval. Only values for which the corresponding reference or retrieved IWC was larger than $10^{-6}$ kg m$^{-3}$ are shown here.





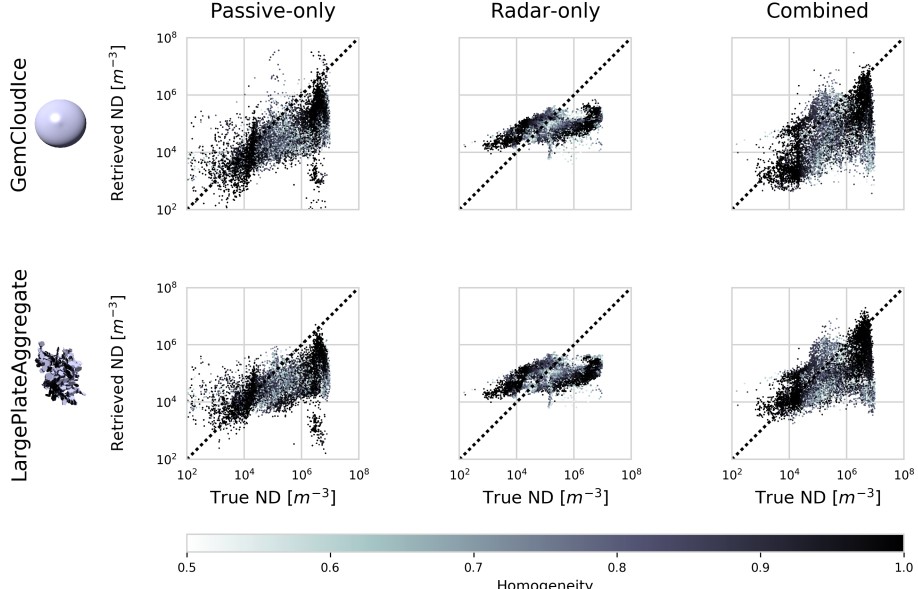

**Figure 13.** Scatter plots of the retrieved particle number densities at grid points with reference mass density larger than $10^{-5}$ kg m$^{-3}$. Rows show the results for the different particle models used in the retrieval while column display the results for the different retrieval methods. The marker color encodes the homogeneity of the corresponding ice mass, which is computed as the ratio of the maximum mass density of any of the frozen hydrometeor species and total IWC.

is observed for both scenes for the $N_0^*$ parameter for ice hydrometeors. Interestingly, this increase is observed even though

the information content in the passive-only observations for $N_0^*$ is close to zero. For the $D_m$ parameter, a small decrease is observed with respect to the radar-only retrieval for both scenes. Since the calculation of the AVK involves the forward model Jacobian, this effect must be related to the non-linearity of the forward model.

### 3.2.4 Impact of assumed ice particle shape

To further investigate the effect of the assumed ice particle shape on the retrieval results, the mass density relations for the tested

particle models are displayed in Panel (a) of Fig. 15. As can be seen from this plot, the GemCloudIce particle clearly stands out due to its large mass. Except for the fact that the GemSnow particle does not reach down to small particle sizes, the remaining particle models have quite similar in mass-size relations. The extreme density of the GemCloudIce particle model for large particle sizes likely explains the bad performance observed in the results presented above. Similarly, the bad performance of the GemSnow model in terms of retrieved IWC and IWP is likely due to it not covering small particle sizes.

Also displayed in Fig. 15 (panel (b) and (c)) are the $\chi_y^2$ values of the combined retrieval obtained for the tested particle models. Since the particle shape has considerable effect on sub-millimeter observations (Ekelund et al., 2019), one could hope that the retrieval results can be used to infer the prevailing ice particle type based on the how well the retrieval can





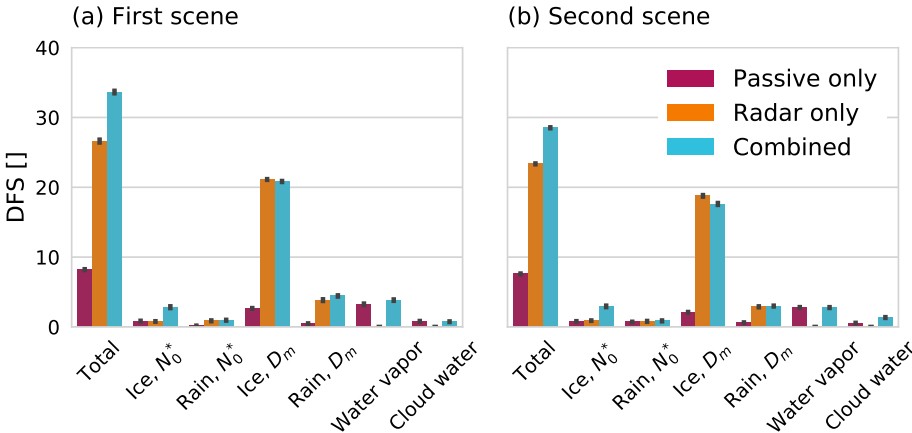

**Figure 14.** Distributions of degrees of freedom of signal displayed as bar plots grouped by retrieval quantity and method. Results for the first test scene are displayed in Panel (a) and for the second test scene in Panel (b). Markers on the top of bars mark the extent of one standard deviation around the mean of each distribution.

fit the observations. Unfortunately, such clear conclusions cannot be drawn from the results. In the first test scene, the best fit is obtained by the GemSnow, GemCloudIce and the LargePlateAggregate particle models, although the GemSnow and

GemCloudIce models quite clearly yield the worst retrieval performance. For the second scene, similar results are observed. Here, the GemSnow particle consistently gives the lowest $\chi^2_y$ value but comparison with Fig. 11 clearly shows that it does not yield the best retrieval performance.

### 3.2.5 Humidity and cloud water

The developed passive and combined retrieval algorithms also retrieve profiles of humidity and liquid cloud mass density. For

relative humidity, both retrievals demonstrate sensitivity but no improvement could be observed in the results of the combined retrieval compared to the passive-only retrieval. Moreover, no suitable retrieval setup was found within the scope of this study which would yield throughout satisfactory performance. Since we do not consider our results representative of what could be achieved with the observational approach, they are not included here.

The liquid cloud retrieval, however, revealed an additional synergy of the radar and passive microwave observations. The

retrieval results are therefore shown in Fig. 16 to serve as a preview for potential additional applications of the combined retrieval approach. Panel (a) of the figure shows the reference and retrieved column-integrated LWC, here referred to as liquid water path (LWP). Although the total LWC is still underestimated, the combined observations clearly improve the LWP retrieval in all regions except those covered by thick clouds.

Panel (b) displays the reference LWC drawn as contours on top of the total hydrometeor content. Panel (c) and (d) show the

retrieved LWC drawn on top of the retrieved IWC for the passive-only and the combined retrieval. These results show clearly that the combined retrieval is able to detect and retrieve liquid clouds even when they overlap with ice clouds. Although some



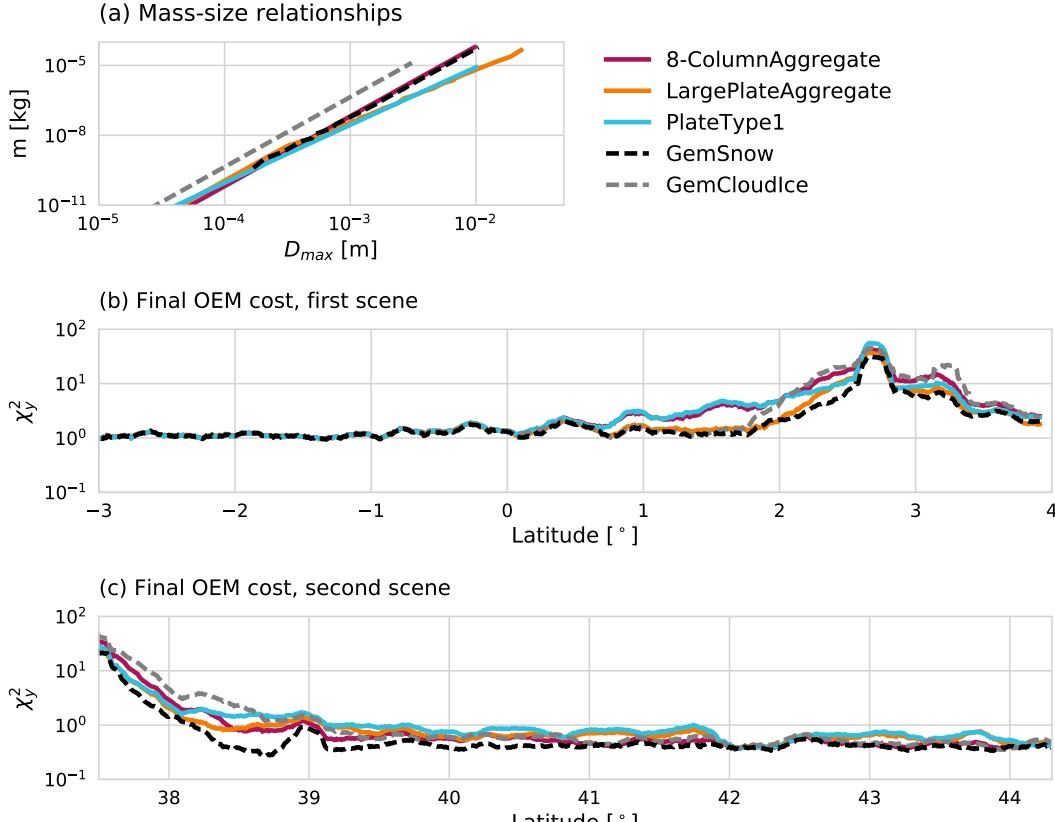

**Figure 15.** Mass-size relations (Panel (a)) and $\chi_y^2$ values for the two test scenes (Panel (b) and (c)). The final cost curves where smoothed using a running average filter of a width of 20 profiles.

sensitivity of the passive-only retrieval to LWC can be observed as well, the retrieval puts the cloud too high in the troposphere and underestimates its LWC. This indicates that the radar reflectivity profile contains useful information for the retrieval to better locate cloud water in the atmospheric column.

**4 Discussion**

The principal aim of this study was to investigate the synergies between radar and passive sub-millimeter observations. To this end, a simplified numerical experiment has been presented, that qualitatively demonstrates the existence of complementary information in the radar and passive microwave observations. Furthermore, a combined retrieval algorithm has been developed to demonstrate the feasibility of the synergistic retrievals and further explore their potential as well as current limitations.



**Figure 16.** Reference and retrieved LWC. Panel (a) shows the reference and retrieved LWP for each profile. Panel (b) displays reference LWC contours drawn on top of the total hydrometeor content. Retrieval results for passive-only and combined retrieval are given in Panel (c) and (d).



## 4.1 Fundamental synergies

The experiment presented in the first part of this study aimed to establish the fundamental synergies of the active and passive microwave observations. It compared the cloud signals observed by a radar, a millimeter-wave radiometer and a sub-millimeter-wave radiometer. The results show that the combined observations can simultaneously constrain the horizontal and vertical scaling of the particle size distribution. However, the complementary information content between the active and passive observations depends on both the properties of the observed cloud and the frequency of the observations. For the lower frequencies considered in this study, i.e. the highest channels of the MWI radiometer, the regions where both observations provide complementary information on the particle size distribution of the cloud are limited to very high mass densities and particle sizes. It should be noted, however, that since the radar simulations neglect multiple scattering, the results are likely less accurate in this region of the cloud-parameter space. As the passive observing frequency increases, the regions of complementary information content extend down to smaller particle sizes and cloud mass density. Especially the highest-frequency channels of the ICI radiometer can therefore be expected to provide additional information on the particle size distribution of ice clouds.

## 4.2 Combined cloud retrieval

In the second part of the study, we have presented results from a combined, variational cloud retrieval applied to synthetic observations from two test scenes from a high-resolution atmosphere model. The results of the combined retrieval were compared to that of a passive- and a radar-only version of the retrieval algorithm. The simulated observations neglected potential errors caused by different or non-overlapping antenna beams as well as inhomogeneity of the atmosphere across the beams. On the other hand, a source of forward model error was included by applying a more complex microphysics scheme in the simulations than the one used in the retrieval. This allows assessing the retrieval error caused by the simplified modeling of cloud microphysics in the retrieval.

### 4.2.1 Retrieval performance

Of the three considered retrieval implementations, the passive-only retrieval clearly performs worst in terms of retrieved IWC. It should be noted, however, that the passive only retrieval presented here has not been fully optimized and should therefore not be taken as representative of the potential performance of the MWI and ICI radiometers for IWC retrievals. To ensure a fair comparison, the retrieval uses almost the same a priori assumptions as the other two retrievals, which in the presented case provide only very limited information on the vertical structure of the cloud. As has been shown also by other studies, the passive observations do provide information on the vertical distribution of ice in the atmospheric column (Wang et al., 2017; Grützun et al., 2018), but the information content is limited to a few degrees of freedom. It is therefore unlikely that the vertical resolution of the passive-only retrieval can be improved drastically without further constraining it a priori, as it is typically done in retrievals that use Monte Carlo integration or neural networks (Pfreundschuh et al., 2018).





With respect to IWP, however, the passive retrieval can perform as well or even better than the radar-only and the combined retrieval. Furthermore, the results in Figure 12 indicate that the passive observations provide some information on the particle number concentrations, which is not the case for the radar observations. This in itself is an interesting result as it shows that even when considered separately, observations from active and passive microwave sensors should be considered complementary to
each other in their information content.

As expected, the radar-only retrieval provides much better IWC retrievals than the passive-only version. However, the results exhibit systematic deviations from the reference values in certain regions of the cloud. The analysis of the retrieval performance shown in Figure 7, 8 and 10 revealed that these are caused by systematic errors in the retrieval of specific hydrometeor species from the GEM model. A likely explanation for this is that the priori assumptions applied in the retrieval do not fit the specific
microphysical properties of the species in the model. This hypothesis is confirmed by the radar-retrieved number density fields shown in Fig. 12 and Fig. 13. While the reference distribution has two modes corresponding to ice and snow, the retrieved values are nearly the same throughout the whole scene. Viewed from an information content perspective, this is plausible since the radar provides only one piece of independent information at each range gate, which is insufficient to determine the two degrees of freedom ($N_0^*$ and $D_m$) of the PSD. The information on the second degree of freedom must therefore come from the
a priori assumptions.

The a priori assumptions which were used in this study were similar but not identical to what is used in the DARDAR retrievals. Also here it should be noted, that the presented results should not be taken to be representative for the DARDAR product. Rather than this, the DARDAR a priori settings were chosen since they represent well established and validated assumptions for ice cloud retrievals and therefore should provide a reasonable starting point for the development of a combined
cloud retrieval. The fact that the a priori assumptions used in the DARDAR retrieval do not agree with the microphysical properties of ice and snow in the GEM model, does not say much about the general validity of these assumptions.

Despite the certain visible artifacts in the retrieved IWC field (Fig. 6), the analysis of the results of the combined retrieval presented in Figs. 7 , 8 and in particular 11 shows that it yields, at least for most of the tested particle models, the best retrieval performance for IWC and IWP. The benefit of the combined observations is even more pronounced in the retrieved number
density fields (Fig. 12). Here, the passive- and radar-only retrieval showed little to no skill in retrieving the particle number concentrations. The combined retrieval, however, was able to reproduce the general structure of the number concentration fields in regions where the cloud composition is homogeneous (Fig. 13). In particular this showed that the combined retrieval is able to distinguish the microphysical properties of ice and snow in the model.

### 4.2.2   Impact of the assumed particle shape

Although the combined retrieval can reduce systematic errors in the retrieved IWC and IWP, its performance can even degrade if an unrealistic particle habit is used, as observed in Fig. 11. In general, the passive-only and the combined retrievals display stronger sensitivity to the assumed particle shape than the radar-only retrieval. This is plausible since the increased sensitivity especially of the sub-millimeter radiometer channels has been highlighted in several studies (Ekelund and Eriksson, 2019; Fox et al., 2019).





Given the increased sensitivity of the passive-only and combined retrieval to the assumed particle shape, it would be desirable to know which of the properties of a particle model are most critical for its representativeness. Five different particle models were tested here: The two most dominant from the GEM model and three additional models taken from the ARTS SSDB. The two GEM particles both showed the worst retrieval performance. For the GemCloudIce model, a likely explanation for its bad performance is its very high density. The GemSnow model has similar density as the 8-ColumnAggregate, but does not

reach down to small particle sizes, possibly explaining why it is unsuitable for the retrieval. Nonetheless, small performance differences are observed also for the other three models, but no clear connection to their mass-size relation can be established. This indicates that also its specific scattering properties are important factors that determine representativeness of a particle model.

Furthermore, it has been briefly investigated whether the goodness of the fit to the observations can provide information on

the suitability of the chosen particle model. In particular, we aimed to address the question whether the combined observations can constrain the dominant particle shape or whether a good fit to the observations can be obtained regardless of the applied particle model. Unfortunately, no evidence of a relation between the $\chi_y^2$ value and the retrieval performance was observed. It thus remains an open question whether and how information on the ice particle shape can be extracted from microwave observations of ice particles.

### 4.2.3   Humidity and cloud water

As an outlook, we have also included results from the liquid cloud retrieval, that clearly shows its capability to retrieve liquid cloud mass densities even within mixed-phase clouds. Although certain sensitivity to cloud water is observed also for the passive-only retrieval, the addition of the radar signal clearly improved the localization of the cloud in the atmosphere. This explains the observed improvement in the retrieved LWP, since at lower altitude a thicker cloud is required to yield the same

passive cloud signal. This shows that combined radar and microwave radiometer observations can also be used for the profiling of warm and supercooled liquid clouds.

Although no satisfactory results were obtained from the water vapor retrieval, the retrieval results still indicate sensitivity of this setup for retrieving atmospheric humidity. The full exploration of the potential of the combined observations for liquid cloud and water vapor is out of the scope of this study and is left to future investigation.

### 4.2.4   Retrieval method

The combined retrieval implementation showed robust performance on fairly distinct and complex cloud scenes. Despite this, both scenes that were considered here contained parts where the OEM minimization did not find a state that results in a good fit to the observations. In contrast to that, the radar-only retrieval did converge well in most regions where the final cost of the combined retrieval remained high. The inability of the retrieval to fit the observations indicates additional information that

is contained in the combined observations but which the retrieval method cannot disentangle. Furthermore, the results exhibit visible profile-to-profile variability as well as some artifacts in the form of high-frequency vertical oscillation. We have tried to counteract these by increasing the vertical spatial correlation but to no avail.





This raises the question of the suitability of the OEM method applied here. The combined retrieval violates the two funda-mental assumptions of the OEM method: The forward model is non-linear and the assumed Gaussian a priori assumptions do not describe reality very well. In addition to that, the current implementation of the retrieval is computationally very expensive. For further development of the combined retrieval concept it may therefore be advisable to revisit the applied retrieval method in search for a potentially more suitable alternative.

### 4.2.5 Limitations

Finally, it is important to consider the limitations of this study. The results presented here are purely based on simulations and restricted to two selected model test scenes. The validity of the presented results thus to some extent depends on how well cloud microphysics are represented in GEM model. While this may affect the specific performance results for the tested retrieval methods, the main findings of this work, namely that the combined retrieval shows greater sensitivity to the microphysical properties of ice hydrometeors than the radar- or passive-only retrievals, should be independent of the realism of the test scenes.

Furthermore, the forward simulations used to generate the synthetic observations do not consider beam filling issues, assume a slightly unrealistic viewing geometry and neglect multiple scattering in the radar simulations. For a realistic assessment of the potential retrieval performance this should certainly be taken into account. Again, it is important to understand the results presented here as a study of the fundamental synergies of active and passive microwave observations rather than an accurate performance assessment of the combined retrieval.

## 5   Conclusions

The main conclusions from the results presented above are:

1. The complementary information in active and passive microwave observations can constrain two degrees of freedom of the PSD of frozen hydrometeors.

2. This reduces systematic retrieval errors for specific hydrometeor species whose properties are not well described by the a priori assumptions.

3. Especially the sub-millimeter channels of the ICI radiometer contribute to the synergistic information content for ice particles.

In addition to this, the combined retrieval also shows improved profiling capabilities for warm and supercooled liquid clouds. The results presented in this study particularly highlight the complementarity of the active and passive observations: Al-though the radar provides observations at high vertical resolution, they contain insufficient information on the microphysical properties of hydrometeors. The passive-only observations, on the contrary, have low vertical resolution, but are more sensitive to cloud microphysics allowing a potentially more accurate IWP retrieval than what can be obtained from the radar alone.





A synergistic retrieval using both types of observations allows combining the high vertical resolution of the radar with the sensitivity to cloud microphysics of the passive observations, which yields more accurate retrievals of IWC, IWP and particle
number densities.

Synergistic retrievals from active and passive microwave observations ideally complement currently available observation systems that combine radar with observations in the visible or infrared. The advantage of combined microwave observations is that they provide sensitivity throughout the whole cloud, where visible and infrared observations would be saturated. Where only information from the radar is available, a retrieval based on optical or infrared observations has to rely on a priori assumptions, which may cause similar systematic errors as what has been observed in this study. In addition to this, our results underline the benefits of ICI's sub-millimeter channels, which significantly improve the sensitivity of the passive observations to smaller particle sizes and mass densities and thus narrow the sensitivity gap between the observing frequencies of traditional microwave imagers and observations in the infrared and visible domain.

The upcoming launch of the ICI and MWI radiometers thus provides a great opportunity for a potential synergistic cloud-
radar missions. Such a mission would have a unique scientific value for the study of frozen hydrometeors because of its ability to better determine the microphysical properties of hydrometeors even inside of thick clouds. Since such information is currently simply not available at a global scale, such a mission would be valuable not only in itself but also for other earth observation systems by establishing more reliable a priori assumptions on cloud microphysics.

The results presented in this study not only show the potential of the combined retrieval approach but also demonstrate its
feasibility. Although further work will be required to fully understand the effect of particle shape and PSD, the concept is mature enough to be applied to real observations. Since airborne demonstrators of sub-millimeter radiometers are available already today, the combined retrievals could be applied in future field campaigns to study ice cloud microphysics.

Overall, the combined active and passive microwave retrievals are a promising concept that deserves further exploration. Regardless whether airborne or spaceborne, combined active and passive microwave observations have great potential to improve
the understanding of the microphysical properties of ice hydrometeors.

*Code availability.*   All code used to produce the results in this study is publicly available online. (Simon Pfreundschuh, 2019).

*Data availability.*   Data to reproduce the simulations leading to the presented results will be made available on request.

*Author contributions.*   Simon Pfreundschuh has implemented the retrieval, performed the data analysis and written the manuscript. Patrick Eriksson and Richard Larsson have added code to the ARTS radiative transfer model that was required to perform the presented calculations.
Stefan A. Buehler, Patrick Eriksson, Manfred Brath and Simon Pfreundschuh have collaborated on the study that lead to the results presented here. David Duncan and Robin Ekelund have contributed to the conceptualization of the study through comments and advice.



*Competing interests.* No competing interests are present.

*Acknowledgements.* The combined and radar-only were developed as part of the ESA-funded study "Scientific Concept Study for Wide-Swath High-Resolution Cloud Profiling" (Contract number: 4000119850/17/NL/LvH). The authors would like to thank study manager Tobias
Wehr for his valuable input and guidance.

Furthermore, the authors would like to acknowledge the work of Zhipeng Qu, Howard Barker, and Jason Cole from Environment and Climate Change Canada who produced the model scenes that were used to test the retrieval.

The work of SP, PE and RE on this study was financially supported by the Swedish National Space Agency (SNSA) under grants 150/14 and 166/18.

SB was supported by the Deutsche Forschungsgemeinschaft (DFG, German Research Foundation) under Germany's Excellence Strategy — EXC 2037 'Climate, Climatic Change, and Society' — Project Number: 390683824, contributing to the Center for Earth System Research and Sustainability (CEN) of Universität Hamburg.

The computations for this study were performed using several freely available programming languages and software packages, most prominently the Python language (The Python Language Foundation, 2018), the IPython computing environment (Perez and Granger, 2007),
the numpy package for numerical computing (van der Walt et al., 2011) and matplotlib for generating figures (Hunter, 2007).



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
