# Peer review of "Synergistic radar and radiometer retrievals of ice hydrometeors"

_Atmospheric Measurement Techniques, 2019_

## Referee Comment (RC1) · Anonymous Referee #1 · 19 Nov 2019

Summary of manuscript:

This is a conceptual study evaluating the ability of passive microwave and sum-mm measurements to complement radar reflectivity profiles to synergistically constrain profiles of ice particle size and concentration. An optimal estimation framework is applied to the problem in radiometer-only, radar-only, and combined radar-radiometer mode. The primary finding is that the measurements do indeed complement each other and the combined retrievals are less biased and have smaller errors than the radiometer-only or radar-only retrievals with respect to several parameters (mean ice particle size, number concentration, and cloud liquid water content). Sensitivity tests to the particle scattering model assumed in the retrieval forward model are also described in detail.

Summary of Review:

[Figure]

Overall, this study is quite thorough in several aspects: the description of the forward model assumptions are clearly stated, sufficient information is provided about the parameterization of the state vector to contextualize the optimal estimation results, and the retrieval of many parameters is analyzed in detail. The largest issue I find with respect to the methodology is the characterization of the measurement and state vector error covariance matrices. I suspect that this has lead to some ambiguous or sub-optimal (for the setup used by the authors) results, but I don't believe the general conclusions would be substantially affected by a more realistic treatment. Therefore, my disposition is for minor revisions (see specific comments).

General Comments:

1. As noted in Section 4.2.4, the a priori assumptions do not describe reality very well. In particular, I suspect that the information content of Dm and N0* is highly dependent on the a priori assumptions of these two variables in the retrieval framework. Especially with a radar measurement, since Z is sensitive to both parameters over a wide range of the parameter space, the relative sensitivity and therefore information content will almost entirely depend on the relative constraints on these parameters imposed by Xa and Sa. As such it is imperative to accurately characterize these. I understand the choice to use the DARDAR constraints, but it's clear from the cross-section plots that the model ice particle concentrations vary over a much wider range than the roughly 2 orders of magnitude that Eq. 4 provides over a 220-272 K temperature range. So, when the retrieval results are compared to model "reality", it seems that a lot of N0* variability is folded into Dm and this is especially evident in Figures 13 and 14. My overall concern is that it is difficult to interpret some of the results when the model fields and the a priori assumptions differ so strongly.

2. Forward model error is introduced when the different species present in the model microphysics are combined into one species and when different scattering models are used to represent the ice particles. That this is not represented in Se could lead to over-fitting and poor convergence (I suspect this is part of the reason why the normal-

ized cost is much higher for the radiometer-including retrievals). It should be relatively easy to quantify this error by re-running the simulations with the retrieval assumptions (combining ice species, different scattering models), and I suspect that this error term would dominate the instrument noise term for many channels.

Specific Comments:

1. Lines 85-88: I recommend the use of geographical spatial references (i.e., north/south rather than left/right)

2. Line 98 (also 176,252,449): Instead of vertical/horizontal (which are dependent one the convention used for plotting), I recommend the use of concentration/size to characterize the dimensions of the particle size distribution.

3. Line 100: A few more details on the Milbrant and Yau microphysics sheme that are relevant to this study would be helpful here. For example: What is the assumed shape (functional form) of the particle size distribution, and what are the prognostic variables (e.g., number concentration, mixing ratio)?

4. Line 135: Does the ARTS radar solver also provide analytic Jacobians?

5. Line 187: "particles" should be "particle"

6. Line 198: Is Dm also only retrieved at these 10 points, or just N0* (and Dm retrieved in each radar range gate as in Grecu et al. 2016)?

7. Line 256: Actually, this is only one example of how the radar and radiometer measurements can be complementary. Even if the lines were parallel (and thus no information distinguishing size from concentration could be obtained), the radar still locates the cloud and describes its vertical structure. One can imagine a cloud of the same ice water path and particle size at two different heights having different brightness temperatures due to changes in the water vapor absorption above the cloud – having the radar information would provide increased information content about the ice water path in this case than the radiometer measurement alone.
8. Table 4: Why are the values for GemSnow and GemGraupel different than in Table 1?

9. Figures 7 and 8: I'm not sure why these are separate figures – it seems like all panels could fit on one page.

10. Figure 10 is missing from the manuscript.

11. Line 374: recommend using "represent" instead of "predict"

12. Line 382: should be "reference" instead of "references"

13. Line 414: How are the truncated PSDs (using GemSnow) represented in the forward simulations? Is total ice water content conserved? If so, how is it spread among the valid particle sizes – equally, or is the truncated mass allocated to the smallest size bin?

14. Figure 16: The figure labels/captions aren't clear if they refer to total liquid water content/path or just the cloud liquid water/path.

15. Line 518: It's interesting that the Plate Aggregate provides the most accurate retrieval results, even though it isn't similar to the models used in the synthetic measurement simulations. Does the decreasing density with size better replicate the combination of high-density GemCloudIce (which tends to be present in high concentrations at small sizes) and lower-density GemSnow (which tends to be dominant at larger sizes)?

---

## Referee Comment (RC2) · Anonymous Referee #2 · 2 Dec 2019

With the launch of the second generation of European meteorological operational satellites (MetOp) approaching the interest in exploiting the measurements by the novel Ice Cloud Imager (ICI) is growing. While several studies already investigated the potential of these submillimeter measurements in deriving information on frozen hydrometeors alone or in conjunction with the Microwave Imager onboard of the same satellite, this manuscript addresses the synergy of ICI/MWI and a hypothetical 94 GHz cloud radar as a potential further development of space instrumentation. An optimal estimation framework for joint retrievals is developed and a comprehensive assessment of the information gain from individual (passive or radar) and combined retrievals is performed. Not surprisingly a significant benefit of the combined approach is demonstrated on the basis of two scenes simulated by a cloud resolving model. However, the effects depend

on the parameters of interest and assumptions on particle type.

The study original and timely. It is well designed and assumptions and limitations are clearly described though some aspects on the retrieval need to be clarified (see below). In general, the paper is well written but rather lengthy and I would appreciate if the authors would follow my suggestions to shorten the paper.

MAJOR POINTS

1) A major aspect of the study concerns the representation of the particle size distribution which is retrieved by two free parameters (different from the 2 moments of the atmospheric model GEM used to provide the test scenes) and the assumptions of the particle type. The difficulty of connecting atmospheric model output to single scattering properties (which is one of the fundamental assumptions) could be better explained. The motivation why the authors choose their approach and why they test certain settings need to be discussed in the beginning. Couldn't Tab. 1 and 4 be combined and better explained which is used for which purpose? Why is cloud ice the same and GEMsnow and GEMGraupel different in both?

2) Although different parameterizations of the hydrometeor types are used to study their effects, vertical changes (development of sedimenting particles) is are not considered. Similar polarization effects are not mentioned in the discussion on particle shape. Otherwise the paper nicely discusses the different aspects but in the end I am missing a clear message on the outcome of the test (choice of particle types). What is recommended for the future?

3) Not only the two moments of the ice PSD but further variables are retrieved and their information content is nicely shown in Fig. 14. I am surprised that the he information on moisture is so low although information along three water vapor lines is provided? This should at least in the upper atmosphere provide information? Is it due to the choice of relative humidity which mainly depends on temperature? I am also skeptical about the results of Fig. 16. Basically, there should be no liquid for temperatures colder than –

40 deg C (freezing) but it even reference LWC goes up to 13 km? I would not support the statement on L568 – where is the evidence? Similar L527 – liquid water estimation within mixed phase clouds is extremely difficult and if ICI and radar could really do that together this would be worth a separate paper. To better understand the information content, I suggest to plot the profiles of cumulative degrees of freedom for the different retrievals as this could help interprete where and how the synergy works.

4) The manuscript is rather lengthy making it difficult for the reader to extract the major points. I strongly suggest a) to move part of the analysis into an appendix (, b) remove double statement (see minor comments, also the LWC plot) and c) to remove figure caption like information (for example L92 or "filled contours") from the text. The text must make sense without looking at the figure. Figure only support the statements made in the text. Lengthy descriptions such as "The plot shows.." need to be avoided.

MINOR COMMENTS:

L39: Is sensitivity really the right word? Range resolution is the main advantage – signal-to noise range depends on distance and hydrometeor distribution,

L48: MWI will also cover new spectral channels, e.g. 118 GHz

L62: "high-resolution" is always relative for a model. I would recommend avoiding this term and use Cloud resolving Model (CRM).

L68: After you mention GPM (with scanning radar) it might be good to say that you are only looking at a nadir pointing radar (curtain). The swath center cam e bit as a surprise.

L70: There has been quite some literature about combining active and passive MW using a Bayesian framework which should be acknowledged, e.g. Grecu, M., & Olson, W. S. (2006), Johnson et al. (2012) , Munchak, S. J., & Kummerow

L84: Test scenes have a grid resolution of 1 km horizontally. As this is not the true model resolution I would have recommended to coarse sample the model data (maybe

every 5th data point) and include more diverse profiles instead. This might be especially interesting for the scatter plots.

L91: Motivation lacking: "To perform RT simulations for each GEM profile the PSD needs to be diagnosed from the prognostic GEM variables, i.e. N and m.."

L92: "prognoses" means forward in time - you mean diagnosed, calculated, determined...

L98: I find the term "horizontal and vertical scaling" strange – why not saying PSD shape is similar but scaling in respect to diameter and number density. At least define the term clearly the first time that you use it or define a short for it.

L103: model test – be careful also at other instances that "model" can mean too many things. Here I would say GEM test scenes.

L119: Need to clearly say that polarization effects are neglected though these can be several Kelvin, e.g. Xie et al., 2015. You ignore this effect but even consider noise reduction.

L157-159: needs to be better motived, references?

L172: I doubt that the model has constant vertical resolution. It will be better close to the surface and worse aloft. This should be mentioned than GEM is introduced.

L 174: for all hydrometeor species of the model? It would be helpful to first introduce all retrieval quantities – I was missing a motivation for the paragraph around L195. How do you define the freezing layer (and later the troposphere)? How do they vary in both test scenes? The model also likely has supercooled liquid water above the freezing layer – how is this treated?

L 198: Vertical resolution of retrieval grid: Why 4 points? The freezing layer must be very different for both cased. Maybe a sketch would be helpful as later on lines 230 the different vertical resolutions for other variables is discussed?

L281: How do I know that Large Plate is the best performing model? Which parameter, plot, table does show that?

L283-L307: Can be shortened significantly

L332: There can I see that? Give figure?

L325: The two paragraph here give similar information -> streamline

L333-344: I would put this to the appendix

L444: Here it needs to be made clearer how this goes beyond what GPM is doing.

L495: "does not say much about the general validity of the assumption". Here you should dig in a bit more. What is the role of a priori and covariances?

L560: Rethink the bullet structure. 2. Is not an independent result. For each result refer to the part of the manuscript where you can clearly see that. Especially result 3 should be detailed – how do ICI channel advance the currently available data?

Fig. 3: Is it really worth having the slightly different size distribution shapes for frozen und liquid? Isn't there a stronger difference between different frozen hydrometeors?

Fig. 4 and also in text: "cloud signal" say that this is dTB.

Fig. 5: Can you add freezing layer height?

Fig. 6: It would be nice to see the absolute values of IWP somewhere. Maybe you could add another time series with IWP as the sum of the different components such that the reader can see where the different categories (cloud, graupel, snow and hail) contribute most?

Fig. 7: I think it is retrieved vs. truth. The word following is not really exact. Why not put 7 and 8 together?

Fig. 9: Could go to the appendix

Fig. 10 I only see the caption???

Tab. 1. Assumed particle model information for each hydrometeor class given by GEM model. In fact it could be good to combine it

Tab.3 : I would recommend to add a first column with a spelled out name

Grecu, M., & Olson, W. S. (2006). Bayesian estimation of precipitation from satellite passive microwave observations using combined radar–radiometer retrievals. Journal of applied meteorology and climatology, 45(3), 416-433.

Johnson, B. T., Petty, G. W., & Skofronick-Jackson, G. (2012). Microwave properties of ice-phase hydrometeors for radar and radiometers: Sensitivity to model assumptions. Journal of Applied Meteorology and Climatology, 51(12), 2152-2171.

Munchak, S. J., & Kummerow, C. D. (2011). A modular optimal estimation method for combined radar–radiometer precipitation profiling. Journal of Applied Meteorology and Climatology, 50(2), 433-448.

Xie, X, S. Crewell, U. Löhnert, C. Simmer, J. Miao, Polarization Signatures and Brightness Temperatures Caused by Horizontally-Oriented Snow Particles at Microwave Bands: Effects of Atmospheric Absorption, J. Geophys. Res., doi:10.1002/2015JD023158.

GRAMMAR, TYPOS AND REFORMULATIONS

L59: "..constrain the microphysics.." better information on microphysical parameters

L82: I was expecting to see horizontal maps in Fig.2 – mention vertical profiles

L87: over the North Atlantic region

L88: not a single cloud -> cloud system

L106: cite Table2

L108: radar with similar characteristics as the CloudSat

L118: the reader does not know yet that these are double-sideband

L123: Higher freq channels were mentioned at the beginning of the section – only needed once.

L207: "..chosen as a function of temperature t "

L210: cloud water is also a liquid hydrometeor type

L213: permitted region?

L233: numerical experiment -> idealized experiment. Everything you do is numerical

L250: density rho not defined. The following paragraph until L262 can be shortened significantly.

L265: "..ice particle models for the retrieved frozen hydrometeor"

L358: retrieval error

L463: "On the other hand" only once you had "on the one hand"

L544: "..but no avail..?"

Fig 1 caption: explain m as mass density, „

––––––––––––––––––––––––––––––

---

## Referee Comment (RC3) · Anonymous Referee #3 · 4 Dec 2019

The paper presents a methodology to assess the benefit of radar-radiometer synergies when retrieving ice particles. The topic is extremely timely (given the upcoming launch of ICI) and relevant for the cloud community. The paper is generally clear in its scope though it is indeed too long and not concise as it could be. The style must be substantially improved, the number of figures reduced, most of the OE description has been now reported in numerous papers (maybe include them in an Appendix).

There are however several major points that must be clarified. I have picked here some that must be addressed.

1) The paper is presented as an application for ICI in combination with a Cloudsat like configuration but it is not clear to me what geometry of observations the authors are thinking about. They state "As mentioned above, the same incidence angle as for the

passive radiometers is assumed also for the radar. In practice, this could be achieved by remapping the radar observations to the lines of sights of the passive beam". Are they thinking about a scanning W-band radar? or at a off-nadir pointing radar? If the former is true then they should discuss what is a realistic technological solution (and what are the consequences in terms of sensitivity) and the authors should refer to state of the art scanning W-band radar concepts (there is none at the moment!); if the latter is true they should discuss what are the consequences of such a selection (e.g. for ground clutter) and they need to convince me that what we could gain from such a configuration compensate from the loss of information introduced by pointing in such a slanted direction. There should be a certain degree of "realism" in what we are trying to simulate, especially if this was part of an ESA study. 2) "the beams of all three sensors are modeled as perfectly coincident pencil beams". Again this is quite an assumption. Non uniform beam filling will play a key factor. This is one of the many simplifications (no polarization, no multiple scattering,1D, ...) that needs to be clearly listed at the beginning of Sect.2.2.1 (some appear only at page 27). For this reason I would actually pitch more towards an airborne configuration where these simplification indeed can be realistically assumed or of a radar with a radiometric mode (where you can actually match footprints). Otherwise the (not massive) gain of having a radar-radiometer combination that you show later on can be completely washed out by the errors introduced to these assumptions. I imagine that you may also have airborne data where to test how realistic your forward model is. 3) Fig2: these PSDs look very weird to me. Why do they have the plateau at small sizes? y-axis units are obviously wrong unless you are renormalizing by some mass (but it is not explained). 4) Fig3: sorry I do not follow what is this (what is the y-axis?), and why this plot is meaningful. 5) Eq.6 clearly with values lower than 230 K it does not make any sense (negative RH, or large than 1.1???) 6) Line 210; this means that the vertical resolution changes with the surface temperature, really weird choice. 7) fIG4 : not clear to me why the scattering depression is not increasing at higher frequencies. I would expect that the optical thickness would drastically increase increasing frequency. Is this due to very large asymmetry

parameters then? But this is not what I do see in Fig.5 (though Fig4 is of course a very idealized case) If this is the case then results will be very dependent on particle habits (which may introduce additional uncertainties in the retrieval). 8) Line 275: not clear what you mean, in Tab.4 there are 6. 9) "extends below the sensitivity limit of the passive-only observations around $10-5$ kg m$-3$" : very sloppy sentence. Passive microwave radiometer are sensitive to integrated contents! 10) Fig 6d: this retrieval looks really weird. Where are all the stripes coming from? Certainly this does not look like a cloud, or? What kind of constraint have you imposed on the cloud top? 11) "In general, the radar-only results exhibit only very weak dependency on the particle model, making the results for different particle shapes virtually indistinguishable." Again another dangerous sentence. We know (unfortunately) that this is not true (otherwise our ice problems would be sorted). Here my guess is that you have not properly explored the backscattering variability (particularly looking at the different degree of riming). It is not clear to me whether there is enough variability in your ARTS database, I guess you are more focused at ice particles (including aggregates) but you are not considering really rimed particles. Regions where graupel is present should we avoided from the discussion of the radar-only retrieval for the simple reason that in those regions attenuation correction and multiple scattering effects make the problem very tricky. I guess that the radiometer as well is in serious trouble when entering those areas. Again I would not start tackling regions the observation system is not tailored for. 12) Fig.10 is missing!!! 13) "Since the calculation of the AVK involves the forward model Jacobian, this effect must be related to the non-linearity of the forward model" well I would avoid such very speculative statements. 14) You need to be very careful how you present the results in Fig.14. The conclusions that I can draw is the following: a CloudSat like radar is providing much more information than the ICI+MWI radiometers when characterizing ice particles (really the radiometer is providing some additional water vapour information). As a result we should invest in the former and not the latter. While I may agree with the previous statement and strongly support a CloudSat-like radar on an operational mission my feeling is that you are pitching your radiometer system at the wrong kind

of scenes (I already see an improvement going from the first to the second scene). I would have selected completely different scenes (including high latitude clouds with mixed phase). It is to me an overkill to try to retrieve D_M of rain for these scenes from your PMW radiometer suite of sensor. If you have any skill in warm rain you should properly prove it. 15) LWP and Fig.16. I have a serious problem here. The cloud I see on the right is a liquid cloud. So how it is possible that your radiometer is doing so badly in the LWP retrieval and why the combined is so much better? I guess this must go back to understanding surface emissivity and integrated water vapour (maybe some comments there should be made to explain what kind of surface/IWP we are dealing with). You have not included radar path integrated attenuation in your retrieval (like is typically done in radar retrievals) but this could of course help in this case. 16) I do not think that for OE to work The forward model must be linear as stated at line 544. 17) Sect.4 and 5: a lot of waffling here (e.g. the three bullet conclusion, you need to be much more quantitative and linked to what you have proved; the three statements are something I could have formulated on my own without making any simulation). Again the conclusions must be related to the cloud regime you are considering (and cannot be valid for all!)

Minor comment: I would avoid the use of "ice mass density" and use "ice water content" Table 2: it would be good to see footprints as well Line 130: dBZ are the wrong units for a std of a reflectivity! Line 180: "The remaining shape of each PSD is described by the shape parameters alpha and beta, not to be confused with the parameters of the mass-size relationship shown in Tab. 1."; very confusing. Why are you using the same letters???? Line 193:wrong units Line 199: English Line 35 page 2 (not really limited, this is a wide range!!) Line 54 page 2. maybe it is worth mentioning all the heritage coming from radar-radiometer retrievals with W-band (Ka and Ku-band) radars with PMW radiometers. Line 229: "troposphere" is too generic Line 250: rho is not defined Line 4: 272.5???? Fig 4 caption: you need to include how thick is the layer.

---

## Author Comment (AC1) · 10 Feb 2020

**Response to reviewer 1**

We thank the referee for the time he/she has put on reading our manuscript and providing feedback. Based on the combined comments of the referees, we have decided to implement these general changes:

- We will switch to an airborne measurement set-up and the introduction section will be modified accordingly

- Text in the result section will be shortened significantly

- Redundant results for scene 2 will be placed in an appendix

- The selection of tested retrieval habits will be revised/changed

Below we respond to the main questions raised by the referee, and outline how we will revise the manuscript.

**1 General comments**

**Reviewer comment 1**

As noted in Section 4.2.4, the a priori assumptions do not describe reality very well. In particular, I suspect that the information content of Dm and N0* is highly dependent on the a priori assumptions of these two variables in the retrieval framework. Especially with a radar measurement, since Z is sensitive to both parameters over a wide range of the parameter space, the relative sensitivity and therefore information content will almost entirely depend on the relative constraints on these parameters imposed by Xa and Sa. As such it is imperative to accurately characterize these. I understand the choice to use the DARDAR constraints, but it's clear from the cross-section plots that the model ice particle concentrations vary over a much wider range than the roughly 2 orders of magnitude that Eq. 4 provides over a 220-272 K temperature range. So, when the retrieval results are compared to model "reality", it seems that a lot of N0* variability is folded into Dm and this is especially evident in Figures 13 and 14. My overall concern is that it is difficult to interpret some of the results when the model fields and the a priori assumptions differ so strongly.

**Author response:**

To avoid potential misunderstanding we would like to point out that the variation of the a priori mean with temperature, which is given by Eq. 4, does not limit the retrieved values of $N_0^*$ to this range. How much $N_0^*$ is allowed to vary around the a priori mean is determined by the covariance matrix. Since the standard deviation for $\log_{10}(N_0^*)$ at each grid point was set to 2 (c.f. Tab. 3), $N_0^*$ is free to vary over several orders of magnitude in addition to the variation of the a priori profile.

Furthermore, the sentence in Section 4.2.4 was badly formulated and did not really express what we wanted to express there. The a priori assumptions are not generally bad for the model (after all the averaged results for the first scene are good). Rather, they are insufficient to accurately describe the (co-)variability of $D_m$ and $N_0^*$.

Nonetheless, the point raised by the reviewer certainly remains valid: In absolute terms, the interpretation of the retrieval results is dependent on the a priori assumptions. We argue here, however, that by applying equivalent a priori assumptions in all retrievals, we can still derive conclusions on the benefits of the combined retrieval approach based on a relative interpretation of the retrieval results. Especially because our results indicate that the combined retrieval has to rely less on a priori assumptions than the radar-only retrieval, this can be an important advantage of the combined retrieval since if $D_m$ and $N_0^*$ could be constrained reliably a priori we would not have the uncertainties in the observational record of ice hydrometeors that we have today.

To address the issues raised by the reviewer we propose to make the following changes in the manuscript:

- To extend the discussion of the role of the a priori (around L. 491) and its impact on the results.

- To add a paragraph to the discussion of the limitations of the study (around L. 549) which clearly states that the retrieval results should not be interpreted in absolute terms

- To rephrase the sentence in Sect. 4.2.4 (L. 545) to stress that is refers to the Gaussian nature of the a priori rather then the a priori itself.

**Reviewer comment 2**

Forward model error is introduced when the different species present in the model microphysics are combined into one species and when different scattering models are used to represent the ice particles. That this is not represented in Se could lead to over-fitting and poor convergence (I suspect this is part of the reason why the normalized cost is much higher for the radiometer-including retrievals). It should be relatively easy to quantify this error by re-running the simulations with the retrieval assumptions(combining ice species, different scattering models), and I suspect that this error term would dominate the instrument noise term for many channels.

**Author response**

It is certainly true that the simplified forward model used in the retrieval introduces a forward modeling error and that it will likely dominate the sensor noise. However, we do not agree with the reviewer that this error was easy to quantify. First of all, the error will not be Gaussian and will depend on the cloud composition and the assumed particle shape, so that a more sophisticated error model would be required to describe the error accurately. Fitting such a model to the test scenes would likely yield overly optimistic results as this would mean making use of information which would not be available for real retrieval observations.

Because of these difficulties, we decided to not pursue this approach in the study. However, since this is an important point to mention, we will add a paragraph on this issue in the discussion.

**2 Specific comments**

**Reviewer comment 1**

Lines 85-88: I recommend the use of geographical spatial references (i.e.,north/south rather than left/right)

**Author response**

The proposed change will be adopted in the revised version of the manuscript.

**Reviewer comment 2**

Line 98 (also 176,252,449): Instead of vertical/horizontal (which are dependent on the convention used for plotting), I recommend the use of concentration/size tocharacterize the dimensions of the particle size distribution.

**Author response**

The proposed change will be adopted in the revised version of the manuscript.

**Reviewer comment 3**

Line 100: A few more details on the Milbrant and Yau microphysics sheme that are relevant to this study would be helpful here. For example: What is the assumed shape(functional form) of the particle size distribution, and what are the prognostic variables(e.g., number concentration, mixing ratio)?

**Author response**

We will follow the reviewers comment and add the requested information to the manuscript.

**Reviewer comment 4**

Line 135: Does the ARTS radar solver also provide analytic Jacobians?

**Author response**

Yes, it does. A sentence will be added to the description of the forward model to clarify this.

**Reviewer comment 5**

Line 187: "particles" should be "particle"

**Author response**

The sentence will be removed in the revised version of the manuscript since the information it conveyed was deemed irrelevant.

**2.1 Reviewer comment 6**

Line 198: Is Dm also only retrieved at these 10 points, or just N0* (and Dm retrieved in each radar range gate as in Grecu et al. 2016)?

**Author response**

$D_m$ is actually retrieved at the resolution of the GEM model scenes. Since questions about the retrieval grids were also raised by the other reviewers, we will add an illustration of the grids applied in the different retrieval configurations to the manuscript.

**2.2 Reviewer comment**

7. Line 256: Actually, this is only one example of how the radar and radiometer measurements can be complementary. Even if the lines were parallel (and thus no information distinguishing size from concentration could be obtained), the radar still locates the cloud and describes its vertical structure. One can imagine a cloud of the same ice water path and particle size at two different heights having different brightness temperatures due to changes in the water vapor absorption above the cloud – having the radar information would provide increased information content about the ice water pathin this case than the radiometer measurement alone.

**Author response**

It is certainly correct that when a radar sensor is added to a passive observation system one of the advantages will be the increased resolution. However, what we are interested in are the advantages that neither of the two instruments can provide on its own. If it was only about vertical resolution, then the radar alone would be the ideal observation

system. In this sense, we do not consider the vertical resolution a synergy of the two sensors.

To make this clear, we will add an explanation of our definition of synergies between the active and passive observations to the section which discusses the complementary information content.

**Reviewer comment 8**

Table 4: Why are the values for GemSnow and GemGraupel different than in Table 1?

**Author response**

This was by mistake and will be corrected in the revised version of the manuscript.

**Reviewer comment 9**

Figures 7 and 8: I'm not sure why these are separate figures – it seems like all panels could fit on one page.

**Author response**

Figures 7 and 8 will be combined into a single figure in the revised manuscript.

**Reviewer comment 10**

Figure 10 is missing from the manuscript.

**Author response**

Figure 10 will be included in an Appendix to the revised manuscript with the rest of the analysis of the results from the second test scene.

**Reviewer comment 11**

Line 374: recommend using "represent" instead of "predict"

**Author response**

The proposed change will be adopted in the updated version of the manuscript.

**Reviewer commene 12**

Line 382: should be "reference" instead of "references"

**Author response**

This will be corrected in the updated version of the manuscript.

**Reviewer comment 13**

Line 414: How are the truncated PSDs (using GemSnow) represented in theforward simulations? Is total ice water content conserved? If so, how is it spread amongthe valid particle sizes – equally, or is the truncated mass allocated to the smallest size bin?

**Author response**

Total IWC is not conserved in the handling of PSDs. The point raised by the reviewer has been investigated by assessing the effect of the truncation on the water content of snow in the forward simulations. The results of the analysis are given in the figure below. As these results show, the effects of the truncation in the forward simulations are negligible. However, when the GemSnow particle model is used in the retrieval it can introduce significant errors. For this reason as well as another reviewers' comment regarding the choice of tested particles, the selection of particles to be used in the retrieval will be changed for the revised manuscript and the GemSnow particle will be replaced by a habit mix which uses the GemSnow particle for large diameters.

[Figure]

Figure 1: Joint distribution of truncated and full snow water content (SWC) for the two test scenes.

**Reviewer comment 14**

Figure 16: The figure labels/captions aren't clear if they refer to total liquid water content/path or just the cloud liquid water/path.

**2.2.1 Author response**

We will clarify that the contours refer to liquid cloud water content in the revised version of the manuscript.

**Reviewer comment 15**

Line 518: It's interesting that the Plate Aggregate provides the most accurate re-trieval results, even though it isn't similar to the models used in the synthetic measure-ment simulations. Does the decreasing density with size better replicate the combina-tion of high-density GemCloudIce (which tends to be present in high concentrations atsmall sizes) and lower-density GemSnow (which tends to be dominant at larger sizes)?

**2.2.2 Author response**

Unfortunately, we cannot give a definitive answer to this question. As panel (a) in Fig. 15 shows, the density of the LargePlateAggregate habit is actually lower than that of snow for large particle sizes. Moreover, the scattering properties certainly also play a role here. At this point we are therefore not able to postulate any direct causality between the particle density and the performance in the retrieval.

**References**

Buehler, S. A., Mendrok, J., Eriksson, P., Perrin, A., Larsson, R., and Lemke, O.: ARTS, the Atmospheric Radiative Transfer Simulator – version 2.2, the planetary toolbox edition, Geosci. Model Dev., 11, 1537–1556, https://doi.org/10.5194/gmd-11-1537-2018, URL https://www.geosci-model-dev.net/11/1537/2018/, 2018.

Milbrandt, J. and Yau, M.: A multimoment bulk microphysics parameterization. Part II: A proposed three-moment closure and scheme description, J. Atmos. Sci., 62, 3065–3081, https://doi.org/10.1175/JAS3534.1, 2005.

---

## Author Comment (AC2) · 10 Feb 2020

**Response to reviewer 2**

We thank the referee for the time he/she has put on reading our manuscript and providing feedback.

Based on the combined comments of the referees, we have decided to implement these general changes:

- We will switch to an airborne measurement set-up and the introduction section will be modified accordingly

- The text in the result section will be shortened significantly

- Redundant results for scene 2 will be placed in an appendix

- The selection of tested retrieval habits will be revised/changed

Below we respond to the main questions raised by the referee, and outline how we will revise the manuscript.

**1 Major points**

**Review comment 1**

A major aspect of the study concerns the representation of the particle size distribution which is retrieved by two free parameters (different from the 2 moments of the atmospheric model GEM used to provide the test scenes) and the assumptions of the particle type. The difficulty of connecting atmospheric model output to single scattering properties (which is one of the fundamental assumptions) could be better explained. The motivation why the authors choose their approach and why they test certain settings need to be discussed in the beginning. Couldn't Tab. 1 and 4 be combined and better explained which is used for which purpose? Why is cloud ice the same and GEMsnow and GEMGraupel different in both?

**Author response:**

We agree with the comment that the rather arbitrary choice of tested particles is a weak spot of the study. To improve this, the experiments will be repeated with a more principled selection of particles. The new selection is based on the particle properties described in Ekelund et al. (2019) and covers a broader range of mass-size relationships and scattering parameters. In particular, the GemSnow model has been removed from the selection of test particles because it does not cover small ice particle sizes.
We propose to make the following changes to the manuscript:

- Add a paragraph to the description of the GEM test scenes which explains the particle models that have been developed to match the assumptions of the GEM microphysics scheme and that are used to simulate observations.

- Add a paragraph to the description of the retrieval implementation that explains the difficulty of representing the complex mixture of different particles in the GEM model scenes with a single particle model as well as the new selection of particle models and habit mixes.

- Correct the error in the reported parameters of the mass-size relationship for the GEM Snow and GEM graupel.

However, because of these changes it will not be possible to combine Tab. 1 and Tab. 4, since the convey slightly different information.

**Reviewer comment 2**

Although different parameterizations of the hydrometeor types are used to study their effects, vertical changes (development of sedimenting particles) are not considered. Similar polarization effects are not mentioned in the discussion on particleshape. Otherwise the paper nicely discusses the different aspects but in the end I ammissing a clear message on the outcome of the test (choice of particle types). What isrecommended for the future?

**Author response**

The first statement made by the reviewer is not fully correct. Since the retrieval can reduce the concentration of particles and increase their size it can modify the ratio of small and large particles and thus represent the effects of sedimentation on the PSD.
Vertical changes in particle shape, i.e. transition from single crystals to aggregates, are represented indirectly through the particle size. The particle models used here are taken from standard habits from the ARTS SSDB described in Eriksson et al. (2018). Some of them combine pristine crystals at small particle sizes with aggregate shapes at larger sizes.
Polarization effects in the simulations were ignored for the simple reasons that the model scenes do not provide information on particle orientation or aspect ratios and that suitable scattering data for oriented particles has only recently been released (Brath et al., 2019).

For the revised version the sensors are assumed to point at nadir, which justifies neglecting polarization effects. Nonetheless, particle orientation can still have an effect on the observations. We will state clearly in the revised manuscript that polarization effects will have non-negligible impact on the observations of the MWI and ICI sensors.

We agree that in choice of the particle shape was not described well in the manuscript. To address this as well as to provide a clearer message on the outcome of our tests we propose the following changes for the revised manuscript:

- State clearly in Sect. 2.2 that for MWI and ICI polarization effects can not be neglected.

- Improve the description of the employed particle models in Sect. 2.3.

- Extend the discussion of the tested particle shapes to derive clearer recommendations for the future.

**Reviewer comment 3**

Not only the two moments of the ice PSD but further variables are retrieved and their information content is nicely shown in Fig. 14. I am surprised that the information on moisture is so low although information along three water vapor lines is provided? This should at least in the upper atmosphere provide information? Is it due to the choice of relative humidity which mainly depends on temperature? I am also skeptical about the results of Fig. 16. Basically, there should be no liquid for temperatures colder than 40 deg C (freezing) but it even reference LWC goes up to 13 km? I would not support the statement on L568 – where is the evidence? Similar L527 – Liquid water estimation within mixed phase clouds is extremely difficult and if ICI and radar could really do that together this would be worth a separate paper. To better understand the information content, I suggest to plot the profiles of cumulative degrees of freedom for the different retrievals as this could help interprete where and how the synergy works.

**Author response**

As can be seen in Fig. 8 from Eriksson et al. (2019), the information content on water vapor from ICI alone are at most 4 degrees of freedom for clear-sky scenarios. Since in the retrieval also the channels from MWI are included, the expected information content on water vapor should be somewhat higher. However, this is for clear-sky scenarios. In the presence of clouds, the information content will be significantly reduced.

Regarding the results of the retrieved cloud liquid water content (CLWC), Fig. 16 shows quite clearly an improvement, both in terms of CLWC and cloud liquid water path (CLWP), in the results of the combined retrieval compared to the passive-only retrieval. Yes, liquid clouds droplets are present at high altitudes in the first model scene but only below the 230 K isotherm. However, since this is the case only for the first scene, it does not seem relevant for the interpretation of Fig. 16.

To shed more light onto the information content regarding humidity and CLWC we will follow the referee's suggestion and replace the bar diagrams in the manuscript with a

figure (Fig. 1 shown below) which displays the cumulative DFS for all profiles in the test scenes.

[Figure]

Figure 1: Degrees of freedom for signal for all retrieval configurations and both test scenes obtained with the Large Plate Aggregate model. The colored areas in each plot represent the contribution to the cumulative degrees of freedom from each retrieval quantity. Results for the first and second test scene are displayed in the first and second row, respectively. The first, second and third panel in each row show the results for the passive-only, radar-only and the combined retrieval.

**Reviewer comment 4**

The manuscript is rather lengthy making it difficult for the reader to extract the major-points. I strongly suggest a) to move part of the analysis into an appendix (, b) remove double statement (see minor comments, also the LWC plot) and c) to remove figurecaption like information (for example L92 or "filled contours") from the text. The text must make sense without looking at the figure. Figure only support the statements made in the text. Lengthy descriptions such as "The plot shows.." need to be avoided.

**Author response**

We will follow all the referee's recommendations to make the manuscript more concise.

**2 Minor comments**

**Reviewer comment 1**

L39: Is sensitivity really the right word? Range resolution is the main advantage –signal-to noise range depends on distance and hydrometeor distribution,

**Author response**

Following the reviewer's suggestion the sentence will be rewritten.

**Reviewer comment 2**

L48: MWI will also cover new spectral channels, e.g. 118 GHz

**Author response**

We will incorporate this information into the introduction.

**Reviewer comment 3**

L62: "high-resolution" is always relative for a model. I would recommend avoiding this term and use Cloud resolving Model (CRM).

**Author response**

The proposed improvement will be adopted in the revised version of the manuscript.

**Reviewer comment 4**

L68: After you mention GPM (with scanning radar) it might be good to say that you are only looking at a nadir pointing radar (curtain). The swath center came bit as a surprise.

**Author response**

We will incorporate this information in the introduction as suggested.

**Reviewer comment 5**

L70: There has been quite some literature about combining active and passive MWusing a Bayesian framework which should be acknowledged, e.g. Grecu, M., & Olson,W. S. (2006), Johnson et al. (2012) , Munchak, S. J., & Kummerow

[Figure]

Figure 2: Scatter plots for the second test scene showing the retrieved IWC plotted against the IWC in the GEM model scene for the passive-only, radar-only and combined retrieval.

**Author response**

Following the suggestion of the reviewer a paragraph listing previous work on synergistic retrievals using radar and passive radiometers at lower microwave frequencies will be added to the introduction.

**Reviewer comment 6**

L84: Test scenes have a grid resolution of 1 km horizontally. As this is not the true model resolution I would have recommended to coarse sample the model data (maybe every 5th data point) and include more diverse profiles instead. This might be especially interesting for the scatter plots.

**Author response**

The point raised by the reviewer here is certainly correct. However, the decision to restrict simulations to two test scenes was motivated primarily by the computational costs of performing the retrievals. The scatter plot in Fig. 10 (shown in Fig. 2 below), which was unfortunately missing from the manuscript , shows that the emergent structures are consistent for both test scenes. This indicates that the scenes cover sufficient profile variability to be statistically representative.

**Reviewer comment 7**

Motivation lacking: "To perform RT simulations for each GEM profile the PSD needs to be diagnosed from the prognostic GEM variables, i.e. N and m.."

**Author response**

Since also reviewer 1 requested changes in the corresponding paragraph, it will be rewritten taking into account the reviewer's suggestion.

**Reviewer comment 8**

L92:"prognoses" means forward in time - you mean diagnosed, calculated,determined..

**Author response**

The word will be replaced by derived in the revised version of the manuscript.

**Reviewer comment 9**

L98: I find the term "horizontal and vertical scaling" strange – why not saying PSD shape is similar but scaling in respect to diameter and number density. At least definethe term clearly the first time that you use it or define a short for it.

**Author response**

This issue was also mentioned by reviewer 1 and will be addressed in the revised version of the manuscript.

**Reviewer comment 10**

L103: model test – be careful also at other instances that "model" can mean too many-things. Here I would say GEM test scenes.

**Author response**

The proposed change will be adopted in the revised version of the manuscript.

**Reviewer comment 11**

L119: Need to clearly say that polarization effects are neglected though these can be-several Kelvin, e.g. Xie et al., 2015. You ignore this effect but even consider noise reduction.

**Author response**

See response to general comment 2.

**Reviewer comment 12**

L157-159: needs to be better motived, references?

**Author response**

We will provide a better motivation of the use of the $\chi^2$ statistic in the revised manuscript.

**Reviewer comment 13**

L172: I doubt that the model has constant vertical resolution. It will be better close to the surface and worse aloft. This should be mentioned than GEM is introduced.

**Author response**

As suggested by the reviewer, this will be mentioned when the GEM test scenes are introduced. Moreover, a sketch will be added to the manuscript which displays the GEM model grid and the grids of all retrieval quantities for the retrievals.

**Reviewer comment 14**

L 174: for all hydrometeor species of the model? It would be helpful to first introduce all retrieval quantities – I was missing a motivation for the paragraph around L195. How do you define the freezing level (and later the troposphere)? How do they vary in both test scenes? The model also likely has supercooled liquid water above the freezing layer – how is this treated?

**Author response**

As mentioned above, a figure will be added to the manuscript that displays all retrieval variables as well as the freezing level and the tropopause. Moreover, we will add an explanation of how the freezing level and tropopause are defined in the manuscript.
For the simulated observations, supercooled liquid is treated in just the same way as liquid water below the freezing level. As described in the paragraph around L. 211, cloud liquid water in the retrieval is treated as purely absorbing and simulated using a parametrized absorption model. Moreover, it is restricted to temperatures of 230 K and up.

**Reviewer comment 15**

L 198: Vertical resolution of retrieval grid: Why 4 points? The freezing layer must be very different for both cased. Maybe a sketch would be helpful as later on lines 230 the different vertical resolutions for other variables is discussed?

**Author response**

We have revised the retrieval implementation and now use fixed retrieval grids with a resolution of 2 km for the $N_0^*$ parameters. Reducing the resolution of the retrieval grids for the $N_0^*$ parameters was found to aid the convergence of the retrieval.

The freezing level does indeed vary between the two scenes. The freezing levels of both scenes will be added to Fig. 1.

**2.1 Reviewer comment 16**

L281: How do I know that Large Plate is the best performing model? Which parameter,plot, table does show that?

**Author response**

This section will be revised to make it clear which results show that the Large Plate Aggregate yields the best performance.

**Reviewer comment 17**

L283-L307: Can be shortened significantly

**Author response**

The proposed change will be adopted in the revised version of the manuscript.

**Reviewer comment 18**

L332: There can I see that? Give figure?

**Author response**

A reference to the figure will be added in the revised version of the manuscript.

**Reviewer comment 19**

L325: The two paragraph here give similar information -> streamline

**Author response**

The proposed change will be adopted in the revised version of the manuscript.

**Reviewer comment 20**

L333-344: I would put this to the appendix

**Author response**

We will follow the reviewers advice and put the analysis of the second test scene into the appendix.

**Reviewer comment 21**

L444: Here it needs to be made clearer how this goes beyond what GPM is doing.

**Author response**

To clarify how our work goes beyond what GPM a paragraph detailing this will be added to the discussion.

**Reviewer comment 22**

L495: "does not say much about the general validity of the assumption". Here you should dig in a bit more. What is the role of a priori and covariances?

**Author response**

Following the suggestion of the reviewer we will extend the discussion of the a priori assumptions.

**Reviewer comment 23**

L560: Rethink the bullet structure. 2. Is not an independent result. For each result refer to the part of the manuscript where you can clearly see that. Especially result 3 should be detailed – how do ICI channel advance the currently available data?

**Author response**

The bullet points will be remove in the revised manuscript and replaced by a text which presents the conclusion in a logically coherent way.

**Reviewer comment 24**

Fig. 3: Is it really worth having the slightly different size distribution shapes for frozen and liquid? Isn't there a stronger difference between different frozen hydrometeors

**Author response**

This is certainly true but in most clouds ice and rain can be distinguished fairly well a priori, which simplifies treating them as different species using different PSD shapes. Distinguishing between different frozen hydrometeors is difficult to do a priori and using multiple species of hydrometeors in the retrieval was found to cause ambiguities which the retrieval is not able to resolve.

**Reviewer comment 25**

Figures 7 and 8: I'm not sure why these are separate figures – it seems like allpanels could fit on one page.

**Author response**

Figures 7 and 8 will be combined into a single figure in the revised manuscript.

**Reviewer comment 26**

Fig. 4 and also in text: "cloud signal" say that this is dTB.

**Author response**

Following the reviewers recommendation, the passive cloud signal will be referred to in the text as $\Delta T_B$ and the radar signal as $\mathrm{dBZ_{max}}$.

**Reviewer comment 27**

Fig. 5: Can you add freezing layer height?

**Author response**

We propose to add the freezing layer height to Fig. 1, since this will allow showing the freezing level height for both scenes, which addresses minor comment 15 as well.

**Reviewer comment 28**

Fig. 6: It would be nice to see the absolute values of IWP somewhere. Maybe you could add another time series with IWP as the sum of the different components such that the reader can see where the different categories (cloud, graupel, snow and hail) contribute most?

**Author response**

To address the reviewers request we will add absolute IWP of the reference scene and its decomposition into different hydrometeor classes to Fig. 6.

**Reviewer comment 29**

Fig. 7: I think it is retrieved vs. truth. The word following is not really exact. Why not put 7 and 8 together?

**Author response**

Fig. 7 and 8 will be merged and the caption will be corrected.

**Reviewer comment 30**

Fig. 9: Could go to the appendix

**2.1.1 Author response**

Fig. 9 will be moved to the appendix.

**Reviewer comment 31**

Fig. 10 I only see the caption???

**Author response**

Fig. 10 was unfortunately missing from the manuscript. The figure will be included in the appendix with the analysis of the second test scene.

**Reviewer comment 32**

Tab. 1. Assumed particle model information for each hydrometeor class given by GEM-model. In fact it could be good to combine it

**Author response**

In order to give a better overview of the particle models that are used in the retrieval Tab. 4 will be extended in the revised version of the manuscript. This, however, would make merging Tab. 1 and 4 slightly confusing so we decided against the reviewers recommendation.

**Reviewer comment 33**

Tab.3 : I would recommend to add a first column with a spelled out name

**Author response**

We will add the column to the revised version of the manuscript.

**Grammar, typos and reformulations**

The authors would like to thank the reviewer for the additional comments, all of which will be incorporated into the revised manuscript.

**References**

Brath, M., Ekelund, R., Eriksson, P., Lemke, O., and Buehler, S. A.: Microwave and submillimeter wave scattering of oriented ice particles, Atmospheric Measurement Techniques Discussions, 2019, 1–38, https://doi.org/10.5194/amt-2019-382, URL `https://www.atmos-meas-tech-discuss.net/amt-2019-382/`, 2019.

Ekelund, R., Eriksson, P., and Pfreundschuh, S.: Using passive and active microwave observations to constrain ice particle models, Atmospheric Measurement Techniques Discussions, 2019, 1–30, https://doi.org/10.5194/amt-2019-293, URL `https://www.atmos-meas-tech-discuss.net/amt-2019-293/`, 2019.

Eriksson, P., Ekelund, R., Mendrok, J., Brath, M., Lemke, O., and Buehler, S. A.: A general database of hydrometeor single scattering properties at microwave and submillimetre wavelengths, Earth Syst. Sci. Data, 10, 1301–1326, https://doi.org/10.5194/essd-10-1301-2018, 2018.

Eriksson, P., Rydberg, B., Mattioli, V., Thoss, A., Accadia, C., Klein, U., and Buehler, S. A.: Towards an operational Ice Cloud Imager (ICI) retrieval product, Atmos. Meas. Tech., 2019, 1–30, https://doi.org/10.5194/amt-2019-312, URL `https://www.atmos-meas-tech-discuss.net/amt-2019-312/`, 2019.

---

## Author Comment (AC3) · 10 Feb 2020

**Response to reviewer 3**

We thank the referee for the time he/she has put on reading our manuscript and providing feedback.

Based on the combined comments of the referees, we have decided to implement these general changes:

- We will switch to an airborne measurement set-up and the introduction section will be modified accordingly

- The text in the result section will be shortened significantly

- Redundant results for scene 2 will be placed in an appendix

- The selection of tested retrieval habits will be revised/changed

Below we respond to the main questions raised by the referee, and outline how we will revise the manuscript.

**Major comments**

**Reviewer comment 1**

The paper is presented as an application for ICI in combination with a Cloudsat like configuration but it is not clear to me what geometry of observations the authors are thinking about. They state "As mentioned above, the same incidence angle as for the passive radiometers is assumed also for the radar. In practice, this could be achieved by remapping the radar observations to the lines of sights of the passive beam". Are they thinking about a scanning W-band radar? or at a off-nadir pointing radar? If the former is true then they should discuss what is a realistic technological solution (and what are the consequences in terms of sensitivity) and the authors should refer to state of the art scanning W-band radar concepts (there is none at the moment!); if the latter is true they should discuss what are the consequences of such a selection (e.g. forground clutter) and they need to convince me that what we could gain from such a configuration compensate from the loss of information introduced by pointing in such a slanted direction. There should be a certain degree of "realism" in what we are trying to simulate, especially if this was part of an ESA study.

**Author response:**

The reviewer raises a very relevant point with his comment. To address this, we changed our simulation setup to simulate perfectly co-located observations at nadir. Realistic modeling of a space-borne viewing geometry (at least in a variational retrieval) is currently not feasible due to the computational complexity. We still deem this sufficient for the scope of the study, i.e. studying the fundamental synergies between active and passive observations. In addition to this, we follow the recommendation made in the second comment and will pitch the application more towards air borne observations.

**Reviewer comment 2**

"the beams of all three sensors are modeled as perfectly coincident pencil beams". Again this is quite an assumption. Non uniform beam filling will play a key factor. This is one of the many simplifications (no polarization, no multiple scattering,1D, ...) that needs to be clearly listed at the beginning of Sect.2.2.1 (some appear only at page 27). For this reason I would actually pitch more towards an airborne configuration where these simplification indeed can be realistically assumed or of a radar with a radiometric mode (where you can actually match footprints). Otherwise the (not massive) gain of having a radar-radiometer combination that you show later on can be completely washed out by the errors introduced to these assumptions. I imagine that you may also have airborne data where to test how realistic your forward model is.

**Author response**

As mentioned above, we will follow the reviewer's suggestion to pitch the application of the combined retrievals more towards combined retrievals. We will also make these limitations more clear in Sect. 2.2.1 and discuss their implications more thoroughly.

**Reviewer comment 3**

Fig 2: these PSDs look very weird to me. Why do they have the plateau at small sizes? y-axis units are obviously wrong unless you are renormalizing by some mass (but it is not explained).

**Author response**

The reviewer is of course right, the units on the axis of the plots were indeed wrong and will be corrected in the revised manuscript. Otherwise, the PSDs correspond to the modified-gamma functions that are assumed in the Milbrandt and Yau (2005) microphysics scheme.

**Reviewer comment 4**

Fig 3: sorry I do not follow what is this (what is the y-axis?), and why this plot is meaningful.

**Author response**

We will remove this plot from the revised version of the manuscript.

**Reviewer comment 5**

Eq.6: Clearly with values lower than 230 K it does not make any sense (negative RH, or large than1.1???)

**Author response**

We would like to thank the author to point out this inconsistency, as there are indeed two mistakes in Eq. 6. The right equation should be

$$
\phi(t) = \begin{cases} 0.7, & 270 \text{ K} < t \\ 0.7 + 0.01 \cdot (t - 270), & 220 < t \leq 270 \text{ K} \\ 0.2, & t < 220 \end{cases} \tag{1}
$$

This will of course be corrected in the updated version of the manuscript.

**Reviewer comment 6**

Line 210; this means that the vertical resolution changes with the surface temperature, really weird choice.

**Author response**

We agree with the reviewer that the chosen retrieval grids may not be optimal. We will change them to fixed-resolution grids for the revised manuscript.

**Reviewer comment 7**

fIG4 : not clear to me why the scattering depression is not increasing at higher frequencies. I would expect that the optical thickness would drastically increase increasing frequency. Is this due to very large asymmetry parameters then? But this is not what I do see in Fig.5 (though Fig4 is of course a very idealized case) If this is the case then results will be very dependent on particle habits (which may introduce additional uncertainties in the retrieval)

**Author response**

It is correct that extinction increases rapidly with frequency, but the final scattering depression depends also on other factors. One consideration is the background absorption due to gases. A higher gas absorption decreases the effect of scattering, and this effect generally increases with frequency. It is correct that also the asymmetry parameter needs

to be considered, which increases with frequency. A higher asymmetry parameter gives a lower depression for a given cloud optical depth, see Fig. 5 of Eriksson et al. (2015). It can be hard to judge the scattering depression in a figure like Fig. 5, as the clear-sky values differ between the channels. In the version found below, extracted scattering depressions are shown in the second panel. For high-clouds with moderate cloud optical depth, the scattering depression increases monotonously with frequency, while in the most dense cloud region (around lat 2.7) this is not the case for the reasons discussed above.

[Figure]

Figure 1: Simulated brightness temperatures (Panel (a)) and cloud signal depressions computed for selected channels of the MWI and ICI radiometers for the first test scene.

**Reviewer comment 8**

8) Line 275: not clear what you mean, in Tab.4 there are 6.

**Author response**

What was meant here is that different ice shapes are tested for the single frozen hydrometeor species which is used in the retrieval. Tab. 4 lists the different shape models that were investigated.

Since the section describing the selection of particle models will be rewritten, this sentence will be reformulated to make it clearer.

**Reviewer comment 9**

9) "extends below the sensitivity limit of the passive-only observations around 10-5 kg m-3" : very sloppy sentence. Passive mi-crowave radiometer are sensitive to integrated contents!

**Author response**

As response to a comment from another reviewer the corresponding paragraph will be rewritten and this sentence will be removed.

**Reviewer comment 10**

Fig 6d: this retrieval looks really weird. Where are all the stripes coming from? Certainly this does not look like acloud, or? What kind of constraint have you imposed on the cloud top?

**Author response**

It is true that the passive only retrieval does not perform well in terms of the vertical structure of IWC. The reason for this is that the passive observations alone do not provide much information on the vertical distribution of ice. To correct for this, further regularization would be necessary which is not applied here in order to keep the comparison to the other retrieval methods fair. All of this is discussed in the discussion section of the manuscript.

**Reviewer comment 11**

"In general, the radar-only results exhibit only very weak dependency on the particle model, making the results for different particle shapes virtually indistinguishable." Again another dangerous sentence. We know (unfortunately) that this is not true (otherwise our ice problems would be sorted). Here my guess is that you have not properly explored the backscattering variability (particularly looking at the different degree of riming). It is notclear to me whether there is enough variability in your ARTS database, I guess you are more focused at ice particles (including aggregates) but you are not considering really rimed particles. Regions where graupel is present should be avoided from the discussion of the radar-only retrieval for the simple reason that in those regions attenuation correction and multiple scattering effects make the problem very tricky. I guess that the radiometer as well is in serious trouble when entering those areas. Again, I would not start tackling regions the observation system is not tailored for.

**Author response**

As mentioned above, we will revise the particle habits used in the retrieval, but we expect that particle shape will continue to have a smaller impact on our radar-only retrieval. What the results shown in scatter plot in Fig. 7 and 8 indicate is that the uncertainty which can be attributed to the particle size distribution (PSD) is larger than that introduced by the assumed particle shape. However, it is difficult in general to draw a clear line between particle shape and PSD. This is especially true if particle size is described by $D$max, and the PSD is defined accordingly. In this case, IWC of a given PSD will depend on the particle's effective density, and e.g. degree of riming becomes

critical. Accordingly, to what extent retrieval errors are due to shape or PSD, depend partly on definitions.

The ARTS single scattering database does include several types of rimed particles. Two of them are the GEM Graupel and GEM Hail models which are used in the simulation of the synthetic observations. For the retrieval, however, it is true that we did not include rimed particles in the tested particle models but this will be changed for the revised version of the manuscript.

Both the forward simulations and the retrieval handle attenuation consistently. We therefore think it is worth considering even regions where graupel is present as this allows us to assess the uncertainties caused by not having a realistic representation of rimed particles in the retrieval.

It is certainly correct that for space-borne observations multiple scattering needs to be considered and this will add complexity to the retrieval. Here, however, we can avoid this extra complexity as we use simulated observations which do not include multiple scattering.

**Reviewer comment 12**

Fig.10 is missing!!!

**Author response**

Fig. 10 was unfortunately missing from the manuscript. The figure will be included in the appendix of the revised version together with the analysis of the second test scene.

**Reviewer comment 13**

"Since the calculation of the AVK involves the forward model Jacobian, this effectmust be related to the non-linearity of the forward model" well I would avoid such veryspeculative statements.

**Author response**

Following the suggestion of the reviewer, the sentence will be removed from the manuscript.

**Reviewer comment 14**

You need to be very careful how you present the results in Fig. 14. The conclusions that I can draw is the following: a CloudSat like radar is pro-viding much more information than the ICI+MWI radiometers when characterizing ice particles (really the radiometer is providing some additional water vapour information). As a result we should invest in the former and not the latter. While I may agree with the previous statement and strongly support a CloudSat-like radar on an operational mission my feeling is that you are pitching your radiometer system at the wrong kind of scenes (I already see an improvement going from the first to the second scene). I would have selected completely

different scenes (including high latitude clouds with mixed phase). It is to me an overkill to try to retrieve D_M of rain for these scenes from your PMW radiometer suite of sensor. If you have any skill in warm rain you should properly prove it

**Author response**

Our interest in this study is neither arguing for one nor the other observation system. The question that we want to address is whether combined observations have extra value compared to separate observations. Such combined observations could be achieved by performing joint flights with the aircraft carrying the ISMAR sub-millimeter radiometer and another one carrying a radar, by flying a cloud radar in constellation to Metop-SG, or by adding a sub-millimeter radiometer to the platform carrying some future cloud radar. We consider it out of the scope of this study to judge the cost effectiveness of either of these solutions.

As the referee clearly favours radars, we would like to balance this by mentioning that passive instruments have an additional strength in their much higher areal coverage. The swath of ICI and MWI is about three orders of magnitude broader than that of CloudSat and EarthCARE.

Although a cloud radar certainly provides more information on frozen hydrometeors than ICI, our results clearly show that also radar observations alone are insufficient to accurately determine the microphysical properties of ice hydrometeors (Fig. 4, 7). The passive adds information on the microphysics of the clouds to the radar (note the significant increase in information content on $N_0^*$ in Fig. 14) which helps to reduce retrieval uncertainties (Fig. 11). Although it is not clear whether these improvements carry over to space-borne observations, our results clearly show this as a synergy between the passive and active observations (esp. Fig. 4, 11, 14).

The cloud scenes used in the manuscript were selected with the aim of providing a representative sampling of the type of clouds present in the two model scenes that were available for the study. We did not want to cherry pick scenes were the retrieval works well to provide a more realistic assessment of the retrieval.

Rain must be handled in the retrieval due to its effect on the passive radiances. However, we never claim that we have any skill in retrieving warm rain and so we do not agree that we are required to prove to have it.

**Reviewer comment 15**

LWP and Fig.16. I have a serious problem here. The cloud I see on the right is a liquid cloud. So how it is possible that your radiometer is doing sobadly in the LWP retrieval and why the combined is so much better? I guess this must go back to understanding surface emissivity and integrated water vapour (maybe somecomments there should be made to explain what kind of surface/IWP we are dealingwith). You have not included radar path integrated attenuation in your retrieval (like istypically done in radar retrievals) but this could of course help in this case.

**Author response**

The cloud in the right of the scene is a mixed-phase cloud. There are several explanations for why the retrieval does not work well here: First of all, our observations setup does not make use of the channels around 23 GHz, which are typically used for retrieving LWP. And also here the performance of the passive-only retrieval suffers from the lack of a priori information on the vertical position of the cloud. Since liquid water at higher altitudes has a stronger impact on the observations, the retrieval puts too little cloud water too high in the atmosphere because of its inability to locate it properly. This is discussed in Sect. 4.2.3 of the manuscript.

**Reviewer comment 16**

I do not think that for OE to work The forward model must be linear as stated at line 544.

**Author response**

The OEM can of course be applied to non-linear problems but a complication that arises is that it can get stuck in secondary minima. The sentence will be corrected in the revised version of the manuscript.

**Reviewer comment 17**

17)Sect.4 and 5: a lot of waffling here (e.g. the three bullet conclusion, you need to bemuch more quantitative and linked to what you have proved; the three statements aresomething I could have formulated on my own without making any simulation). Again the conclusions must be related to the cloud regime you are considering (and cannot be valid for all!)

**Author response**

One of the main advantages that we see in the combined retrieval is that it actually works for a wide range of different cloud regimes. If the cloud regime was known a priori, good results can probably be achieved using only a radar and suitable a priori assumptions. In general, however, this is not the case, which leads to the uncertainties that we currently have in the observational record for IWP and IWC.
For the revised manuscript, we will rewrite the conclusion and parts of the discussion to make it more concise and the point mentioned above more clear.

**1 Minor comments**

**Reviewer comment 1**

I would avoid the use of "ice mass density" and use "ice water content"

**Author response**

The proposed changes will be adopted in the revised version of the manuscript.

**Reviewer comment 2**

Table 2: it would be good to see footprints as well

**Author response**

Since in the revised manuscript an airborne viewing geometry will be considered the footprint sizes of MWI and ICI are not relevant anymore.

**Reviewer comment 3**

Line 130: dBZ are the wrong units for a std of a reflectivity!

**Author response**

We are unsure what the reviewer is referring to here since quantifying uncertainty in the radar observations in dBZ seems to fairly common. This is for example how it is handled in the DARDAR cloud (Delanoë and Hogan, 2010) product as well as in the study by Jiang et al. (2019).

**Reviewer comment 4**

Line 180: "The remaining shape of each PSD is described by the shape parameters alpha and beta, not to be confused with the parameters of themass-size relationship shown in Tab. 1."; very confusing. Why are you using the same letters????

**Author response**

We used the same letters to be consistent with the definition and used in Delanoë et al. (2014) and Cazenave et al. (2018). However, since the explicit values of the $\alpha$ and $\beta$ parameters are probably of little interest for the average reader, we will simply refer to Cazenave et al. (2018) and not name the parameters explicitly.

**Reviewer comment 5**

Line 193: wrong units

**Author response**

This will be corrected in the revised version of the manuscript.

**Reviewer comment 6**

Line 199: English

**Author response**

This will be corrected in the revised version of the manuscript.

**Author response 7**

Line 35 page 2 (not really limited,this is a wide range!!)

**Author response**

The corresponding sentence will be reformulated in the revised manuscript.

**Reviewer comment 8**

Line 54 page 2. maybe it is worth mentioning all the heritage coming from radar-radiometer retrievals with W-band (Ka and Ku-band) radars with PMW radiometers.

**Author response**

Following the suggestion of the reviewer, a paragraph that mentions previous work on synergistic retrievals using radar and passive radiometers at lower microwave frequencies will be added to the introduction.

**Reviewer comment 9**

Line 229: "troposphere" is too generic Line

**Author response**

The use of the word *troposphere* and should have been *tropopause*. This will be corrected in the revised version of the manuscript.

**Reviewer comment 10**

Fig 4 caption: you need to include how thick is the layer.

**Author response**

This will be included in the revised version of the manuscript.

**Author response 11**

250: rho is not defined

**Reviewer comment**

$\rho$ will be defined in the revised version of the manuscript.

**Reviewer comment 12**

Line 4: 272.5????

**Author response**

This mistake will be corrected in the revised version of the manuscript.

**Reviewer comment 13**

Fig 4 caption: you need to include how thick is the layer.

**Author response**

The layer thickness will be added to the figure caption in the revised version of the manuscript.

**References**

Cazenave, Q., Ceccaldi, M., Delanoë, J., Pelon, J., Groß, S., and Heymsfield, A.: Evolution of DARDAR-CLOUD ice cloud cloud retrieval: new parameters and impacts on the retrieved microphysical properties, Atmos. Meas. Tech. Discuss., 2018, 1–24, https://doi.org/10.5194/amt-2018-397, URL `https://www.atmos-meas-tech-discuss.net/amt-2018-397/`, 2018.

Delanoë, J. and Hogan, R.: DARDAR-CLOUD, URL `www.icare.univ-lille1.fr//projects_data/dardar/docs/varcloud-algorithm_description-v1.0.pdf`, 2010.

Delanoë, J., Heymsfield, A. Protat, A., Bansemer, A., and Hogan, R.: Normalized particle size distribution for remote sensing application, J. Geophys. Res.-Atmos., 119, 4204–4227, https://doi.org/10.1002/2013JD020700, 2014.

Eriksson, P., Jamali, M., Mendrok, J., and Buehler, S. A.: On the microwave optical properties of randomly oriented ice hydrometeors, Atmos. Meas. Tech., 8, 1913–1933, https://doi.org/10.5194/amt-8-1913-2015, 2015.

Jiang, J. H., Yue, Q., Su, H., Kangaslahti, P., Lebsock, M., Reising, S., Schoeberl, M., Wu, L., and Herman, R. L.: Simulation of Remote Sensing of Clouds and Humidity From Space Using a Combined Platform of Radar and Multifrequency Microwave Radiometers, Earth Space. Sci., 6, 1234–1243, https://doi.org/10.1029/2019EA000580, URL `https://agupubs.onlinelibrary.wiley.com/doi/abs/10.1029/2019EA000580`, 2019.

Milbrandt, J. and Yau, M.: A multimoment bulk microphysics parameterization. Part II: A proposed three-moment closure and scheme description, J. Atmos. Sci., 62, 3065–3081, https://doi.org/10.1175/JAS3534.1, 2005.

---

## Author Response (AR1)

**Response to comments**

**Synergistic radar and radiometer retrievals of ice hydrometeors**

This document contains the responses to the comments of each reviewer followed by the marked-up differences of the manuscript and the revised version. For each comment the author's response and, if applicable, the corresponding changes in the manuscript are listed. Line numbers of changes are given with respect to the revised manuscript.

In response to the combined comments of the referees, all computations have been repeated and most of the manuscript has been rewritten. The following general changes have been implemented:

- We have switched to an airborne measurement set-up and the manuscript has been modified accordingly.

- The text in the result section has been shortened.

- Redundant results for scene 2 have been placed in an appendix.

- The selection of tested retrieval habits has been revised and changed.

**1 Comments from referee 1**

**1.1 General comments**

**Reviewer comment 1**

As noted in Section 4.2.4, the a priori assumptions do not describe reality very well. In particular, I suspect that the information content of Dm and N0* is highly dependent on the a priori assumptions of these two variables in the retrieval framework. Especially with a radar measurement, since Z is sensitive to both parameters over a wide range of the parameter space, the relative sensitivity and therefore information content will almost entirely depend on the relative constraints on these parameters imposed by Xa and Sa. As such it is imperative to accurately characterize these. I understand the choice to use the DARDAR constraints, but its clear from the cross-section plots that the model ice particle concentrations vary over a much wider range than the roughly 2 orders of magnitude that Eq. 4 provides over a 220-272 K temperature range. So, when the retrieval results are compared to model reality, it seems that a lot of N0* variability is folded into Dm and this is especially evident in Figures 13 and 14. My overall concern is that it is difficult to interpret some of the results when the model fields and the a priori assumptions differ so strongly.

**Author response**

To avoid potential misunderstanding we would like to point out that the variation of the a priori mean with temperature, which is given by Eq. 4, does not limit the retrieved values of $N_0^*$ to this range. How much $N_0^*$ is allowed to vary around the a priori mean is determined by the covariance matrix. Since the standard deviation for $\log_{10}(N_0^*)$ at each grid point was set to 2 (c.f. Tab. 3), $N_0^*$ is free to vary over several orders of magnitude in addition to the variation of the a priori profile.

Furthermore, the sentence in Section 4.2.4 was badly formulated and did not really express what we wanted to say there. The a priori assumptions are not generally bad for the model (after all the averaged results for the first scene are good). Rather, they are insufficient to accurately describe the (co-)variability of $D_m$ and $N_0^*$.

Nonetheless, the point raised by the reviewer certainly remains valid: In absolute terms, the interpretation of the retrieval results is dependent on the a priori assumptions. We argue here, however, that by applying equivalent a priori assumptions in all retrievals, we can still derive conclusions on the benefits of the combined retrieval approach based on a relative interpretation of the retrieval results. Our results indicate that the combined retrieval has to rely less on a priori assumptions than the radar-only retrieval. This is an

important advantage of the combined retrieval since if $D_m$ and $N_0^*$ could be constrained reliably a priori we would not have the uncertainties in the observational record of ice hydrometeors that we see today.

**Changes in manuscript**

- The discussion of the role of the a priori and its impact on results has been extended by adding the following paragraph to the manuscript:

   **Changes starting in line 481:**

   The a priori assumptions used in this study were chosen similar to those of the DARDAR-CLOUD product since they represent well established and validated assumptions for ice cloud retrievals. The role of the a priori is to complement the observations with additional information required to make the retrieval problem tractable. For the hydrometeor retrieval this means that the a priori determines how information from the observations, which alone is insufficient to determine both degrees of freedom of the PSD, is distributed between its $D_m$ and $N_0^*$ parameters. For the radar-only retrieval, this works well for cloud systems containing both ice and snow but leads to biased retrievals of both IWC and IWP when this is not the case (Fig. 9.  ). The DARDAR product uses co-located lidar observations to resolve the ambiguity where observations from both sensors are available. As our results show, this can be achieved also by combining a radar with passive microwave radiometers. However, while the overlap between lidar and radar is restricted to relatively thin clouds, microwave radiometers can provide sensitivity even inside thick clouds.

- The following paragraph has been added to the discussion of the limitations of the study which clearly states that the retrieval results should not be interpreted in absolute terms:

   **Changes starting in line 529:**

   An important limitation of this study is its scope: The aim here was not to develop a production-ready combined retrieval product but rather a proof-of-concept to explore this observational approach. The retrieval results presented here should therefore not be interpreted in absolute terms. The primary results are based on the relative performances of the three retrieval methods: Given equivalent a priori assumptions , the combined retrieval demonstrates higher sensitivity to the microphysical properties than the radar-only retrieval

and lower errors in terms of IWC than the passive-only retrieval.

- The paragraph starting on line 524 on the limitations of the OEM as retrieval method has been removed since its interpretation caused confusion and it was deemed to be of minor importance for the overall results of the study.

**Reviewer comment 2**

Forward model error is introduced when the different species present in the model microphysics are combined into one species and when different scattering models are used to represent the ice particles. That this is not represented in Se could lead to over-fitting and poor convergence (I suspect this is part of the reason why the normalized cost is much higher for the radiometer-including retrievals). It should be relatively easy to quantify this error by re-running the simulations with the retrieval assumptions(combining ice species, different scattering models), and I suspect that this error term would dominate the instrument noise term for many channels.

**Author response**

It is certainly true that the simplified forward model used in the retrieval introduces a forward modeling error and that it will likely dominate the sensor noise. However, we do not agree with the reviewer that this error is easy to quantify. First of all, the error will not be Gaussian and will depend on the cloud composition and the assumed particle shape, so that a more sophisticated error model would be required to describe the error accurately. Fitting such a model to the test scenes would likely yield overly optimistic results as this would mean making use of information which would not be available for a real retrieval scenario.

Because of these difficulties, we decided to not pursue this approach in the study. However, since this is an important point to mention, we will add a paragraph on this issue in the discussion.

**Changes in manuscript**

**Changes starting in line 513:**

It should be noted, that none of the presented retrievals accounts for the error caused by the simplified forward model and the choice of the particle model. This has not been pursued here because of the difficulty of fitting a suitable error model to these errors, which are likely non-Gaussian and scene-dependent. However, it is likely that accounting for them can improve retrieval performance and weaken the impact of the particle choice on the retrieval results.

**1.2 Specific comments**

**Reviewer comment 1**

Lines 85-88: I recommend the use of geographical spatial references (i.e.,north/south rather than left/right)

**Author response**

The proposed change has been adopted in the revised version of the manuscript.

**Changes in manuscript**

> **Changes starting in line 95:**
>
> The first test scene, shown in panel (a), is located in the tropical Pacific and contains a  mesoscale convective system in the  northern half of the scene and its anvil which extends into the southern half. The second scene, shown in panel (b), is located in the North Atlantic and contains an ice cloud in the  southern part and a low-level, mixed-phase cloud in the northern part.

**Reviewer comment 2**

Line 98 (also 176,252,449): Instead of vertical/horizontal (which are dependent on the convention used for plotting), I recommend the use of concentration/size tocharacterize the dimensions of the particle size distribution.

**Author response**

The proposed change has been adopted in the revised version of the manuscript.

**Changes in manuscript**

> **Changes starting in line 105:**
>
> The  assumed particle size distributions across different ice species vary mostly in their  scaling with respect to size and concentration, whereas the  normalized shape shows less variability.

> **Changes starting in line 177:**
>
> The PSDs of frozen hydrometeors and rain are represented using the normalized

particle size distribution formalism proposed by Delanoë et al. (2005). The PSD of a hydrometeor species at a given  altitude is modeled using a generalized gamma distribution function with four parameters. The mass-weighted mean diameter $D_m$, which scales the PSD along the size dimension, and the normalized number density $N_0^*$.

~~The retrieval computes vertical profiles of the two scaling parameters $D_m$ and $N_0^*$ for each of the two hydrometeor species. The remaining shape of each PSD is described by the shape parameters $\alpha$ and $\beta$, not to be confused with the parameters of the mass-size relationship shown in Tab. 1. The shape parameters are set to fixed, species-specific values. This principle is illustrated in Fig. 3. The plot displays the a-priori-assumed shapes of the particle size distribution of frozen and liquid hydrometeors. The retrieved horizontal and vertical scaling parametersand $N_0^*$, are used as units for the axes of the plot so that~~ which scales the particle concentration, are the two retrieved degrees of freedom of the PSD.

**Changes starting in line 274:**

 The cloud signal in the radiometer observations is the difference between the cloudy- and clear-sky brightness temperatures ($\Delta T_B$). The signal in the active observations is here defined as the maximum of the measured profile of radar reflectivity $\mathrm{dBZ_{max}}$. Figure 9 displays the contours of

**Changes starting in line 429:**

The results  indicate that the combined observations can  constrain the  size and concentration of particles in the cloud.

**Reviewer comment 3**

Line 100: A few more details on the Milbrant and Yau microphysics sheme that are relevant to this study would be helpful here. For example: What is the assumed shape(functional form) of the particle size distribution, and what are the prognostic variables(e.g., number concentration, mixing ratio)?

**Author response**

We followed the reviewers comment and added the requested information to the manuscript.

**Changes in manuscript**

> **Changes starting in line 99:**
>
> The GEM model uses  six types of hydrometeors to represent clouds and precipitation (Milbrandt and Yau, 2005): Two classes of liquid hydrometeors (rain and liquid cloud) and four of frozen hydrometeors (cloud ice, snow, hail and graupel). The particle size distribution (PSD) of each hydrometeor  class is described by a three-parameter gamma distribution. The prognostic parameters of the model are the slope and intercept parameters of the  PSD, which are derived from the predicted mixing ratios and number concentrations. The third parameter, which defines the shape of the  PSD, is set to a fixed, species-specific value. For each hydrometeor species a specific mass-size relationship is assumed.

**Reviewer comment 4**

Line 135: Does the ARTS radar solver also provide analytic Jacobians?

**Author response**

Yes, it does. A sentence will be added to the description of the forward model to clarify this.

**Changes in manuscript**

> **Changes starting in line 146:**
>
> Radar reflectivities are computed using ARTS' built-in single-scattering radar solver, which provides analytic Jacobians.

**Reviewer comment 5**

Line 187: particles should be particle

**Author response**

The sentence has been removed in the revised version of the manuscript since the information it conveyed was deemed irrelevant.

[Figure]

Figure 1.1: Illustration of retrieval quantities and their respective retrieval grids. Grey, dashed lines in the background display the vertical grid of the GEM model. Black, solid lines on the left side display the range bins of the radar observations. Filled markers represent the retrieval grids of each retrieval quantity for the combined, radar-only and passive-only configurations of the retrieval algorithm.

**Reviewer comment 6**

Line 198: Is Dm also only retrieved at these 10 points, or just N0* (and Dm retrieved in each radar range gate as in Grecu et al. 2016)?

**Author response**

$D_m$ is actually retrieved at the resolution of the GEM model scenes. Since questions about the retrieval grids were raised also by the other reviewers, we will add an illustration of the grids applied in the different retrieval configurations to the manuscript.

**Changes in manuscript**

The figure shown in Fig. 1.1 has been added to the manuscript to clarify which variables are retrieved at which resolutions.

**Reviewer comment 7**

7. Line 256: Actually, this is only one example of how the radar and radiometer measurements can be complementary. Even if the lines were parallel (and thus no information distinguishing size from concentration could be obtained), the radar still locates the cloud and describes its vertical structure. One can imagine a cloud of the same ice water path and particle size at two different heights having different brightness temperatures

due to changes in the water vapor absorption above the cloud having the radar information would provide increased information content about the ice water path in this case than the radiometer measurement alone.

**Author response**

It is certainly correct that, when a radar sensor is added to a passive observation system, one of the advantages will be the increased resolution. However, what we are interested in are the advantages that neither of the two instruments can provide on its own. If it was only about vertical resolution, then the radar alone would be the ideal observation system. In this sense, we do not consider the vertical resolution a synergy of the two sensors.

To make this clear, we will add an explanation of our definition of synergies between the active and passive observations to the section which discusses the complementary information content.

**Changes in manuscript**

**Changes starting in line 262:**

A fundamental question regarding the benefit of combining two remote sensing observations in a retrieval is to what extent the observations contain non-redundant information. The degree of non-redundancy in the combined observations is what we refer to here as complementary information content. We are thus interested in the information that cannot be provided by either of the instruments alone. The higher resolution achieved by adding radar observations to passive ones is therefore not considered as complementary information since the radar alone can provide the increased resolution.

**Reviewer comment 8**

Table 4: Why are the values for GemSnow and GemGraupel different than in Table 1?

**Author response**

This was by mistake and has been corrected in the revised version of the manuscript.

**Changes in manuscript**

Tables 1 and 4 have been corrected and extended in the revised version of the manuscript.

**Reviewer comment 9**

Figures 7 and 8: Im not sure why these are separate figures it seems like all panels could fit on one page.

**Author response**

Figures 7 and 8 have been combined into a single figure in the revised manuscript.

**Changes in manuscript**

Figures 7 and 8 have been combined into a single figure and now look as shown in Fig. 1.2.

**Reviewer comment 10**

Figure 10 is missing from the manuscript.

**Author response**

Figure 10 has been included in an Appendix to the revised manuscript with the rest of the analysis of the results from the second test scene.

**Changes in manuscript**

The figure shown in Fig.1.3 has been added in the appendix of the manuscript.

**Reviewer comment 11**

Line 374: recommend using represent instead of predict

**Author response**

The proposed change has been adopted in the revised version of the manuscript.

**Changes in manuscript**

> **Changes starting in line 365:**
>
> Since snow will have  a stronger impact on the observations, the retrieval in these regions  will likely tend to represent snow rather than ice, which leads to the low retrieved number concentrations.

**Reviewer comment 12**

Line 382: should be reference instead of references

**Author response**

This has been corrected in the updated version of the manuscript.

[Figure]

Figure 1.2: Retrieved IWC plotted against reference IWC for the tested retrieval configurations. Each row shows the retrieval results for the particle shape shown in the first panel. The following panels show the retrieval results for the passive-only (first column), the radar-only (second column) and the combined retrieval (third column). Markers are colored according to the prevailing hydrometeor type at the corresponding grid point in the test scene. Due to their sparsity, markers corresponding to graupel are drawn at twice the size of the other markers.

[Figure]

Figure 1.3: Scatter plots of the reference and retrieved IWC for the second test scene. The rows show the retrieval results for a given assumed ice particle model. The first column of each row displays a rendering of the particle model. The following rows display the results for the passive-only, the radar-only and the combined retrieval.

**Changes in manuscript**

**Changes starting in line 373:**

The radar-only retrieval does not exhibit any retrieval skill, hardly reproducing any

of the variation of the  reference values.

**Reviewer comment 13**

Line 414: How are the truncated PSDs (using GemSnow) represented in the forward simulations? Is total ice water content conserved? If so, how is it spread amongthe valid particle sizes  equally, or is the truncated mass allocated to the smallest size bin?

**Author response**

Total IWC is not conserved in the handling of PSDs. The point raised by the reviewer has been investigated by assessing the effect of the truncation on the water content of snow in the forward simulations. The results of the analysis are given in the figure below. As these results show, the effects of the truncation in the forward simulations are negligible.
However, when the GemSnow particle model is used in the retrieval it can introduce significant errors. For this reason as well as another reviewers' comment regarding the choice of tested particles, the selection of particles to be used in the retrieval will be changed for the revised manuscript and the GemSnow particle will be replaced by a habit mix which uses the GemSnow particle for large diameters.

[Figure]

Figure 1.4: Joint distribution of truncated and full snow water content (SWC) for the two test scenes.

**Reviewer comment 14**

Figure 16: The figure labels/captions aren't clear if they refer to total liquid water content/path or just the cloud liquid water/path.

**Author response**

We will clarify that the contours refer to liquid cloud water content in the revised version of the manuscript.

**Changes in manuscript**

To make clear what the labels refer to, they have been changed to liquid cloud water content (LCWC) in the figure. The new figure is shown in Fig. 1.5.

**Reviewer comment 15**

Line 518: Its interesting that the Plate Aggregate provides the most accurate re-trieval results, even though it isnt similar to the models used in the synthetic measurement simulations. Does the decreasing density with size better replicate the combina-tion of high-density GemCloudIce (which tends to be present in high concentrations atsmall sizes) and lower-density GemSnow (which tends to be dominant at larger sizes)?

**Author response**

Unfortunately, we cannot give a definitive answer to this question. As panel (a) in Fig. 15 shows, the density of the LargePlateAggregate habit is actually lower than that of snow for large particle sizes. Moreover, the scattering properties certainly also play a role here. From these results alone, we are therefore not able to postulate any direct causality between the particle density and the performance in the retrieval.

**Changes in manuscript**

To provide more definitive recommendations regarding the choice of the particle model, we have reconsidered the selection of models to test and added the figure shown in Fig. 1.6 to the manuscript. These results indicate that a potential explanation of the good performance of the Large Plate Aggregate is that its scattering properties are intermediate to those of GEM cloud ice and GEM snow.

[Figure]

Figure 1.5: Reference and retrieved CLWC and IWC. Panel (a) shows the reference and retrieved LWP for each profile. Panel (b) displays reference LWC contours drawn on top of the total hydrometeor content. Retrieval results for passive-only and combined retrieval are given in Panel (c) and (d).

[Figure]

Figure 1.6: Bulk mass backscattering efficiency $Q_b$ at 94.1 GHz (a) and mass attenuation coefficients $Q_e$ at frequencies 175.3 GHz (b), 314.2 GHz (c) and 657.3 GHz (d) for the particle models used in the simulated observations and the retrieval. Different colors show the bulk properties for different values of the $N_0^*$ parameter of the PSD.

**2 Comments from referee 2**

**2.1 Major points**

**Reviewer comment 1**

A major aspect of the study concerns the representation of the particle size distribution which is retrieved by two free parameters (different from the 2 moments of the atmospheric model GEM used to provide the test scenes) and the assumptions of the particle type. The difficulty of connecting atmospheric model output to single scattering properties (which is one of the fundamental assumptions) could be better explained. The motivation why the authors choose their approach and why they test certain settings need to be discussed in the beginning. Couldnt Tab. 1 and 4 be combined and better explained which is used for which purpose? Why is cloud ice the same and GEMsnow and GEMGraupel different in both?

**Author response**

We agree with the comment that the rather arbitrary choice of tested particles was a weak spot of the study. To improve this, the experiments have been repeated with a more principled selection of particles. The new selection is based on the particle properties described in Ekelund et al. (2020) and covers a broader range of mass-size relationships and scattering parameters. In particular, the GemSnow model has been removed from the selection of test particles because it does not cover small ice particle sizes.

**Changes in manuscript**

- A paragraph has been added to the description of the GEM test scenes which explains the particle models that have been developed to match the assumptions of the GEM microphysics scheme and that are used to simulate observations.

  **Changes starting in line 110:**

  In order to simulate observations from the GEM model scenes, the hydrometeor classes of the GEM microphysics scheme must be associated with particle shapes to define their radiometric properties. The ARTS single-scattering database, described in more detail below, contains particle models which were designed to be consistent with the mass-size relationships assumed in the GEM model. The particle shapes used to represent the GEM model's different hydrometeor types are listed together with their properties in Tab. 1.

- The following text has been added to the description of the retrieval implementation which discusses the difficulty of representing the complex mixture of different particles in the GEM model scenes with a single particle model as well as the new selection of particle models and habit mixes.

> **Changes starting in line 228:**
>
> A major difficulty for cloud retrievals is that the observations may not provide sufficient information to distinguish different hydrometeor species. Due to this ambiguity, frozen hydrometeors in the proposed retrieval algorithm are represented using only a single hydrometeor species. It is therefore necessary to find a suitable representation for frozen hydrometeors, which can capture the variability of the four frozen hydrometeor species in the GEM model and ideally also that of real ice hydrometeors.
>
> The differences between hydrometeor species in the test scenes are due to their different concentrations, sizes and shapes (c.f. Fig. 2). Since two parameters of the PSD of frozen hydrometeor species are retrieved, the retrieval is able to represent the characteristic number concentrations and particle sizes of different hydrometeor species. Variations in particle shape which correlate with particle size can be represented using a habit mix combining crystal shapes at small sizes with aggregates or rimed particles at larger sizes. This provides the retrieval with some flexibility to represent the different shapes present in the test scenes.
>
> Even with this configuration the simplified retrieval forward model
>
>  will not be able to represent every possible configuration of mixes of the four ice hydrometeor species in the GEM model. It thus remains unclear which particle shape should be used to best represent this mixture. We therefore choose a set consisting of multiple particle shapes and habit mixes for which we investigate the impact of the particle choice on the retrieval results. The selected particles are listed in Tab. 4. Three of them, GEM Cloud Ice, GEM Snow, and GEM Graupel, correspond to the shapes present in the GEM model scenes. The GEM Snow and Graupel habits were mixed with crystal shapes to ensure that they cover sizes down to around $10\,\mu$m. In addition to this, two of the habit mixes distributed with the ARTS SSDB, the Large Plate Aggregate and Large Column Aggregate standard habits, are included in the selection to increase the range of scattering properties it covers.

- The errors in the reported parameters of the mass-size relationship for the GEM Snow and GEM graupel particles have been corrected.

It was, however, not possible to combined Tab. 1 and Tab. 4 because they now convey

slightly different information.

**Reviewer comment 2**

Although different parameterizations of the hydrometeor types are used to study their effects, vertical changes (development of sedimenting particles) are not considered. Similar polarization effects are not mentioned in the discussion on particle shape. Otherwise the paper nicely discusses the different aspects but in the end I ammissing a clear message on the outcome of the test (choice of particle types). What is recommended for the future?

**Author response**

The first statement made by the reviewer is not fully correct. Since the retrieval can reduce the concentration of particles and increase their size it can modify the ratio of small and large particles and thus represent the effects of sedimentation on the PSD.

Vertical changes in particle shape, i.e. transition from single crystals to aggregates, are represented indirectly through the particle size. The particle models used here are taken from the standard habits of the ARTS SSDB described in Eriksson et al. (2018). Some of them combine pristine crystals at small particle sizes with aggregate shapes at larger sizes.

Polarization effects in the simulations were ignored for the simple reasons that the model scenes do not provide information on particle orientation or aspect ratios and that suitable scattering data for oriented particles has only recently been released (Brath et al., 2019). For the revised version the sensors are assumed to point at nadir, which justifies neglecting polarization effects. Nonetheless, particle orientation can still have an effect on the observations. We will state clearly in the revised manuscript that polarization effects will have non-negligible impact on the observations of the MWI and ICI sensors. We agree that the choice of the particle models was described and motivated poorly in the manuscript. To address this, we will extend the description of the chosen particle models and try to provide clearer conclusions regarding the outcome of our experiments.

**Changes in manuscript**

- The following paragraph has been added to Sect. 2.2 stating that for MWI and ICI polarization effects can not be neglected.

  **Changes starting in line 119:**

  Moreover, the incidence angles of the beams of ICI and MWI will be around 53° at the Earth's surface. This further complicates the radiative transfer modeling since it requires treating a more complex co-location geometry of the nadir-pointing radar and the passive instruments. At off-nadir viewing angles, polarization also needs to be taken into account, the effects of which can be several Kelvin at the typical viewing angles of microwave imagers

(Xie et al., 2015).

- The discussion of the tested particle shapes has been extended to derive clearer recommendations for the future.

    **Changes starting in line 492:**

     Only the combined retrieval was able to yield accurate IWC retrievals for both test scenes for suitable choices of the particle model. However, if an unsuitable particle shape is chosen, the induced errors may actually outweigh the benefits of the combined retrieval as is the case for the Large Column Aggregate and the GEM Cloud Ice shapes (Fig. 9). Judging from the particle properties displayed in Fig. 4, a likely explanation for  the good performance of the Large Plate Aggregate and the GEM Graupel particle is that their properties are intermediate to those of GEM Cloud Ice and GEM Snow, which are the dominating shapes in the test scenes. For the test scenes considered here, this means that accurate IWC retrievals can be achieved using only a single hydrometeor species with suitable scattering properties which are intermediate to snowflakes and heavily rimed particles. This is in agreement with Ekelund et al. (2020) who found the Large Plate Aggregate to yield good agreement with observations from the GPM Microwave Imager at $183.31 \pm 7$ GHz.

- The following paragraph has been added to the conclusions:

    **Changes starting in line 555:**

    Regarding the representation of hydrometeors in the retrieval, our results indicate that complex mixes of hydrometeors can be accurately represented using a single, suitable habit mix. In particular, our results indicate that a suitable habit should have scattering properties that are intermediate between strongly rimed and more snow-flake like particles (Fig. 4, 9).

**Reviewer comment 3**

Not only the two moments of the ice PSD but further variables are retrieved and their information content is nicely shown in Fig. 14. I am surprised that the information on moisture is so low although information along three water vapor lines is provided? This should at least in the upper atmosphere provide information? Is it due to the choice of relative humidity which mainly depends on temperature? I am also skeptical about the results of Fig. 16. Basically, there should be no liquid for temperatures colder than 40 deg C (freezing) but it even reference LWC goes up to 13 km? I would not support the statement on L568 where is the evidence? Similar L527 Liquid water estimation within mixed phase clouds is extremely difficult and if ICI and radar could really do that together this would be worth a separate paper. To better understand the information content, I suggest to plot the profiles of cumulative degrees of freedom for the different retrievals as this could help interprete where and how the synergy works.

**Author response**

As can be seen in Fig. 8 from Eriksson et al. (2019), the information content on water vapor from ICI alone are at most 4 degrees of freedom for clear-sky scenarios. Since in the retrieval also channels from MWI are included, the expected information content on water vapor should be somewhat higher. However, this is for clear-sky scenarios. In the presence of clouds, the information content will be significantly reduced.
Regarding the results of the retrieved cloud liquid water content (CLWC), Fig. 16 shows quite clearly an improvement, both in terms of CLWC and cloud liquid water path (CLWP), in the results of the combined retrieval compared to the passive-only retrieval. Yes, liquid clouds droplets are present at high altitudes in the first model scene but only below the 230 K isotherm. However, since this is the case only for the first scene, it does not seem relevant for the interpretation of Fig. 16.

**Changes in manuscript**

To provide a more detailed analysis of the information content regarding humidity and CLWC we followed the referee's suggestion and replaced the bar diagrams in the manuscript with a figure (Fig. 2.1 shown below) displaying the cumulative DFS for all profiles in the test scenes.

**Reviewer comment 4**

The manuscript is rather lengthy making it difficult for the reader to extract the majorpoints. I strongly suggest a) to move part of the analysis into an appendix (, b) remove double statement (see minor comments, also the LWC plot) and c) to remove figure caption like information (for example L92 or filled contours) from the text. The text must make sense without looking at the figure. Figure only support the statements made in the text. Lengthy descriptions such as "The plot shows.." need to be avoided.

[Figure]

Figure 2.1: Degrees of freedom for signal for all retrieval configurations and both test scenes obtained with the Large Plate Aggregate model. The colored areas in each plot represent the contribution to the cumulative degrees of freedom from each retrieval quantity. Results for the first and second test scene are displayed in the first and second row, respectively. The first, second and third panel in each row show the results for the passive-only, radar-only and the combined retrieval.

**Author response**

We have followed the referee's recommendations to make the manuscript more concise.

**Changes in manuscript**

- Descriptions of the figures which display the results in Sect. 3 have been removed.

- The analysis of the results from the second test scene have been moved to an appendix.

**2.2 Minor comments**

**Reviewer comment 1**

L39: Is sensitivity really the right word? Range resolution is the main advantage signal-to noise range depends on distance and hydrometeor distribution,

**Author response**

Following the reviewer's suggestion the sentence has been rewritten.

**Changes in manuscript**

Most of the introduction has been rewritten and the corresponding expression has been removed.

**Reviewer comment 2**

L48: MWI will also cover new spectral channels, e.g. 118 GHz

**Author response**

The 118 GHz spectral band is employed already today by the FY-3 satellites. Nevertheless, we mention the band now also in the introduction.

**Changes in manuscript**

> **Changes starting in line 63:**
>
> MWI will complement ICI's observations with measurements at traditional millimeter wavelengths as well as a spectral band around the 118 GHz oxygen line. The observations of MWI, which cover the frequency range from 19 GHz up to 183 GHz, will provide additional sensitivity to liquid and frozen precipitation as well as water vapor.

**Reviewer comment 3**

L62: high-resolution is always relative for a model. I would recommend avoiding this term and use Cloud resolving Model (CRM).

**Author response**

The proposed improvement has been adopted in the revised version of the manuscript.

**Changes in manuscript**

> **Changes starting in line 74:**
>
> For this, a combined, variational retrieval is developed and applied to simulated observations of scenes from a  cloud-resolving model (CRM).

**Reviewer comment 4**

L68: After you mention GPM (with scanning radar) it might be good to say that you are only looking at a nadir pointing radar (curtain). The swath center came bit as a surprise.

**Author response**

The radar type is now stated more explicitly in the introduction.

**Changes in manuscript**

> **Changes starting in line 75:**
>
> An airborne viewing geometry is assumed for the simulations with all sensors pointing at nadir and close-to overlapping antenna beams.

**Reviewer comment 5**

L70: There has been quite some literature about combining active and passive MW using a Bayesian framework which should be acknowledged, e.g. Grecu, M., & Olson,W. S. (2006), Johnson et al. (2012) , Munchak, S. J., & Kummerow

**Author response**

A paragraph listing previous work on synergistic retrievals using radar and passive radiometers at lower microwave frequencies has been added to the introduction.

**Changes in manuscript**

> **Changes starting in line 51:**
>
> Prominent examples of satellite missions that exploit both of these synergies are the
> the Tropical Rainfall Measuring Mission (TRMM, Kummerow et al. (1998); Grecu et al. (2004); Muncha
> ) and the Global Precipitation Measurement (GPM, Hou et al. (2014); Grecu et al. (2016); Kummerow e
> ) mission. Since the principal target of these missions are retrievals of liquid hydrometeors,
> they make use of sensors at comparably low microwave frequencies and hence provide
> only limited sensitivity to frozen hydrometeors.

**Reviewer comment 6**

L84: Test scenes have a grid resolution of 1 km horizontally. As this is not the true model resolution I would have recommended to coarse sample the model data (maybe every 5th data point) and include more diverse profiles instead. This might be especially interesting for the scatter plots.

**Author response**

The point raised by the reviewer here is certainly correct. However, the decision to restrict simulations to two test scenes was motivated primarily by the computational costs of performing the retrievals. The scatter plot in Fig. 10 (shown in Fig. 1.3 in this document), which was unfortunately missing from the manuscript , shows that the emergent structures are consistent for both test scenes. This indicates that the scenes cover sufficient profile variability to be statistically representative.

**Reviewer comment 7**

Motivation lacking: To perform RT simulations for each GEM profile the PSD needs to be diagnosed from the prognostic GEM variables, i.e. N and m..

**Author response**

The corresponding paragraph has been rewritten and the sentence removed.

**Reviewer comment 8**

L92:prognoses means forward in time - you mean diagnosed, calculated, determined.

**Author response**

The corresponding paragraph has been rewritten and the sentence removed.

**Reviewer comment 9**

L98: I find the term horizontal and vertical scaling strange why not saying PSD shape is similar but scaling in respect to diameter and number density. At least definethe term clearly the first time that you use it or define a short for it.

**Author response**

C.f. Comments from Referee 1 - General comment 2.

**Reviewer comment 10**

L103: model test be careful also at other instances that model can mean too manythings. Here I would say GEM test scenes.

**Author response**

Since much of the manuscript has been rewritten, this exact sentence is not present anymore. However, attention has been paid to the use of the word model and to ensure that its interpretation is unambiguous.

**Reviewer comment 11**

L119: Need to clearly say that polarization effects are neglected though these can beseveral Kelvin, e.g. Xie et al., 2015. You ignore this effect but even consider noise reduction.

**Author response**

C.f. Comments from Referee 2 - General comment 2.

**Reviewer comment 12**

L157-159: needs to be better motived, references?

**Author response**

To provide better motivation for the use of the $\chi^2$ statistic the text given below been added to the manuscript.

**Changes in manuscript**

> **Changes starting in line 159:**

 The quality of a retrieved state $\hat{\mathbf{x}}$ and corresponding simulated  observations $\hat{\mathbf{y}} = \mathbf{F}(\hat{\mathbf{x}})$  is assessed using the following diagnostic quantity:

$$\chi_y^2 = \delta\Delta\mathbf{y}^T\mathbf{S}_e^{-1}\delta\Delta\mathbf{y}, \tag{2.1}$$

 Here, $\Delta\mathbf{y} = \mathbf{y} - \hat{\boldsymbol{y}}$ is the difference between the fitted and true observations and $\mathbf{S}_e$ is the covariance matrix describing the measurement errors. The quantity $\chi_y^2$  corresponds to the sum of squared errors in the fitted observations weighted by the uncertainties in each channel or range bin. It should be noted that the quantity has no meaningful interpretation in terms of $\chi^2$  -statistic for the errors in the fitted observations since they will neither be independent (c.f. Chapter 12 in Rodgers (2000))

$$\mathbf{A} = (\mathbf{K}^T\mathbf{S}_e^{-1}\mathbf{K} + \mathbf{S}_a^{-1})^{-1}\mathbf{K}^T\mathbf{S}_e^{-1}\mathbf{K}.$$

nor Gaussian due to the presence of forward model error. The value is therefore used here solely as a heuristic to quantify the goodness of the fit to the true observations.

**Reviewer comment 13**

L172: I doubt that the model has constant vertical resolution. It will be better close to the surface and worse aloft. This should be mentioned than GEM is introduced.

**Author response**

As suggested by the reviewer, this is mentioned in the revised manuscript when the GEM test scenes are introduced. Moreover, a sketch will be added to the manuscript which displays the GEM model grid and the grids of all retrieval quantities for the retrievals.

**Changes in manuscript**

The following text has been added to the description of the model scenes.

**Changes starting in line 92:**

> The vertical resolution of the model scenes varies between 250 and 500 m below an altitude of 18 km and decreases steadily above that.

In addition to this, the figure shown in Fig. 1.1 has been added in Sect. 2.3.2 of the revised manuscript.

**Reviewer comment 14**

L 174: for all hydrometeor species of the model? It would be helpful to first introduce all retrieval quantities  I was missing a motivation for the paragraph around L195. How do you define the freezing level (and later the troposphere)? How do they vary in both test scenes? The model also likely has supercooled liquid water above the freezing layer  how is this treated?

**Author response**

For the simulated observations, supercooled liquid is treated in just the same way as liquid water below the freezing level. As described in the paragraph around L. 211 (old manuscript version), cloud liquid water in the retrieval is treated as purely absorbing and simulated using a parametrized absorption model. Moreover, it is restricted to temperatures of 230 K and up.

**Changes in manuscript**

Fig. 1.1 has been added to the manuscript, which displays all retrieval variables as well as the freezing level and the tropopause. Moreover, the following text explaining the definitions of freezing level and tropopause has been added to the manuscript:

> **Changes starting in line 193:**
>
> As additional constraints, the retrieval of frozen hydrometeors is restricted to the region between the freezing  level, here defined simply as the 273.15 K-isotherm, and the approximate altitude of the tropopause. The altitude of the tropopause is approximated as the first grid point at which the lapse rate is negative and temperature below 220 K. The retrieval of rain hydrometeors is restricted to below the freezing   level.

**Reviewer comment 15**

L 198: Vertical resolution of retrieval grid: Why 4 points? The freezing layer must be very different for both cased. Maybe a sketch would be helpful as later on lines 230 the different vertical resolutions for other variables is discussed?

**Author response**

We have revised the retrieval implementation and now use fixed retrieval grids with a resolution of 2 km for the $N_0^*$ parameters. Reducing the resolution of the retrieval grids for the $N_0^*$ parameters was found to aid the convergence of the retrieval.

**Changes in manuscript**

The sketch requested by the reviewer has been added to the manuscript (c.f. comment above).

**2.2.1 Reviewer comment 16**

L281: How do I know that Large Plate is the best performing model? Which parameter,plot, table does show that?

**Author response**

This sentence has been removed from the revised version of the manuscript.

**Reviewer comment 17**

L283-L307: Can be shortened significantly

**Author response**

The proposed change has been adopted in the revised version of the manuscript.

**Changes in manuscript**

> **Changes starting in line 304:**
>
> The $\chi_y^2$ values of the three retrieval configurations, displayed in Panel (a)of the figure displays the $\chi_y^2$ value (normalized by the dimension of the measurement space) for each profile in the scene. A high value of $\chi_y^2$ indicates that the retrieved state is not consistent with the input observations. The $\chi_y^2$ value for , give an indication of how well the retrievals are able to fit the observations. For the radar-only retrievalis remarkably low throughout most , the values are much smaller than 1 for most parts of the scene. This may indicate that the retrieval is insufficiently regularized, allowing it to fit the noise in the observations. The , while for the passive-only and combined retrieval , on the contrary, have a normalized $\chi_y^2$ value around 1 over most of the scene. Since the presented values are normalized, the value 1 corresponds to they are around the expected value of the approximated chi-square distribution of $\chi_y^2$. All three retrievals exhibit a region of elevated $\chi_y^2$ values near the core of the convective system. In particular the high values of 1. This indicates that the radar-only retrieval overfits the observations, while the passive-only and combined

retrievals

 achieve the expected fit. The exception is the  region around $3° N$, where the cloud is particularly thick and consists of a mix of different hydrometeor types. Here, especially the passive-only retrieval has problems fitting the observations.

In terms of IWP, all methods provide fairly good estimates of the reference values  with the combined retrieval consistently yielding the smallest deviations. Larger differences between the methods are observed when comparing the retrieval

 results in terms of IWC. While the vertical structure of the cloud

 is captured only very roughly by the passive retrieval, it is better resolved by the radar-only  and the combined retrieval. On closer inspection, however, it becomes evident that the radar-only retrieval  deviates systematically from the reference IWC in specific regions of the cloud, such as for example the upper part of the cloud

 between $0°N$ and $2°N$. These deviations are corrected in the results from the combined retrieval, although certain retrieval artifacts are visible.

**Reviewer comment 18**

L332: There can I see that? Give figure?

**Author response**

The sentence has been removed from the revised version of the manuscript.

**Reviewer comment 19**

L325: The two paragraph here give similar information -> streamline

**Author response**

The proposed change have be adopted in the revised version of the manuscript.

**Changes in manuscript**

> **Changes starting in line 320:**
>
> ~~Similar to the radar-only retrieval, the results of the combined retrieval are located close to the diagonal. But the clusters observed in the radar-only results are to large extent merged in the combined results. Moreover, except for the results obtained with the GemCloudIce particle shape, the two clusters move in closer towards the diagonal. The combined retrieval thus improves the IWC retrieval for the specific hydrometeor species in the scene.~~ is biased for specific hydrometeor classes. In the combined and even the passive-only results, this effect is weaker and the clusters are generally moved towards the diagonal. For graupel, all retrievals perform badly but this is likely due to it being present only in the core of the convective system where the signals from all sensors can be expected to be saturated.
>  Comparing the results  for different particle models, a clear dependency is evident in the passive-only and the combined results while the radar-only retrieval

**Reviewer comment 20**

L333-344: I would put this to the appendix

**Author response**

Following the reviewer's advice, the analysis of the second test scene has been moved to the appendix.

**Reviewer comment 21**

L444: Here it needs to be made clearer how this goes beyond what GPM is doing.

**Author response**

To clarify how our work goes beyond what GPM does, a paragraph detailing this has been added to the discussion.

**Changes in manuscript**

> **Changes starting in line 421:**
>
> The novelty of this work for lies, in part, in the application of ICI's sub-millimeter channels, which sets it apart from the combined retrievals developed for the TRMM and GPM missions. Moreover, the development of a fully consistent variational retrieval in which all retrieval quantities are retrieved simultaneously using the observations from all sensors is also novel. This allows comparison of the combined retrieval to equivalent radar-only and passive-only configurations and therefore a direct analysis of the synergies between the active and passive observations.

**Reviewer comment 22**

L495: does not say much about the general validity of the assumption. Here you should dig in a bit more. What is the role of a priori and covariances?

**Author response**

Following the suggestion of the reviewer we will the discussion of the a priori assumptions has been extended.

**Changes in manuscript**

C.f. the first change listed for Comments from Referee 1 - General comment 1.

**Reviewer comment 23**

L560: Rethink the bullet structure. 2. Is not an independent result. For each result refer to the part of the manuscript where you can clearly see that. Especially result 3 should be detailed  how do ICI channel advance the currently available data?

**Author response**

The bullet points have been removed in the revised manuscript and replaced by a text which presents the conclusion in a logically coherent way.

**Changes in manuscript**

> **Changes starting in line 540:**

The main  conclusion from this work is that the combination of radar and sub-millimeter radiometer observations can, to some extent, constrain both the size and number concentration of frozen hydrometeors
(Fig. 5). The increased sensitivity of the combined retrieval to the microphysical properties of hydrometeors helps to improve the accuracy of IWC retrievals and avoid systematic errors observed in an equivalent radar-only retrieval (Fig. 8, 9). Moreover, the combined retrieval showed clear sensitivity to particle number concentrations and was able reproduce their vertical structure in regions where the cloud composition is homogeneous (Fig. 10, 11).
The results  particularly highlight the ~~complementarity of the active and passive observations : Although the radar provides observations at high vertical resolution, they contain insufficient information on the microphysical properties of hydrometeors . The passive-only observations, on the contrary, have low vertical resolution, but are more sensitive to cloud microphysics allowing a potentially more accurate IWP retrieval than what can be obtained from the radar alone. A synergistic retrieval using both types of observations allows combining the high vertical resolution of the radar with the sensitivity to cloud microphysics~~ importance of sub-millimeter observations for combined retrievals of frozen hydrometeors. While observations at currently available microwave frequencies provide information complementary to that from a radar-only for thick clouds with very large particles ($D_m > 800 \ \mu$m, IWC $> 10^{-4}$ kg m$^{-3}$), frequencies above 200 GHz provide additional information on cloud microphysics (Fig. 5) at smaller particles sizes and water content ($D_m > 200 \ \mu$m, IWC $> 10^{-5}$ kg m$^{-3}$).

**Reviewer comment 24**

Fig. 3: Is it really worth having the slightly different size distribution shapes for frozen and liquid? Isnt there a stronger difference between different frozen hydrometeors

**Author response**

This is certainly true but in most clouds ice and rain can be distinguished fairly well a priori, which simplifies treating them as different species using different PSD shapes. Distinguishing between different frozen hydrometeors is difficult to do a priori and using multiple species of hydrometeors in the retrieval was found to cause ambiguities which the retrieval is not able to resolve.

**Reviewer comment 25**

Figures 7 and 8: Im not sure why these are separate figures it seems like allpanels could fit on one page.

**Author response**

Figures 7 and 8 will be combined into a single figure in the revised manuscript.

**Changes in manuscript**

C.f. Comments from Referee 1 - Specific comment 9.

**Reviewer comment 26**

Fig. 4 and also in text: cloud signal say that this is dTB.

**Author response**

Following the reviewers recommendation, the passive cloud signal will be referred to in the text as $\Delta T_B$ and the radar signal as $dBZ_{max}$.

**Changes in manuscript**

Fig. 5 and its caption have been changed as shown in Fig. 2.2. In addition to this, the paragraph describing the results has been changed as follows.

> **Changes starting in line 272:**
>
>
>
> $$m = \frac{\pi \rho}{4^4} N_0^* D_m^4.$$
>
>  The cloud signal in the radiometer observations is the difference between the cloudy- and clear-sky brightness temperatures ($\Delta T_B$). The signal in the active observations is here defined as the maximum of the measured profile of radar reflectivity $dBZ_{max}$. Figure 4 displays the contours of $\Delta T_B$ and $dBZ_{max}$ with respect to $D_m$ and the cloud's water content, which

[Figure]

Figure 2.2: Simulated observations of a homogeneous, 5 km thick cloud layer with varying water content $m$ and mass-weighted mean diameter $D_m$. The panels display the maximum radar reflectivity in dBZ (dBZ$_{\text{max}}$) overlaid onto the cloud signal ($\Delta T_B$) measured by selected radiometer channels of the MWI (first row) and ICI radiometers (second row).

is proportional to $N_0^*$:

$$\text{WC} = \frac{\pi \rho}{4^4} N_0^* D_m^4, \tag{2.2}$$

with $\rho$ the density of ice.

**Reviewer comment 27**

Fig. 5: Can you add freezing layer height?

**Author response**

Freezing level height has been added to Fig. 5.

[Figure]

Figure 2.3: Total water content (WC) and simulated observations for the first test scene. Panel (a) displays the total water content in the scene, i.e. the sum of the water content of all hydrometeor species of the GEM model. Panel (b) shows the simulated radar reflectivities. Panel (c) displays the simulated brightness temperatures for a selection of channels of the MWI and ICI radiometers.

**Changes in manuscript**

The freezing level has been added to Fig. 5, which now looks as shown in Fig. 2.3.

**Reviewer comment 28**

Fig. 6: It would be nice to see the absolute values of IWP somewhere. Maybe you could add another time series with IWP as the sum of the different components such that the reader can see where the different categories (cloud, graupel, snow and hail) contribute most?

**Author response**

We will add absolute IWP and its decomposition into different hydrometeor classes to Fig. 6.

**Changes in manuscript**

Total IWP and its decomposition into contributions from different hydrometeor classes have been added to panel (b) of Fig. 6, which now looks as shown in Fig. 2.4.

[Figure]

Figure 2.4: Results of the ice hydrometeor retrieval for the first test scene using the Large Plage Aggregate particle shape. Panel (a) displays the value of the $\chi_y^2$ diagnostic normalized by the dimension of the measurement space of the corresponding retrieval. Panel (b) displays retrieved IWP in dB relative to the reference IWP. Reference IWP and the contributions from different hydrometeor classes are displayed by the filled areas in the background. Panel (c) shows the reference IWC from the model scene. Panel (d), (e) and (f) display the retrieval results for the passive-only, radar-only and combined retrieval, respectively.

**Reviewer comment 29**

Fig. 7: I think it is retrieved vs. truth. The word following is not really exact. Why not put 7 and 8 together?

**Author response**

Fig. 7 and 8 have been merged and the caption has been corrected.

**Changes in manuscript**

C.f. Comments from Referee 1 - Reviewer comment 9.

**Reviewer comment 30**

Fig. 9: Could go to the appendix

**Author response**

Fig. 9 has been moved to the appendix.

**Reviewer comment 31**

Fig. 10 I only see the caption???

**Author response**

C.f. Comments from Referee 1 - Reviewer comment 10.

**Reviewer comment 32**

Tab. 1. Assumed particle model information for each hydrometeor class given by GEM-model. In fact it could be good to combine it

**Author response**

In order to give a better overview of the particle models that are used in the retrieval Tab. 4 will be extended in the revised version of the manuscript. This, however, would make merging Tab. 1 and 4 slightly confusing so we decided against the reviewers recommendation.

**Reviewer comment 33**

Tab.3 : I would recommend to add a first column with a spelled out name

**Author response**

We have added the column to the revised version of the manuscript.

**Changes in manuscript**

The proposed column has been added to the table which now looks as is shown in Tab. 2.1.

Table 2.1: A priori uncertainties and correlation lengths used in the retrieval.

| Retrieval target | | Combined / Radar-only | | Passive-only | |
|---|---|---|---|---|---|
| Name | Retrieved quantity | $\sigma_q$ | $l_q$ [km] | $\sigma_q$ | $l_q$ [km] |
| Ice, $N_0^*$ | $\log_{10}(N_{0,\mathrm{Ice}}^*)$ | 2 | 2 | 2 | 5 |
| Ice, $D_m$ | Ice $D_{m,\mathrm{Ice}}$ | 300 $\mu$m | 2 | 300 $\mu$m | 5 |
| Rain, $N_0^*$ | $\log_{10}(\mathrm{Rain}\ N_0^*)$ | 2 | 2 | 2 | 5 |
| Rain, $D_m$ | $D_{m,\mathrm{Rain}}$ | 300 $\mu$m | 2 | 300 $\mu$m | 5 |
| Relative humidity (RH) | $\mathrm{arctanh}(\frac{2 \cdot \mathrm{RH}}{1.2} - 1.0)$ | 0.5* | 2* | 0.5 | 2 |
| Cloud liquid water content (CLWC) | $\log_{10}(\mathrm{CLWC})$ | 1* | 2* | 1 | 2 |

*: Not retrieved in radar-only retrieval

**2.3 Grammar, typos and reformulations**

The authors would like to thank the reviewer for the additional comments, all of which will be incorporated into the revised manuscript.

**3 Comments from referee 3**

**3.1 Major comments**

**Reviewer comment 1**

The paper is presented as an application for ICI in combination with a Cloudsat like configuration but it is not clear to me what geometry of observations the authors are thinking about. They state "As mentioned above, the same incidence angle as for the passive radiometers is assumed also for the radar. In practice, this could be achieved by remapping the radar observations to the lines of sights of the passive beam". Are they thinking about a scanning W-band radar? or at a off-nadir pointing radar? If the former is true then they should discuss what is a realistic technological solution (and what are the consequences in terms of sensitivity) and the authors should refer to state of the art scanning W-band radar concepts (there is none at the moment!); if the latter is true they should discuss what are the consequences of such a selection (e.g. forground clutter) and they need to convince me that what we could gain from such a configuration compensate from the loss of information introduced by pointing in such a slanted direction. There should be a certain degree of realism in what we are trying to simulate, especially if this was part of an ESA study.

**Author response:**

The reviewer raises a very relevant point with his comment. To address this, we have changed our simulation setup to simulate perfectly co-located observations at nadir. Realistic modeling of a space-borne viewing geometry (at least in a variational retrieval) is currently not feasible due to the computational complexity. We still deem this sufficient for the scope of the study, i.e. studying the fundamental synergies between active and passive observations. In addition to this, we follow the recommendation made in the second comment and will pitch the application more towards air borne observations.

**Changes in manuscript**

All calculations have been repeated for the assumed airborne viewing geometry and the manuscript has been adapted accordingly.

**Reviewer comment 2**

"The beams of all three sensors are modeled as perfectly coincident pencil beams". Again this is quite an assumption. Non uniform beam filling will play a key factor. This is one

of the many simplifications (no polarization, no multiple scattering,1D, ...) that needs to be clearly listed at the beginning of Sect.2.2.1 (some appear only at page 27). For this reason I would actually pitch more towards an airborne configuration where these simplification indeed can be realistically assumed or of a radar with a radiometric mode (where you can actually match footprints). Otherwise the (not massive) gain of having a radar-radiometer combination that you show later on can be completely washed out by the errors introduced to these assumptions. I imagine that you may also have airborne data where to test how realistic your forward model is.

**Author response**

As mentioned above, we will follow the reviewer's suggestion to pitch the application of the combined retrievals more towards combined retrievals. We also state these limitations more clear in Sect. 2.2.1 and discuss their implications more thoroughly.

**Changes in manuscript**

> **Changes starting in line 116:**
>
>  An airborne sensor configuration is simulated to test the retrieval. The beams of all three sensors are  assumed to point at nadir and to be perfectly coincident pencil beams. ~~Secondly, a synthetic observation is generated for each vertical profile from the model scenes by simulating a one-dimensional, plane-parallel atmosphere, the properties of which are taken from the corresponding model profile. It follows from these modeling decisions that the atmosphere is assumed to be homogeneous across the beams of the active and passive sensors and that they all sense the same atmospheric volume. This is certainly notand will incur a forward modeling error that is not accounted for in this study. Since the focus of this study are the fundamental synergies between the active and passive observations, quantifying the impact of beam width and inhomogeneity is left for future investigation~~ from ICI and MWI. Moreover, the incidence angles of the beams of ICI and MWI will be around 53° at the Earth's surface. This further complicates the radiative transfer modeling since it requires treating a more complex co-location geometry of the nadir-pointing radar and the passive instruments. At off-nadir viewing angles, polarization also needs to be taken into account, the effects of which can be several Kelvin at the typical viewing angles of microwave imagers (Xie et al., 2015).

**Reviewer comment 3**

Fig 2: these PSDs look very weird to me. Why do they have the plateau at small sizes? y-axis units are obviously wrong unless you are renormalizing by some mass (but it is

not explained).

**Author response**

The reviewer is of course right, the units on the axis of the plots were indeed wrong and will be corrected in the revised manuscript. Otherwise, the PSDs correspond to the modified-gamma functions that are assumed in the Milbrandt and Yau (2005) microphysics scheme.

**Changes in manuscript**

The y-units of the figure have been corrected which now looks as shown in Fig. 3.1.

[Figure]

Figure 3.1: Realizations of particle size distributions from the test scenes used in this study. The particle number concentration is plotted with respect to the volume-equivalent diameter $D_{\mathrm{eq}}$. Shown are the PSDs corresponding to 100 randomly chosen grid points with a water content higher than $10^{-6}$ kg m$^{-3}$. Line color encodes the corresponding water content. Inlets display visualizations of the particle shape assumed for each hydrometeor species.

**Reviewer comment 4**

Fig 3: sorry I do not follow what is this (what is the y-axis?), and why this plot is meaningful.

**Author response**

We have removed this plot from the revised version of the manuscript.

**Reviewer comment 5**

Eq.6: Clearly with values lower than 230 K it does not make any sense (negative RH, or large than1.1???)

**Author response**

We would like to thank the author to point out this inconsistency, as there are indeed two mistakes in Eq. 6. The right equation should be

$$\phi(t) = \begin{cases} 0.7, & 270 \text{ K} < t \\ 0.7 + 0.01 \cdot (t - 270), & 220 < t \leq 270 \text{ K} \\ 0.2, & t < 220 \end{cases} \qquad (3.1)$$

This will of course be corrected in the updated version of the manuscript.

**Changes in manuscript**

**Changes starting in line 202:**

$$\underline{\phi\text{RH}}(t) = \begin{cases} 0.7 & , 270 \text{ K} < t \\ 0.7 - 0.01 \cdot (270 - t) & , 220 < t \leq 270 \text{ K} \\ 0.2 & , t < 220\text{K} \end{cases} \qquad (3.2)$$

**Reviewer comment 6**

Line 210; this means that the vertical resolution changes with the surface temperature, really weird choice.

**Author response**

We agree with the reviewer that the chosen retrieval grids may not have been optimal. We will change them to fixed-resolution grids for the revised manuscript.

**Changes in manuscript**

Computations have been repeated with fix-resolution grids and the manuscript has been rewritten accordingly.

**Reviewer comment 7**

fIG4 : not clear to me why the scattering depression is not increasing at higher frequencies. I would expect that the optical thickness would drastically increase increasing frequency. Is this due to very large asymmetry parameters then? But this is not what I do see in Fig.5 (though Fig4 is of course a very idealized case) If this is the case then results will be very dependent on particle habits (which may introduce additional uncertainties in the retrieval)

**Author response**

It is correct that extinction increases rapidly with frequency, but the final scattering depression depends also on other factors. One consideration is the background absorption due to gases. A higher gas absorption decreases the effect of scattering, and this effect generally increases with frequency. It is correct that also the asymmetry parameter needs to be considered, which increases with frequency. A higher asymmetry parameter gives a lower depression for a given cloud optical depth, see Fig. 5 of Eriksson et al. (2015).

It can be hard to judge the scattering depression in a figure like Fig. 5, as the clear-sky values differ between the channels. In the version found below, extracted scattering depressions are shown in the second panel. For high-clouds with moderate cloud optical depth, the scattering depression increases monotonously with frequency, while in the most dense cloud region (around lat 2.7) this is not the case for the reasons discussed above.

[Figure]

Figure 3.2: Simulated brightness temperatures (Panel (a)) and cloud signal depressions computed for selected channels of the MWI and ICI radiometers for the first test scene.

**Reviewer comment 8**

8) Line 275: not clear what you mean, in Tab.4 there are 6.

**Author response**

What was meant here is that different ice shapes are tested for the single frozen hydrometeor species which is used in the retrieval. Tab. 4 lists the different shape models that were investigated.

Since the section describing the selection of particle models will be rewritten, this sentence will be reformulated to make it clearer.

**Changes in manuscript**

C.f. Comments from Referee 2 - General comment 1

**Reviewer comment 9**

9) extends below the sensitivity limit of the passive-only observations around 10-5 kg m-3 : very sloppy sentence. Passive microwave radiometer are sensitive to integrated contents!

**Author response**

As response to a comment from another reviewer the corresponding paragraph will be rewritten and this sentence will be removed.

**Changes in manuscript**

> **Changes starting in line 310:**
>
>

**Reviewer comment 10**

Fig 6d: this retrieval looks really weird. Where are all the stripes coming from? Certainly this does not look like acloud, or? What kind of constraint have you imposed on the cloud top?

**Author response**

It is true that the passive only retrieval does not perform well in terms of the vertical structure of IWC. The reason for this is that the passive observations alone do not provide much information on the vertical distribution of ice. To correct for this, further regularization would be necessary which is not applied here in order to keep the comparison to the other retrieval methods fair. All of this is discussed in the discussion section of the manuscript.

**Reviewer comment 11**

"In general, the radar-only results exhibit only very weak dependency on the particle model, making the results for different particle shapes virtually indistinguishable." Again another dangerous sentence. We know (unfortunately) that this is not true (otherwise our ice problems would be sorted). Here my guess is that you have not properly explored the backscattering variability (particularly looking at the different degree of riming). It is notclear to me whether there is enough variability in your ARTS database, I guess

you are more focused at ice particles (including aggregates) but you are not considering really rimed particles. Regions where graupel is present should be avoided from the discussion of the radar-only retrieval for the simple reason that in those regions attenuation correction and multiple scattering effects make the problem very tricky. I guess that the radiometer as well is in serious trouble when entering those areas. Again, I would not start tackling regions the observation system is not tailored for.

**Author response**

As mentioned above, we will revise the particle habits used in the retrieval, but we expect that particle shape will continue to have a smaller impact on our radar-only retrieval. What the results shown in scatter plot in Fig. 7 and 8 indicate is that the uncertainty which can be attributed to the particle size distribution (PSD) is larger than that introduced by the assumed particle shape. However, it is difficult in general to draw a clear line between particle shape and PSD. This is especially true if particle size is described by $D$max, and the PSD is defined accordingly. In this case, IWC of a given PSD will depend on the particle's effective density, and e.g. degree of riming becomes critical. Accordingly, to what extent retrieval errors are due to shape or PSD, depend partly on definitions.

The ARTS single scattering database does include several types of rimed particles. Two of them are the GEM Graupel and GEM Hail models which are used in the simulation of the synthetic observations. For the retrieval, however, it is true that we did not include rimed particles in the tested particle models but this will be changed for the revised version of the manuscript.

Both the forward simulations and the retrieval handle attenuation consistently. We therefore think it is worth considering even regions where graupel is present as this allows us to assess the uncertainties caused by not having a realistic representation of rimed particles in the retrieval.

It is certainly correct that for space-borne observations multiple scattering needs to be considered and this will add complexity to the retrieval. Here, however, we can avoid this extra complexity as we use simulated observations which do not include multiple scattering.

**Reviewer comment 12**

Fig.10 is missing!!!

**Author response**

Fig. 10 was unfortunately missing from the manuscript. The figure will be included in the appendix of the revised version together with the analysis of the second test scene.

**Changes in manuscript**

C.f. Comments from Referee 1 - Specific comment 10.

**Reviewer comment 13**

Since the calculation of the AVK involves the forward model Jacobian, this effectmust be related to the non-linearity of the forward model well I would avoid such veryspeculative statements.

**Author response**

Following the suggestion of the reviewer, the sentence will be removed from the manuscript.

**Changes in manuscript**

The sentence has been removed from the manuscript.

**Reviewer comment 14**

You need to be very careful how you present the results in Fig. 14. The conclusions that I can draw is the following: a CloudSat like radar is pro-viding much more information than the ICI+MWI radiometers when characterizing ice particles (really the radiometer is providing some additional water vapour information). As a result we should invest in the former and not the latter. While I may agree with the previous statement and strongly support a CloudSat-like radar on an operational mission my feeling is that you are pitching your radiometer system at the wrong kind of scenes (I already see an improvement going from the first to the second scene). I would have selected completely different scenes (including high latitude clouds with mixed phase). It is to me an overkill to try to retrieve D_M of rain for these scenes from your PMW radiometer suite of sensor. If you have any skill in warm rain you should properly prove it

**Author response**

Our interest in this study is neither arguing for one nor the other observation system. The question that we want to address is whether combined observations have extra value compared to separate observations. Such combined observations could be achieved by performing joint flights with the aircraft carrying the ISMAR sub-millimeter radiometer and another one carrying a radar, by flying a cloud radar in constellation to Metop-SG, or by adding a sub-millimeter radiometer to the platform carrying some future cloud radar. We consider it out of the scope of this study to judge the cost effectiveness of either of these solutions.
As the referee clearly favours radars, we would like to balance this by mentioning that passive instruments have an additional strength in their much higher areal coverage. The swath of ICI and MWI is about three orders of magnitude broader than that of CloudSat and EarthCARE.
Although a cloud radar certainly provides more information on frozen hydrometeors than ICI, our results clearly show that also radar observations alone are insufficient to accurately determine the microphysical properties of ice hydrometeors (Fig. 4, 7).

The passive adds information on the microphysics of the clouds to the radar (note the significant increase in information content on $N_0^*$ in Fig. 14) which helps to reduce retrieval uncertainties (Fig. 11). Although it is not clear whether these improvements carry over to space-borne observations, our results clearly show this as a synergy between the passive and active observations (esp. Fig. 4, 11, 14).

The cloud scenes used in the manuscript were selected with the aim of providing a representative sampling of the type of clouds present in the two model scenes that were available for the study. We did not want to cherry pick scenes were the retrieval works well to provide a more realistic assessment of the retrieval.

Rain must be handled in the retrieval due to its effect on the passive radiances. However, we never claim that we have any skill in retrieving warm rain and so we do not agree that we are required to prove to have it.

**Reviewer comment 15**

LWP and Fig.16. I have a serious problem here. The cloud I see on the right is a liquid cloud. So how it is possible that your radiometer is doing sobadly in the LWP retrieval and why the combined is so much better? I guess this must go back to understanding surface emissivity and integrated water vapour (maybe somecomments there should be made to explain what kind of surface/IWP we are dealingwith). You have not included radar path integrated attenuation in your retrieval (like istypically done in radar retrievals) but this could of course help in this case.

**Author response**

The cloud in the right of the scene is a mixed-phase cloud. There are several explanations for why the retrieval does not work well here: First of all, our observations setup does not make use of the channels around 23 GHz, which are typically used for retrieving LWP. And also here the performance of the passive-only retrieval suffers from the lack of a priori information on the vertical position of the cloud. Since liquid water at higher altitudes has a stronger impact on the observations, the retrieval puts too little cloud water too high in the atmosphere because of its inability to locate it properly. This is discussed in Sect. 4.2.3 of the manuscript.

**Reviewer comment 16**

I do not think that for OE to work The forward model must be linear as stated at line 544.

**Author response**

The OEM can of course be applied to non-linear problems but a complication that arises is that it can get stuck in secondary minima. The sentence will be corrected in the revised version of the manuscript.

**Changes in manuscript**

The section discussing limitations of the OEM has been removed from the manuscript since it was deemed to be of minor importance.

**Reviewer comment 17**

17)Sect.4 and 5: a lot of waffling here (e.g. the three bullet conclusion, you need to be much more quantitative and linked to what you have proved; the three statements are something I could have formulated on my own without making any simulation). Again the conclusions must be related to the cloud regime you are considering (and cannot be valid for all!)

**Author response**

One of the main advantages that we see in the combined retrieval is that it actually works for a wide range of different cloud regimes. If the cloud regime was known a priori, good results can probably be achieved using only a radar and suitable a priori assumptions. In general, however, this is not the case, which leads to the uncertainties that we currently have in the observational record for IWP and IWC.
For the revised manuscript, we will rewrite the conclusion and parts of the discussion to make it more concise and the point mentioned above more clear.

**Changes in manuscript**

C.f. Comments from Referee 2 - Specific comment 23

**3.2 Minor comments**

**Reviewer comment 1**

I would avoid the use of ice mass density and use ice water content

**Author response**

The proposed changes will be adopted in the revised version of the manuscript.

**Changes in manuscript**

Water content is now used consistently across the manuscript to refer to the mass density of hydrometeors.

**Reviewer comment 2**

Table 2: it would be good to see footprints as well

**Author response**

Since in the revised manuscript an airborne viewing geometry will be considered the footprint sizes of MWI and ICI are not relevant anymore.

**Reviewer comment 3**

Line 130: dBZ are the wrong units for a std of a reflectivity!

**Author response**

We are unsure what the reviewer is referring to here since quantifying uncertainty in the radar observations in dBZ seems to fairly common. This is for example how it is handled in the DARDAR cloud (Delanoë and Hogan, 2010) product as well as in the study by Jiang et al. (2019).

**Reviewer comment 4**

Line 180: The remaining shape of each PSD is described by the shape parameters alpha and beta, not to be confused with the parameters of themass-size relationship shown in Tab. 1.; very confusing. Why are you using the same letters????

**Author response**

We used the same letters to be consistent with the definition and used in Delanoë et al. (2014) and Cazenave et al. (2019). However, since the explicit values of the $\alpha$ and $\beta$ parameters are probably of little interest for the average reader, we will simply refer to Cazenave et al. (2019) and not name the parameters explicitly.

**Changes in manuscript**

> **Changes starting in line 182:**
>
> ~~The retrieval computes vertical profiles of the two scaling parameters $D_m$ and $N_0^*$ for each of the two hydrometeor species. The remaining shape of each PSD is described by the shape parameters $\alpha$ and $\beta$, not to be confused with the parameters of the mass-size relationship shown in Tab. 1. The shape parameters are set to fixed, species-specific values. This principle is illustrated in Fig. **??**. The plot displays the a-priori-assumed shapes of the particle size distribution of frozen and liquid hydrometeors. The retrieved horizontal and vertical scaling parametersare used as units for the axes of the plot so thatPSD becomes independent of the retrieved mass density and number concentration. For frozen hydrometeors, the values of the shape parameters $\alpha$ and $\beta$ are chosen identical to~~ normalized PSD. The same shape

parameters as in version 3 of the DARDAR-CLOUD product  (Cazenave et al., 2019) are chosen for frozen hydrometeors. For rain, they are chosen to match the shape used  in the GEM model for rain drops.

**Reviewer comment 5**

Line 193: wrong units

**Author response**

This will be corrected in the revised version of the manuscript.

**Changes in manuscript**

**Changes starting in line 188:**

For rain, a fixed value for $N_0^*$ of  $10^6$ m$^{-4}$ is assumed and the a priori profile for $D_m$ is determined similarly as for frozen hydrometeors.

**Reviewer comment 6**

Line 199: English

**Author response**

This will be corrected in the revised version of the manuscript.

**Changes in manuscript**

**Changes starting in line 196:**

 level. The retrieval of the $N_0^*$ parameters is further regularized by retrieving them at reduced vertical resolution of 2 km

**Author response 7**

Line 35 page 2 (not really limited,this is a wide range!!)

**Author response**

The corresponding sentence will be reformulated in the revised manuscript.

**Changes in manuscript**

> **Changes starting in line 41:**
>
>  The currently most accurate information on the global distribution of ice water content (IWC) is provided by the CloudSat radar. A main strength of these observations is their vertical resolution, in the  order of 500 m.

**Reviewer comment 8**

Line 54 page 2. maybe it is worth mentioning all the heritage coming from radar-radiometer retrievals with W-band (Ka and Ku-band) radars with PMW radiometers.

**Author response**

Following the suggestion of the reviewer, a paragraph that mentions previous work on synergistic retrievals using radar and passive radiometers at lower microwave frequencies will be added to the introduction.

**Changes in manuscript**

C.f. Comments from Referee 2 - Minor comment 5

**Reviewer comment 9**

Line 229: troposphere is too generic

**Author response**

The use of the word *troposphere* and should have been *tropopause*. This will be corrected in the revised version of the manuscript.

**Changes in manuscript**

The section has been rewritten

> **Changes starting in line 246:**
>
> Fig. 4 provides an overview of the bulk mass backscattering efficiencies and mass attenuation coefficients of the selected particles computed for three different values of the $N_0^*$ parameter of the PSD. Mass backscattering efficiency and attenuation coefficient are defined as the ratio of the corresponding cross-section $\sigma$ and the bulk water content:

**Author response 11**

250: rho is not defined

**Reviewer comment**

$\rho$ will be defined in the revised version of the manuscript.

**Changes in manuscript**

> **Changes starting in line 276:**
>
> Figure 2.2 displays the contours of $\Delta T_B$ and dBZ$_{\mathrm{max}}$ with respect to $D_m$ and the cloud's water content, which is proportional to $N_0^*$:
>
> $$\mathrm{WC} = \frac{\pi \rho}{4^4} N_0^* D_m^4, \tag{3.3}$$
>
> with $\rho$ the density of ice.

**Reviewer comment 12**

Line 4: 272.5????

**Author response**

This mistake will be corrected in the revised version of the manuscript.

**Changes in manuscript**

**Changes starting in line 186:**

$$N_0^* = \exp\left(-0.076586 \cdot (T - 272.5273.15) + 17.948\right),\qquad (3.4)$$

**Reviewer comment 10**

Fig 4 caption: you need to include how thick is the layer.

**Author response**

This will be included in the revised version of the manuscript.

**Change in manuscript**

This has been changed in the revised manuscript and the figure and caption now looks as is shown in Fig. 2.2.

**4 Marked-up differences**

[revised manuscript text omitted]

---

## Author Response (AR2)

**Response to comments**

**Synergistic radar and radiometer retrievals of ice hydrometeors**

**1 Comments from referee 1**

**1.1 General comments**

**Reviewer comment 1**

The revised manuscript is vastly improved and my concerns have mostly been satisfied in these revisions. I agree that recasting the measurements as airborne is an appropriate way to simplify the radiative transfer assumptions and demonstrate the concept. My second comment, about the inclusion of forward model error, is not directly addressed though changes in the methods but I do agree with the author response that such an error would be scene-dependent and therefore not trivial. However, I think the appropriate caveats have been mentioned in the interpretation of results so as to not interpret the combined or radiometer-only methods as being poor compared to the radar due to the higher chisquared(y) - in fact, it is noted that the radar is instead overfitting the measurements.

I only found a few instances of inconsistent nomenclature (e.g., capitalization of GEM on line 252) that should be corrected prior to publication.

**Author response**

We would like to thank the auto for pointing out the inconsistencies in nomenclature, which we will correct in the revised version of the manuscript.

**Changes in manuscript**

> **Changes starting in line 230:**
>
> GEM Graupel,  GEM Hail and GEM Cloud Ice are more efficient.

**2 Comments from referee 2**

**2.1 General comments**

**Reviewer comment 1**

L 106: that cloud ice particles are small and abundant while snow particles are large and much rare is nature not the model. So I would recommend to change the sentence: " An important characteristic can be identified here.."

**Author response**

The proposed change will be adopted in the revised manuscript.

**Changes in manuscript**

> **Changes starting in line 106:**
>
> An important characteristic  can be identified here, which will help to better understand the retrieval results presented later: Cloud ice  is characterized by high particle number concentrations and small particle sizes, whereas snow has lower number concentrations and larger particles.

**Reviewer comment 2**

L222-223: The two sentences are a bit contradictiry: Just say M1 results will be shown for certain aspects only

**Author response**

The proposed changes will be adopted in the revised manuscript.

**Changes in manuscript**

> **Changes starting in line 220:**
>
> In addition to a two-moment radar-only retrieval, also a one-moment version (M1), in which only the $D_m$ parameter is retrieved has been tested.

remaining results only the two-moment version is considered

**Reviewer comment 3**

L247 WC is used as abbreviation before being defined

**Author response**

The corresponding paragraph will be rewritten taking into account the referees comment.

**Changes in manuscript**

**Changes starting in line 247:**

Mass backscattering  and attenuation coefficients are defined as the ratio of the corresponding  backscattering or attenuation coefficient $\sigma$ and the bulk water content  WC:

$$Q = \frac{\sigma}{\text{WC}}. \tag{2.1}$$

**Reviewer comment 4**

Figure 5 caption: say that this is an ice cloud

**Author response**

The caption of Fig. 5 will be changed in the revised manuscript as shown below.

**Changes in manuscript**

**Changes starting in line 284:**

Simulated observations of a homogeneous, 5 km thick ice cloud  centered at 10 km with varying water content $m$ and mass-weighted mean diameter $D_m$. The panels display the maximum radar reflectivity (dBZ$_{\text{max}}$) overlaid onto the cloud signal ($\Delta T_B$) measured by selected radiometer channels of the MWI (first row) and ICI radiometers (second row).

**Reviewer comment 5**

I really like Fig. 12 It shows the clear contribution in DOF from the different parameters. Just as a quick idea which does not neccessaryl need to be implemented but might strengthen the discussion on LCWC: Could you look at the ratio of the the combined DOF and the sum of the the single retrievals. This could help to explain that the

[Figure]

Figure 2.1: Ratios of the DFS of the combined retrieval ($\mathrm{DFS_{CMB}}$) and the sum of the DFS or the single-instrument retrievals ($\mathrm{DFS_{RO}} + \mathrm{DFS_{PO}}$) for the two test scenes.

ice information is in both radar and passiv and therefore in the combined retrieval the nicrowave information content for ice is not needed (ice contribution is determined) and therefore the information content is transfered to LCWC is transfer. This argumention in 4.2.3 is currenttly not too strong.

**Author response**

As suggested by the referee, we have produced a plot of the ratio of the DFS of the combined retrieval and the sum of DFS of the single-instrument retrievals. Since this plot indeed strengthens our arguments on the combined information content on LCWC, we have extended the discussion in Sect. 4.2.3 as shown below.

**Changes in manuscript**

**Changes starting in line 398:**

In order to allow a more detailed analysis of the complementarity of the information in the passive and active observations, Fig. 2.1 displays the ratio of the DFS of the combined retrieval and the sum of the DFS in the radar- and passive-only retrievals. Comparison with the information content provided by the radar-only observations confirms that the active and passive observations consistently provide a fairly high amount of complementary information across both scenes.

**Changes starting in line 527:**

This conclusion is supported by the information content analysis in Fig. 12  and Fig. 13. In particular, the DFS ratio of the combined retrieval shows a distinct increase around 42 ° N, where the scene contains non-precipitating mixed-phase clouds.

> This coincides with a slight increase in information content on LCWC in the combined compared to the passive-only retrieval show in Fig. 12.

**Reviewer comment 6**

L548: "..was able TO reproduce..

**Author response**

The missing word will be added to the revised manuscript.

**Changes in manuscript**

> **Changes starting in line 548:**
>
> Moreover, the combined retrieval showed clear sensitivity to particle number concentrations and was able to reproduce their vertical structure in regions where the cloud composition ...

**Reviewer comment 6**

Table 3 caption - give symbol lq for correlation length

**Author response**

The caption of Table 3 will be corrected in the revised manuscript, as shown below.

**Changes in manuscript**

> **Changes starting in line 218:**
>
> A priori uncertainties $\sigma_q$ and correlation lengths $l_q$ used in the retrieval.

**3 Comments from referee 3**

We thank the referee for, again, identifying a number of mistakes in the manuscript. In some cases we misunderstood the comments in the previous review, but we also simply failed to implement a correction in one case. That said, co-authors (including the senior ones) read the complete revised manuscript but apparently all focused on the textual changes and the remaining problems went unnoticed. This is a good lesson for the future, that the co-authors should focus on checking different aspects of the manuscript. However, we stress that the mistakes only affect the presentation, in no case we have found any problems in the actual results.

The referee requires a major revision, but exactly why is not obvious. Our impression is that the referee would like that the study addressed different questions, and this is the main reason for his overall judgement. We would like to point out that it is our scientific freedom to decide on the focus of the study (note point three of "General obligations for referees", AMT (2020)).

In any case, the recommendation of a major revision stands in strong contrast to the opinion of the other two referees ( who mark excellent or good for all three judgement categories). We also note that the referee, both in the earlier review and this one, only give marginal credit to the original contributions of the study. To mention only the most obvious, the synergy between a cloud radar and a sub-millimetre radiometer is largely uncharted territory. The only similar journal articles on the subject we have found are the ones of Evans et al. (2005) and Jiang et al. (2019). The scope of our study is most similar to one of Jiang et al. (2019) and our study clearly provides both more quantitative results and a more direct analysis of the nature of the synergy.

**3.1 General comments**

**Reviewer comment 1**

The abstract from line 13 onwards is way too generic. There is not a single statement there that the reader can remember as specific for this work.

**Author response**

In an effort to improve the abstract we will introduce the following changes in the revised manuscript.

**Changes in manuscript**

**Changes starting in line 12:**

retrieval results is assessed for all retrieval implementations. Although they show greater sensitivity to the assumed particle shape, the synergistic  observations can better constrain the microphysics of the cloud, which decreases uncertainties in retrieved ice water content and improves the retrieval of particle number concentrations. Our results also indicate improved sensitivity to liquid cloud water content for the synergistic configuration compared to a passive-only setup. The results of this study  demonstrate the potential of the synergistic sensor configuration to improve retrievals of frozen hydrometeors. The developed synergistic retrieval algorithm can be applied with only minor modifications to suitable airborne observations from sub-millimeter radiometers such as the International Sub-Millimetre Airborne Radiometer.

**Reviewer comment 2**

"Microwave sensors employ wavelengths ranging down to about 1 mm. ... At the same time, they provide the advantage of penetrating even thick clouds." Well if this may be true for wavelength in the cm-region is certainly not true for frequency in the G-band for instance. What are "thick ice clouds"?

**Author response**

We agree with the reviewer here that some of the terminology applied here can be improved. We remain convinced, however, that the general statement made in this paragraph, i.e. that microwave sensors can sense only comparably large ice particles while optical and IR sensors can sense also small ice particles, is true and relevant to put the study in context. We therefore propose to reformulate the paragraph as presented below.

**Changes in manuscript**

**Changes starting in line 29:**

Current operational observation systems used to study clouds can be divided into two groups by virtue of their observing frequency and  corresponding capabilities and limitations. Microwave sensors employ comparably long wavelengths ranging down to about 1 mm.  Since these wavelengths are large compared to the typical sizes of ice particles  in a cloud, microwave sensors are most sensitive to the largest particles and do not provide any sensitivity to the small particles in the cloud. Optical and infrared sensors use radiation with wavelengths from around

15 $\mu$m down to several hundred nano meters.  These relatively short wavelengths make them sensitive  also to the small ice particles  in the cloud. The comparably low sensitivity of microwave sensors to small ice particles allows them to sense the larger, potentially precipitating, particles typically located at the center and base of a cloud, which cannot be sensed at infrared and optical wave lengths due to saturation of the signal.

**Reviewer comment 3**

"Compared to the sizes of ice particles, the wavelengths are very long and therefore sensitive only to very large ice particles.", again this is way too generic (what is very large?), it seems to give the idea that radars (even cloud radars) are not sensitive to 100-200 micron size particles, which is erroneous (see your Fig.5!)

**Author response**

See answer to General comment 2.

**Reviewer comment 4**

"Although radars and lidars allow detection of lower ice water contents than their passive counterparts, they are ultimately limited by the same principles."

I am not sure what the authors are alluding to here. Radars in the G-band are now a reality (e.g. see recent work by Roy et al.,) and the technology for active systems for even higher frequencies (>300 GHz) has already been demonstrated. Anyhow if a radar cannot penetrate a certain cloud the radiometer at the same frequency will suffer from the same issue; actually the radar via PIA technique can provide estimates of optical thicknesses up to 10 and more (where the radiometers are already saturated).

**Author response**

It seems the referee has misunderstood what we wanted to express with this sentence. Although our original statement is in agreement with the point put forward by the referee, we will reformulate the sentence to avoid such misinterpretation.

**Changes in manuscript**

**Changes starting in line 38:**

Active sensors have the advantage of providing high vertical resolution and generally higher sensitivity than their passive counterparts. This, however, typically comes at the expense of lower spectral and spatial coverage of the observations.

**Reviewer comment 5**

"Prominent examples of satellite missions that exploit both of these synergies....."

Again I do not see how the

"non-local synergy which uses the vertically resolved radar observations to support passive-only retrievals across the wide swath of the passive sensor".

What GPM does is use synergistic radar-radiometer retrievals to build a database for a Bayesian inversion, which is a different thing.

**Author response**

Since passive-only observations provide only limited information on the vertical distribution of hydrometeors in the atmosphere, realistic a priori assumptions are necessary in order to produce accurate retrievals of hydrometeor profiles from the passive observations of the GPM constellation. The accuracy of these retrievals directly depends on the realism of the a priori database, which is ensured by deriving it from the combined observations provided by the radar and radiometer suite on the GPM Core Observatory. Because of this, we consider the accuracy of the a priori assumptions used for the GPM passive-only retrievals a synergy between the radiometers and the radar of the constellation. However, since the observations from the Core Observatory and the other radiometers do not need to be co-located to exploit this synergy, we refer to it as the non-local synergy.

To hopefully make our reasoning more clear, we will introduce the following changes in the manuscript.

**Changes in manuscript**

> **Changes starting in line 46:**
>
> Two types of synergies can be distinguished for such an observation scenario: A local synergy, which consists of using the co-located radar and radiometer observations to obtain more accurate hydrometeor retrievals, and  a non-local synergy, which uses the vertically  well-resolved results from the radar-only or combined observations to support passive-only retrievals across the wide swath of the passive sensor, for example by providing realistic a priori constraints.

**Reviewer comment 6**

"and hence provide only limited sensitivity to frozen hydrometeors" but later on you say that MWi "will provide additional sensitivity to liquid and frozen precipitation". Again very generic statements.

**Author response**

We agree with the referee that the terminology can be made more precise. To do so, we will introduce the following changes in the revised manuscript.

**Changes in manuscript**

**Changes starting in line 53:**

Since the principal target of these missions  is the retrieval of precipitation, they make use of  comparably low microwave frequencies and hence provide only  little sensitivity to non-precipitating hydrometeors (Greenwald and Christopher, 2002).

**Reviewer comment 7**

Fig.2: I do not see any change to the figure (despite what the authors say). Units are still the same and wrong.

**Author response**

We have unfortunately missed to correct unit of the PSDs in the plot, however we will ensure that this will be corrected in the revised manuscript.

**Changes in manuscript**

Fig. 2 now looks as show in Fig. 3.1.

[Figure]

Figure 3.1: Realizations of particle size distributions from the test scenes used in this study. The particle number concentration is plotted with respect to the volume-equivalent diameter $D_{\text{eq}}$. Shown are the PSDs corresponding to 100 randomly chosen grid points with a water content higher than $10^{-6}$ kg m$^{-3}$. Line color encodes the corresponding water content. Inlets display visualizations of the particle shape assumed for each hydrometeor species.

**Reviewer comment 8**

Units of standard deviation of Z. The authors are reiterating common mistakes/typos present in literature, even when errors are highlighted. dBZ-dBZ is dB, that's the nature of logarithmic units!

**Author response**

We would like to thank the reviewer for pointing and will correct this in the revised manuscript.

**Changes in manuscript**

**Changes starting in line 138:**

The minimum sensitivity is set to be $-30$ dBZ and the noise at each range gate is modeled to be independent with standard deviation 0.5 dB.

**Reviewer comment 9**

Eq.3 : another wrong equation. Not sure what the authors are doing here but it is simple algebra, the equation is clarly wrong because there is no factor pi involved (see https://agupubs.onlinelibrary.wiley.com/doi/epdf/10.1029/2018JD028603).

**Author response**

The work referenced by the referee does in fact employ a similar transform. The transform in the referenced study, however, uses the inverse tangens, while we employ the inverse *hyperbolic* tangens. A factor of $\pi$ is therefore not required in our case.

The plot given in Fig. 3.2 shows the relation between the quantities $x$ and $RH$ in the equation and proves that the transformation does what it is expected to do: Map any arbitrary value of $x$ to the range $[0.0, 1.2]$.

**Reviewer comment 10**

Eq.6 is another example of very confused terminology (the numerator is not a cross section!! efficiency is usually used for other scattering quantities)

**Author response**

We thank the referee for pointing out the inaccurate terminology that we have applied here. The quantity $\sigma$ is indeed not a cross section and a more consistent term for the quantity denoted as "mass backscattering efficiency" is probably "mass backscattering coefficient". This will be corrected in the revised manuscript.

[Figure]

Figure 3.2: Plot of the relation between relative humidity values and the transformed quantity $x$.

**Changes in manuscript**

**Changes starting in line 242:**

Figure 4 provides an overview of the bulk mass backscattering  and attenuation coefficients of the selected particles at the frequency of the cloud radar and three selected frequencies of the passive radiometers. Mass backscattering  and attenuation coefficients are defined as the ratio of the corresponding  backscattering or attenuation coefficient $\sigma$ and the bulk water content  WC:

$$Q = \frac{\sigma}{\text{WC}}. \tag{3.1}$$

For  each particle shape and frequency, $Q$ has been computed for three different values of the $N_0^*$ parameter of the PSD. For a fixed bulk-mass, the value of the $N_0^*$ parameter of the PSD is related to the size of the bulk particles: For high $N_0^*$ values the number of large particles is decreased while it is increased for low $N_0^*$ values. The variation of the mass backscattering and attenuation coefficients with mass show the non-linear relationship between bulk mass and the particles' radiometric properties. For high values of $N_0^*$, which are typical for cloud ice, the radiometric properties of particle shapes differ only for large masses at the two highest frequencies considered. For low $N_0^*$ values, which are more typical for snow, the particles' properties differ considerably at all masses and frequencies.

**Reviewer comment 11**

Fig.4 also does not make much sense to me (if you are normalising by IWC you should not plot it in the x-axis).

**Author response**

We do not think that what the referee claims here is true. As the plots show, there is a non-linear dependency between water content and scattering properties. Normalization therefore does not cancel the dependency of the mass backscattering coefficient on the water content. Moreover, normalization is required to make the differences between different particles discernable in the plot, which would otherwise be dominated by the strong increase of the backscattering coefficient with bulk mass.

**Reviewer comment 12**

Fig.7 y-label panel b) That is not IWP but a ratio of $IWP_{ret}/IWP_{true}$. Re-label.

**Author response**

We will relabel the figure according to the referee's suggestions.

**Changes in manuscript**

Figure 7 from the manuscript now looks as shown in Fig. 3.3.

**Reviewer comment 13**

Units of IWC in Fig.14 are wrong.

**Author response**

We thank the referee for pointing out this inconsistency and will correct it in the revised manuscript.

**Changes in manuscript**

Fig. 14 now looks as shown in Fig. 3.4.

**3.2 Scientific comments**

**Reviewer comment 1**

I am still convinced about some major issues related to the selection of the scene. The key strength of the combination should be in ice clouds and mixed-phase but no clear message are coming out of this study on this (i.e. what king of cloud-LWP can we retrieve with confidence? how important is to know the location of the SLWC layer?). Profiles with high density ice and rain underneath are much more complicated and certainly should not be the target of such a suite of instruments (but they occupy a large fraction of your scene). The only semi-quantitative statements is indeed present in the conclusions "While observations at currently available microwave frequencies provide information

[Figure]

Figure 3.3: Results of the ice hydrometeor retrieval for the first test scene using the Large Plage Aggregate particle model. Panel (a) displays the value of the $\chi_y^2$ diagnostic normalized by the dimension of the measurement space of the corresponding retrieval. Panel (b) displays retrieved IWP in dB relative to the reference IWP. Reference IWP and the contributions from different hydrometeor classes are displayed by the filled areas in the background. Panel (c) shows the reference IWC from the model scene. Panel (d), (e) and (f) display the retrieval results for the passive-only, radar-only and combined retrieval, respectively.

[Figure]

Figure 3.4: Reference and retrieved CLWC and IWC. Panel (a) shows the reference and retrieved LWP for each profile. Panel (b) displays reference LWC contours drawn on top of the total hydrometeor content. Retrieval results for passive-only and combined retrieval are given in Panel (c) and (d).

complementary to that from a radar only for thick clouds with very large particles ..."
and is actually based on your Fig.5 (no retrieval involved).

**Author response**

We have justified the selection of the test scenes in our previous response: The presented
scenes contain a wide range of different cloud formations, which we chose over cherry
picking specific scenes where the combined retrieval would work well. We did this in
order to provide a more balanced view on the potential and limitations of the combined
retrieval.

The referee goes on to request a more accurate assessment of the performance of the
LWC retrieval, although we mention clearly that the main focus of our article is the
retrieval of ice hydrometeors. We therefore consider the assessment of these specific
cases to be out the scope of this article.

Furthermore, it is true that we refrain from making absolute statements on the per-
formance of the combined retrieval. This is because our intention was never to develop a
production-ready retrieval and give accurate performance estimates. Since this is study
is based purely on simulations the reliability of such an analysis would anyways be ques-
tionable. Although we already state this in the paragraph on the limitations of the study,
we will reformulate the paragraph in an effort to make this point clearer.

**Changes in manuscript**

**Changes starting in line 543:**

Moreover, this study is purely based on simulations ~~and restricted to two selected
model test scenes. The validity of~~ from two selected CRM scenes. These two
scenes are certainly insufficient to accurately represent the variability of clouds in the
atmosphere. Furthermore, the accuracy of the estimated retrieval performance will
depends on the ~~presented results thus depends on how well cloud microphysics are
represented in the GEM model. While this may affect interpretation of the results~~
realism of the test scenes. Because of this, this study does not aim to provide an
accurate assessment of the performance of the combined retrieval in absolute terms,
~~the main findings of this work, which are based on a relative comparison of the
retrieval results , should be less dependent on the realism of the test scenes~~but
instead a qualitative assessment of the potential of a combined retrieval based on
the comparison of its results to the single instrument retrievals.

**Reviewer comment 2**

Fig.3 and retrieval grid: the definition of the retrieval grid must be different if you consider
a radiometer only or a retrieval including the radar. The advantage of a radar is indeed
to produce a cloud mask first (actually with the sensitivity you have used all clouds are
practically detected by the radar). So I really do not understand how it is possible to

see IWC in Fig,.7 panel e) and f) where there are no clouds (this magically disappear in Fig.10???). On the other hand I am also curious to know how with a radiometer-only retrieval you can constrain the cloud top like you are doing.

**Author response**

As explained in the manuscript, performing the passive-only retrieval on the same grids as the combined retrieval is made possible by including spatial correlation in the a priori assumptions. Although limited, the passive observations contain some information on the vertical distribution of hydrometeors, which allows us to constrain the cloud top in the passive-only retrieval to a certain degree. Although it would have been possible to use the radar-only retrieval to produce a cloud mask, this is something we have not pursued in our implementation. All of these points are described in the section covering the retrieval implementation.

The contradicting data that the reviewer points out in Fig. 10, is due to an error in the reported masking threshold applied, which was chosen at $5 \cdot 10^{-6}$kg m$^{-3}$ and not $1 \cdot 10^{-6}$kg m$^{-3}$. We will correct this in the revised manuscript.

**3.2.1 Change in manuscript**

The following changes will be introduced in the revised manuscript:

> **Changes starting in line 355:**
>

[revised manuscript text omitted]